# Asymptotic Theory of Iterated Empirical Risk Minimization, with Applications to Active Learning

**Hugo Cui** [1]  **Yue M Lu** [2]

## Abstract

We study a class of iterated empirical risk minimization (ERM) procedures in which two successive ERMs are performed on the same dataset, and the predictions of the first estimator enter as an argument in the loss function of the second. This setting, which arises naturally in active learning and reweighting schemes, introduces intricate statistical dependencies across samples and fundamentally distinguishes the problem from classical single-stage ERM analyses. For linear models trained with a broad class of convex losses on Gaussian mixture data, we derive a sharp asymptotic characterization of the test error in the high-dimensional regime where the sample size and ambient dimension scale proportionally. Our results provide explicit, fully asymptotic predictions for the performance of the second-stage estimator despite the reuse of data and the presence of prediction-dependent losses. We apply this theory to revisit a well-studied pool-based active learning problem, removing oracle and sample-splitting assumptions made in prior work. We uncover a fundamental tradeoff in how the labeling budget should be allocated across stages, and demonstrate a double-descent behavior of the test error driven purely by data selection, rather than model size or sample count.

Let $\{x_i, y_i\}_{i \in [n]}$ be a dataset composed of $n$ data samples $x_i$ in $d$ dimensions, together with the associated labels $y_i$. We consider the problem of *two successive* Empirical Risk Minimizations (ERMs) performed *on the same dataset*, where

[1]Université Paris-Saclay, CNRS, Laboratoire de mathématiques d'Orsay, 91405, Orsay, France [2]Applied Mathematics, Harvard John A. Paulson School of Engineering and Applied Sciences. Correspondence to: Hugo Cui <hugo.cui@universite-paris-saclay.fr>.

*Proceedings of the $43^{rd}$ International Conference on Machine Learning*, Seoul, South Korea. PMLR 306, 2026. Copyright 2026 by the author(s).

the predictions $\langle \hat{w}_0, x_i \rangle$ of a first ERM estimator

$$\hat{w}_0 = \arg\min_w \frac{1}{n} \sum_{i=1}^n \ell_0(\langle w, x_i \rangle, y_i) + \frac{\lambda_0}{2} \|w\|^2 \quad (1)$$

enter as an argument in the loss function of the second ERM:

$$\hat{w} = \arg\min_w \frac{1}{n} \sum_{i=1}^n \ell(\langle w, x_i \rangle, \langle \hat{w}_0, x_i \rangle, y_i) + \frac{\lambda}{2} \|w\|^2. \quad (2)$$

One of the main motivations for the study of this class of problems in this manuscript is the analysis of *data selection schemes* : in uncertainty-based active learning (Tong & Koller, 2001; Roth & Small, 2006; Wang & Shang, 2014), an estimator is iteratively trained on growing subsets of the dataset, with the predictions of each estimator leveraged to select additional samples to include in the training set of the following estimator. We note that the multi-step generalization of such iterated ERMs also arise naturally across a wide range of important machine learning applications:

- Many classical optimization algorithms, such as Newton's method or the proximal point method (Rockafellar, 1976), can be framed as a sequence of ERMs, whose risk explicitly involves the predictions of the previous iterate.
- In the first step of an AdaBoost iteration (Freund & Schapire, 1997), samples for which the predictions of the first learner are incorrect are upweighted. The following learner is then trained on the sample-reweighted risk.

Despite its ubiquity, this class of iterated ERMs has not yet been the subject of a thorough theoretical treatment. While the properties of the first-stage estimator $\hat{w}_0$ in (1) are by now well understood in the high-dimensional regime $n \asymp d$ (Karoui, 2013; Donoho & Montanari, 2016; Thrampoulidis et al., 2018), the analysis of the second ERM iteration (2) poses qualitatively new challenges. A central difficulty stems from the explicit appearance of the first estimator $\hat{w}_0$ inside the loss function of the second ERM (2). This dependence induces intricate correlations between each single-sample loss term $\ell(\langle w, x_j \rangle, \langle \hat{w}_0, x_j \rangle, y_j)$ and

the entire dataset $\{x_i, y_i\}_{i \in [n]}$ through the definition of $\hat{w}_0$ in (1). As a result, characterizing the statistical properties of the final estimator $\hat{w}$ requires carefully unraveling and controlling these dependencies. Most existing theoretical works on iterated ERMs instead focus on the sample-splitting (or online) setting, where each ERM is trained on a fresh, independent batch of data (Chandrasekher et al., 2024; Cyffers et al., 2025; Lou et al., 2025; Kaushik et al., 2024).

The challenge of data re-use has been partially addressed in recent theoretical analyses, motivated in part by a stream of works in the statistical physics literature (Saglietti & Zdeborová, 2022; Takahashi, 2022; Takanami et al., 2025; Okajima & Takahashi, 2025) that rely on the non-rigorous replica and Franz–Parisi methods (Franz & Parisi, 1997). On the rigorous front, (Garg et al., 2025) study iterated ERMs in the specific case of the square loss, leveraging tools from random matrix theory. More recently, (Celentano et al., 2025) establish a state-evolution description for a broad class of iterated min–max optimization problems under unimodal isotropic Gaussian data, extending the Gaussian min–max theorem (Gordon, 1985; Thrampoulidis et al., 2018). While their framework is, in principle, sufficiently general to encompass the iterated ERM setting considered here, it does not provide explicit, fully asymptotic characterizations of estimator performance in terms of a finite set of scalar order parameters.

Our **main technical contribution** is the following: we establish a sharp and fully asymptotic characterization of the test error of the second-stage estimator for Gaussian mixture data and a broad class of convex loss functions in the high-dimensional regime $n \asymp d$. In contrast to existing rigorous analyses of iterated ERMs, which either focus on specific losses or do not yield explicit asymptotic predictions, our results provide asymptotic characterizations in terms of a finite set of scalar order parameters. The proof relies on a new approach based on two nested applications of the *leave-one-out technique* (Karoui, 2013), and may be of independent technical interest.

**Instantiation for active learning —** The asymptotic characterization developed above finds a natural and important application in *active learning*. In particular, it allows us to revisit a well-studied problem in pool-based data selection and to remove oracle assumptions that are pervasive in prior analyses. Given $n$ unlabeled data samples $\{x_i\}_{i \in [n]}$, practical constraints often limit labeling to only a fraction $\gamma \in (0, 1)$ of the data, yielding a training set of size $\gamma n$. This setting arises, for instance, in medical applications (Liu, 2004), where labeling is expensive (Roh et al., 2019). A classical strategy for selecting such a subset relies on training a preliminary base classifier and using its predictions to identify informative samples, most commonly those lying close to the decision boundary. A now sizable body of recent

work (Kolossov et al., 2023; Sorscher et al., 2022; Askari-Hemmat et al., 2025; Feng et al., 2024; Dohmatob et al., 2025) has analyzed this approach in the high-dimensional regime $n \asymp d$. However, these analyses typically rely on a crucial simplification: the base classifier is assumed to be either an oracle or trained on a separate held-out dataset at zero labeling cost. This assumption is restrictive in several respects. In realistic budget-constrained settings, any data used to train the base classifier must be taken from the same unlabeled pool and deducted from the labeling budget. Moreover, existing works generally discard the base training set when fitting the final classifier, thereby wasting already labeled samples. As a result, these analyses are partially oracle in nature and do not fully capture the tradeoffs inherent to budget-limited data acquisition.

Our results lift these limitations and enable a principled treatment of the fully budget-constrained setting. Given a budget $\gamma \in (0, 1)$, a fraction $\psi \in (0, \gamma)$ of the samples is first allocated to train a base classifier, whose predictions are then used to select the remaining $(\gamma - \psi)n$ samples within the budget constraint $\gamma n$. The final classifier is trained on all $\gamma n$ selected and labeled samples, allowing the initial training set of $\psi n$ samples to be fully or partially re-used. We find that this refined analysis unveils several novel phenomena of interest. Our **main findings** are:

- **Budget-allocation tradeoff.** In the case of separable data and margin-based selection (Kolossov et al., 2023; Sorscher et al., 2022), the analysis reveals that the test error exhibits a non-monotonic dependence on the fraction of data $\psi$ allocated to training the base classifier, for a fixed total budget $\gamma$. In particular, there exists an optimal allocation $\psi^\star$ for which the test error is minimized. This behavior reflects a fundamental tradeoff between allocating sufficiently many samples $\psi$ to obtain an accurate base classifier that can guide informative sample selection, and preserving a sufficient remaining budget $\gamma - \psi$ to label and include such informative samples in the final training set.
- **Real-data validation.** We illustrate that this tradeoff is not merely a theoretical artifact by demonstrating it on real datasets, using pneumonia diagnosis from chest X-rays as a representative example (Kermany et al., 2018).
- **Selection-driven double descent.** We further observe a double-descent behavior of the test error as a function of $\psi$. In contrast to classical double-descent phenomena studied in prior work (Belkin et al., 2019; Geiger et al., 2020; Nakkiran, 2019), this effect arises at fixed model capacity and fixed total sample size $\gamma n$, and is driven purely by changes in the composition of the selected training set.

The rest of the manuscript is organized as follows. Section 1

presents our main technical result, namely a sharp and rigorous asymptotic characterization of the test error of the final estimator $\hat{w}$ in the two-stage iterated ERM (2), in the high-dimensional limit $n \asymp d$. In Section 2, we instantiate this result in the context of margin-based data selection for active binary classification (Kolossov et al., 2023; Sorscher et al., 2022; Askari-Hemmat et al., 2025; Feng et al., 2024; Dohmatob et al., 2025). In particular, our framework allows us to lift oracle and sample-splitting assumptions made in prior work, enabling a principled study of the fully budget-constrained setting. Subsections 2.3 and 2.4 explore the implications of our results in this setting.

# 1. High-dimensional Asymptotics of Iterated ERMs

In this section, we present the main technical result of the manuscript, namely a sharp asymptotic characterization of the estimator $\hat{w}$ obtained from a two-stage iterated ERM in the high-dimensional regime $n \asymp d$. We begin by precisely defining the class of problems covered by our analysis.

**Data distribution** — We consider a collection of $n$ identically and independently distributed covariate–label pairs $\{(x_i, y_i)\}_{i \in [n]}$, where the labels take the form $y_i = \phi(\langle \beta, x_i \rangle, c_i, \epsilon_i)$, for a given link function $\phi$ and a vector $\beta \in \mathbb{R}^d$. The variables $c_i$ and $\epsilon_i$ respectively denote the *class label* of sample $x_i$ and a stochastic noise term. The class labels $\{c_i\}_{i \in [n]}$ are drawn i.i.d. from an arbitrary probability measure on a finite set $\mathcal{V}$. To each class $c \in \mathcal{V}$, we associate a vector $\mu_c \in \mathbb{R}^d$, and conditionally on $c_i$, the covariates are sampled as $x_i \sim \mathcal{N}(\mu_{c_i}, I_d)$. The variables $c_i$ thus induce a random partition of the dataset into $|\mathcal{V}|$ classes. These classes may, for instance, correspond to mixture components in Gaussian mixture models, in which case the vectors $\mu_c$ represent cluster centroids, and may also encode which samples are used in the first ERM, the second ERM, or both. Finally, let $\{\epsilon_i\}_{i \in [n]}$ be a collection of i.i.d. random variables capturing possible noise in the problem, and assume that all moments of $\epsilon_i$ are finite. We further assume that there exist deterministic quantities $\nu \in \mathbb{R}^{|\mathcal{V}|}$, $\rho \in \mathbb{R}^{|\mathcal{V}| \times |\mathcal{V}|}$, and $\varrho \in \mathbb{R}_+$ such that the inner products $\langle \beta, \mu_c \rangle = \nu_c, \langle \mu_c, \mu_{c'} \rangle = \rho_{cc'}, \|\beta\|^2 = \varrho$ are fixed for all $d$ as the limit $n, d \to \infty$ is taken.

**Iterated ERMs** — Given $\{x_i, c_i, \epsilon_i\}_{i \in [n]}$, we consider the following two successive ERM problems. In the first stage, we train a base estimator $\hat{w}_0$ defined as $\hat{w}_0 = \arg\min_{w \in \mathbb{R}^d} \hat{\mathcal{R}}_0(w)$, where the empirical risk is given by

$$\hat{\mathcal{R}}_0(w) = \frac{1}{n} \sum_{i \in [n]} \ell_0(\langle w, x_i \rangle, \langle \beta, x_i \rangle, c_i, \epsilon_i) + \frac{\lambda_0}{2} \|w\|^2 \quad (3)$$

The loss function $\ell_0 : \mathbb{R}^2 \times \mathcal{V} \times \mathbb{R} \to \mathbb{R}_+$ is assumed to be convex and four times differentiable in its first ar-

gument, with derivatives bounded by $O(\text{polylog}(d))$. In the second stage, the final estimator is obtained as $\hat{w} = \arg\min_{w \in \mathbb{R}^d} \hat{\mathcal{R}}(w)$, with empirical risk

$$\hat{\mathcal{R}}(w) = \frac{1}{n} \sum_{i \in [n]} \ell(\langle w, x_i \rangle, \langle \hat{w}_0, x_i \rangle, \langle \beta, x_i \rangle, c_i, \epsilon_i) + \frac{\lambda}{2} \|w\|^2.$$
(4)

Here, the loss $\ell : \mathbb{R}^3 \times \mathcal{V} \times \mathbb{R} \to \mathbb{R}_+$ is convex in its first argument and admits derivatives up to order four in its first two arguments, with all such derivatives bounded by $O(\text{polylog}(d))$. We refer the interested reader to Appendix A for a detailed list of technical assumptions, of which we only highlight the main ones here. The principal difficulty of the problem lies in the explicit dependence of the second-stage loss on the base prediction term $\langle \hat{w}_0, x_i \rangle$, which induces nontrivial correlations between each individual loss term in $\hat{\mathcal{R}}(w)$ and the entire dataset $\{x_j\}_{j \in [n]}$ through the estimator $\hat{w}_0$ trained in (3). Our main theorem characterizes test metrics associated with the second-stage estimator $\hat{w}$.

**Theorem 1.1.** *Consider any test metric*

$$\mathcal{E}_{\text{gen}} = \mathbb{E}_{x,c,\epsilon} \left[ L\left(\langle \hat{w}, x \rangle, \langle \beta, x \rangle, c, \epsilon\right) \right], \quad (5)$$

*where $L$ is $O(\text{polylog}(n))$-Lipschitz in its first variable, and there exists a polynomial $Q_L$ of $O(1)$ degree with $O(\text{polylog}(n))$ coefficients such that $\forall s, c, \epsilon, \; |L(0, s, c, \epsilon)| \le |Q_L(s, \epsilon)|$. As $n, d \to \infty$ with $\alpha = n/d = \Theta_d(1)$, the test metric $\mathcal{E}_{\text{gen}}$ converges in $L_1$ to*

$$\mathcal{E}_{\text{gen}}^* = \mathbb{E}\left[L\left(g_1, g_3, c, \epsilon\right)\right] + \tilde{O}_{L_1}\left(n^{-1/4}\right),$$

*where the $\tilde{O}_{L_1}(n^{-1/4})$ denotes a random variable whose expectation is smaller than $\text{polylog}(n)n^{-1/4}$ in absolute value. Here, $g_1$ and $g_3$ are components of a $(3 + |\mathcal{V}|)$-dimensional Gaussian random vector $g \sim \mathcal{N}(\Psi_c, \Phi)$, with mean and covariance*

$$\Psi_c = \begin{bmatrix} m_c \\ m_{0,c} \\ \nu_c \\ \\ \rho_c \end{bmatrix}, \quad \Phi = \begin{bmatrix} q & t & \theta & m \\ t & q_0 & \theta_0 & m_0 \\ \theta & \theta_0 & \varrho & \nu \\ \hline m & m_0 & \nu & \rho \end{bmatrix},$$

*conditionally on $c$. The summary statistics $q, t, \theta, m, m_0, q_0, \theta_0$ verify the self-consistent equations*

$$\theta = -1/\lambda \mathbb{E}\left[\partial_r \ell(r, u, g_3, c, \epsilon) g_3\right] + \tilde{O}\left(n^{-1/4}\right),$$
$$q = -1/\lambda \mathbb{E}\left[\partial_r \ell(r, u, g_3, c, \epsilon) r\right] + \tilde{O}\left(n^{-1/4}\right),$$
$$t = -1/\lambda \mathbb{E}\left[\partial_r \ell(r, u, g_3, c, \epsilon) u\right] + \tilde{O}\left(n^{-1/4}\right),$$
$$m_c = -1/\lambda \mathbb{E}\left[\partial_r \ell(r, u, g_3, c, \epsilon) g_c\right] + \tilde{O}\left(n^{-1/4}\right),$$

*and*

$$\theta_0 = -1/\lambda_0 \mathbb{E}\left[\partial_u \ell_0(u, g_3, c, \epsilon) g_3\right] + \tilde{O}(n^{-1/4}),$$
$$q_0 = -1/\lambda_0 \mathbb{E}\left[\partial_u \ell_0(u, g_3, c, \epsilon) u\right] + \tilde{O}(n^{-1/4}),$$
$$m_{0,c} = -1/\lambda_0 \mathbb{E}\left[\partial_u \ell_0(u, g_3, c, \epsilon) g_c\right] + \tilde{O}(n^{-1/4}).$$

*We defined the random variables* [1]

$$r = \mathrm{prox}_{V\ell(\cdot, u, g_3, \epsilon)}\left(g_1 + \chi \partial_u \ell_0(u, g_3, c, \epsilon)\right), \quad (6)$$
$$u = \mathrm{prox}_{V_0 \ell_0(\cdot, g_3, c, \epsilon)}(g_2),$$

*and introduced the auxiliary statistics* $V, V_0$ *that satisfy*

$$\frac{1}{\alpha V} = \mathbb{E}\left[\frac{\partial_r^2 \ell(r, u, g_3, c, \epsilon)}{1 + V \partial_r^2 \ell(r, u, g_3, c, \epsilon)}\right] + \lambda + \tilde{O}(n^{-1/4}),$$
$$\frac{1}{\alpha V_0} = \mathbb{E}\left[\frac{\partial_u^2 \ell_0(u, g_3, c, \epsilon)}{1 + V_0 \partial_u^2 \ell_0(u, g_3, c, \epsilon)}\right] + \lambda_0 + \tilde{O}(n^{-1/4}).$$

*The expression of* $\chi$ *is given in equation A.35 in Appendix A.* $\partial_u$ *(resp.* $\partial_r, \partial_u$*) denote the derivatives of* $\ell_0$ *(resp.* $\ell$*) with respect to its first (resp. first two) argument.*

Theorem 1.1 provides an explicit expression for any test metric $\mathcal{E}_{\text{gen}}$ of the form (5) in terms of a collection of scalar summary statistics $q, t, \theta, m_c, q_0, \theta_0, m_{0,c}$, characterized as the solutions of a system of coupled equations. This low-dimensional characterization of a learning problem in high dimensions is made possible by the asymptotic independence of the components of the response vector $X\hat{w}$, which makes it sufficient to describe the statistics of a single, scalar component. The scalar responses $u, r$ of a given component are given by equation 6, and are driven by an underlying low-dimensional Gaussian process $g$, whose covariance structure is given by the parameters $q, m, t, \theta, q_0, \theta_0, m_0$. An attentive reader will recognize that the equations governing the statistics $q_0, \theta_0, m_{0,c}$ associated with the first ERM estimator $\hat{w}_0$ recover the now classical asymptotic results for single-stage ERMs developed in, e.g., (Karoui, 2013; Donoho & Montanari, 2016; Thrampoulidis et al., 2018). The characterization of the statistics $q, \theta, m, t$ associated with the second ERM estimator $\hat{w}$ takes a remarkably similar form, and likewise involves expectations over a scalar random variable $r$ defined through a proximal map (6), in direct analogy with the role played by the variable $u$ for the first ERM. What is fundamentally new in the iterated setting is that the argument of this proximal map is no longer simply Gaussian. Instead, it consists of the sum of a Gaussian term $g_1$ and an additional correction term $\chi \partial_u \ell_0(u, g_3, c, \epsilon)$. This latter term compactly captures the intricate statistical correlations induced by the presence of the first ERM estimator $\hat{w}_0$ in the second ERM (4), whereby each individual

---

[1]We recall that the proximal map associated to a convex function $f$ is defined as $\mathrm{prox}_f(x) = \mathrm{argmin}_z \left[1/2(z-x)^2 + f(z)\right]$

single-sample loss term becomes correlated with the entire dataset through the training of $\hat{w}_0$ in (3).

The proof of Theorem 1.1 is provided in Appendix A. Recent work by (Celentano et al., 2025) develops a state-evolution analysis of iterative min-max optimization problems involving unimodal isotropic Gaussian data, providing an extension of the convex Gaussian min–max theorem (CGMT) (Gordon, 1985; Thrampoulidis et al., 2018). Much like the CGMT itself, this approach is broadly applicable and, in principle, could be specialized—through additional technical work—to the iterated ERM setting considered here. However, such a specialization is not carried out in (Celentano et al., 2025). In particular, their results do not yield explicit, fully asymptotic characterizations of estimator performance in terms of a finite collection of scalar order parameters. By contrast, the characterization of Theorem 1.1 is explicitly fully asymptotic, involves only scalar quantities, and accommodates Gaussian mixture data. This explicitness enables direct quantitative predictions of generalization performance despite data re-use and prediction-dependent losses. From a technical standpoint, our analysis follows a different proof strategy based on nested applications of the leave-one-out approach developed in, e.g., (Karoui, 2013; El Karoui, 2018). We expect this methodology to extend naturally to multi-stage iterated ERMs beyond the two-stage setting studied here. Finally, the leave-one-out method depends less critically on the assumption of a Gaussian covariate distribution. While we assume the latter in the present work for simplicity, we anticipate that the results are amenable to being relaxed to a larger class of data satisfying subgaussian concentration properties for Lipschitz functions (see e.g. (Karoui, 2013)), as well as families of elliptical distributions, as considered e.g. in (El Karoui, 2018; Adomaityte et al., 2023).

The system of self-consistent equations of Theorem 1.1 can generically be solved numerically, by repeated iteration. For some simple loss functions, such as the quadratic loss discussed in Appendix B, these equations are amenable to further simplifications, and a closed-form expression for the test error can be reached. A subtle yet crucial technical point is that the assumption of $O(\mathrm{polylog}(d))$-bounded loss derivatives does not, in its current form, allow a direct application of Theorem 1.1 to the square loss or to non-differentiable test metrics $L$. This restriction is purely technical and stems from the proof technique rather than from any intrinsic limitation of the result. Indeed, such losses and metrics can be approximated arbitrarily well as $d \to \infty$ by smooth functions satisfying the required assumptions, for instance via routine mollification arguments. In the next section, we nonetheless illustrate in Figs. 1 and 3 that even for the quadratic loss, for losses with non-continuous dependence on the second argument of $\ell$, and for non-differentiable test metrics, the characterization

of Theorem 1.1 captures numerical experiments with high accuracy in large but finite dimensions. Finally, we note that commonly used losses such as the logistic loss fully satisfy the assumptions of the theorem without modification.

The problem formulation (7)–(8) provides a flexible framework that encompasses several settings of interest, of which we briefly highlight a subset. As discussed in the introduction, the two-stage iterated ERM formulation can model the initial steps of several standard optimization algorithms. Of particular interest are settings in which the loss function factorizes as $\ell(r, u, c, s, \epsilon) = \pi(u, c, s, \epsilon) \, \tilde{\ell}(r, c, s, \epsilon)$, that is, as a base loss $\tilde{\ell}$ that is *reweighted* by a function $\pi$ depending on the base predictions. Active learning problems, such as the one described in Section 2, correspond to the case where $\pi$ takes values in $\{0, 1\}$ and indicates whether a given sample is included in the training subset. We discuss this example in detail in Section 2. Allowing $\pi$ to take more general values in $\mathbb{R}_+$ naturally leads to a broader class of sample reweighting schemes. A particularly relevant instance in binary classification corresponds to choices of the form $\pi(u, s, c, \epsilon) = \pi(\mathbb{1}_{y \neq \text{sign}(u)})$, whereby samples incorrectly classified by the base model are up-weighted. This can be viewed as a stylized model for the first step of an AdaBoost-type reweighting procedure (Freund & Schapire, 1997). Finally, while we focus on classification tasks in the remainder of this manuscript, we emphasize that Theorem 1.1 also applies to regression settings.

## 2. Applications to Active Learning

Theorem 1.1 affords a tight characterization of iterated ERMs that sharply captures the intricate correlations induced by prediction-dependent losses and data re-use. As discussed above, this class of problems arises across a wide range of machine learning tasks. In the remainder of this manuscript, we focus on a particularly notable application in the context of *pool-based active learning*. The theoretical results of Theorem 1.1 place us in a position to revisit a well-studied problem in data selection for binary classification, previously analyzed in a stream of recent works (Sorscher et al., 2022; Kolossov et al., 2023; Feng et al., 2024; Askari-Hemmat et al., 2025; Dohmatob et al., 2025) under oracle or sample-splitting assumptions. By contrast, our framework fully removes these assumptions and enables a principled treatment of the fully data-reuse, budget-constrained setting. This refined analysis further reveals several novel phenomena of interest. We begin by describing the problem in detail and situating it within the related literature.

### 2.1. Setting

Consider a simple data distribution on the sample-label pairs $(x, y) \in \mathbb{R}^d \times \{-1, 1\}$ where the samples are drawn from an isotropic Gaussian distribution $x \sim \mathcal{N}(0, I_d)$, and

where there exists a fixed vector $\beta \in \mathbb{R}^d$ such that the labels $y$ are deterministically given as $y = \text{sign}(\langle x, \beta \rangle)$. We assume the availability of $n$ independently drawn samples $\{x_i\}_{i \in [n]}$. Importantly, the associated labels are unknown, and the cost of their acquisition is supposed to be high, motivating the *budget constraint* that only a fraction $\gamma \in (0, 1)$ of the $n$ samples may be labeled. The goal of the active learning problem is then to *select an informative subset* $\mathcal{S} \subset [n]$ of size $|\mathcal{S}|/n = \gamma$ to label and use for training. Multiple active selection strategies exist (Settles, 2009), that aim at constructing a training subset $\mathcal{S}$ leading to a final test error smaller than that achieved for a subset naively sampled at random. In this work, following (Kolossov et al., 2023; Sorscher et al., 2022; Askari-Hemmat et al., 2025; Feng et al., 2024; Dohmatob et al., 2025), we focus on the following margin-based strategy.

**Base classifier —** Since it is difficult to discern informative samples from the sole knowledge of the unlabeled data $\{x_i\}_{i \in [n]}$, a natural first step consists in training a first estimator, allowing to obtain a prediction of the unknown labels, that can in turn be leveraged for selection. In order to constitute a training set on which this classifier may be trained, a first training set $\mathcal{S}_0 \subset [n]$ is uniformly subsampled from $[n]$, with each sample independently included at random according to a Bernoulli law $c_i = \mathbb{1}_{i \in \mathcal{S}_0} \sim \mathcal{B}(\psi)$, so that $\mathbb{E}[|\mathcal{S}_0|/n] = \psi$. The associated labels $\{y_i\}_{i \in \mathcal{S}_0}$ are then queried, thereby spending a first fraction $\psi \in (0, \gamma)$ of the budget. A linear base classifier with weights $\hat{w}_0$ can then be trained on the thus constituted dataset $\{x_i, y_i\}_{i \in \mathcal{S}_0}$ through the ERM $\hat{w}_0 = \arg\min_{w \in \mathbb{R}^d} \hat{\mathcal{R}}_0(w)$, with risk

$$\hat{\mathcal{R}}_0(w) = \frac{1}{\psi n} \sum_{i \in [n]} c_i \tilde{\ell}_0(\langle w, x_i \rangle, y_i) + \frac{\lambda_0}{2} \|w\|^2, \quad (7)$$

using a convex loss $\tilde{\ell}_0$, and a $\ell_2$ regularization of strength $\lambda_0$. Note importantly that only the queried labels $\{y_i\}_{i \in \mathcal{S}_0}$ actually appear in the risk with non-zero weight $c_i$.

**Data selection —** The base classifier predictions $\langle \hat{w}_0, x_i \rangle$ can then be used as the argument of a *selection policy* $\pi : \mathbb{R} \to \{0, 1\}$ to select the remaining $(\gamma - \psi)n$ samples allowed by the budget, since $|\mathcal{S}_0| = \psi n$ samples have already been labeled. For $j \in [n] \setminus \mathcal{S}_0$, the sample $x_j$ is added to the final set $\mathcal{S}$ if and only if $\pi(\langle \hat{w}_0, x_j \rangle) = 1$. We choose $\pi$ so that the budget constraint is satisfied on average, namely $\mathbb{E}\left[1/n \sum_{j \in [n] \setminus \mathcal{S}_0} \pi(\langle \hat{w}_0, x_j \rangle)\right] = \gamma - \psi$, where the expectation bears over the draw of $\{x_i\}_{i \in [n]}$ and the sampling of $\mathcal{S}_0$. Although the budget constraint is satisfied only in expectation, we show in Lemma A.36 of Appendix A that the quantity within the expectation in fact concentrates in the considered asymptotic limit under technical assumptions on $\pi$, implying that the budget constraint is in fact also satisfied with high probability. To give a concrete example, in the following, we for example discuss margin-based policies of the

form $\pi(u) = \mathbb{1}_{|u| \leq \kappa}$, for some adjustable threshold $\kappa$, with the intuition that samples closest to the decision boundary are hardest to classify, and will therefore bring the most information once labeled. At the end of the selection step, the labels of the selected samples $\{j \in [n] \setminus \mathcal{S}_0 | \pi(\langle \hat{w}_0, x_j \rangle) = 1\}$ are queried, saturating the labeling budget.

**Training —** Finally, a second linear classifier can be trained on the selected dataset $\mathcal{S} = \mathcal{S}_0 \cup \{j \in [n] \setminus \mathcal{S}_0 | \pi(\langle \hat{w}_0, x_j \rangle) = 1\}$, composed of the union of the base training set $\mathcal{S}_0$ and of the newly selected data. This ERM $\hat{w} = \arg\min_{w \in \mathbb{R}^d} \hat{\mathcal{R}}(w)$ can be written in terms of a reweighted risk

$$\hat{\mathcal{R}}(w) = \frac{1}{\gamma n} \sum_{i \in [n]} \mathbb{1}_{i \in \mathcal{S}} \tilde{\ell}\left(\langle w, x_i \rangle, y_i\right) + \frac{\lambda}{2} \|w\|^2, \quad (8)$$

where $\mathbb{1}_{i \in \mathcal{S}} = (1 - c_i)\pi(\langle \hat{w}_0, x_i \rangle) + c_i$, using a convex loss $\tilde{\ell}$, and a $\ell_2$ regularization of strength $\lambda$. Finally, the accuracy of the trained classifier $\hat{w}$ is quantified by its test error, defined as the probability to misclassify an unseen test sample $\mathcal{E}_{\text{gen}} = \mathbb{E}_{(x,y) \sim \mathbb{P}} \left[\mathbb{1}_{\text{sign}(\langle \hat{w}, x \rangle) \neq y}\right]$.

## 2.2. Related works

**Theory of active learning and subsampling —** The problem of data selection has been the object of a vast line of works in statistical learning theory, surveyed in e.g. (Settles, 2009) and (Hanneke, 2014). The majority of these works prove worst-case bounds which become void as the dimension of the data grows, and thus do not capture typical high-dimensional problems. A large body of works has similarly been devoted to the closely related problem of subsampling, with optimal subsampling probabilities derived in e.g. (Drineas et al., 2006; Ma et al., 2015; 2022) for linear regression, and (Wang et al., 2018) for logistic regression. Almost all these studies however focus on the low-dimensional regime $n \gg d$, leaving the regime $n \lesssim d$ largely uncharted.

**High-dimensional analyses of active learning —** The high-dimensional case $n \asymp d$ was studied in a number of studies, with a particular focus on supervised active binary classification with linear models. The seminal work of (Seung et al., 1992) study a disagreement-based active learning method, assuming the learner can freely query *any* sample. In contrast, in many real applications, the sample rather needs to be *chosen from a pre-existing pool* of available data. In the latter scenario, (Cui et al., 2021) conjecture a tight information-theoretic lower-bound on a notion of test error when the subset is optimally selected, but do not analyze a concrete algorithm. Closer to the present work, (Sorscher et al., 2022; Kolossov et al., 2023; Askari-Hemmat et al., 2025; Dohmatob et al., 2025) analyze the binary classification task described Subsection 2.1. However, these works all make the significant simplification that the base classifier is either an *oracle*, or trained on a *separate held-out dataset* at zero

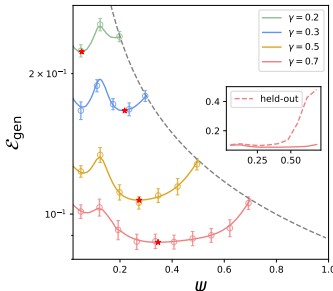

*Figure 1.* Test error $\mathcal{E}_{\text{gen}}$ of the the classifier $\hat{w}$ (8) trained on the subset $\mathcal{S}$, selected using a base classifier $\hat{w}_0$ (7) trained on a fraction $\psi \in (0, \gamma)$ of the $n$ samples for $\alpha = 8$, $\lambda = 0.01$, losses $\ell(y, z), \ell_0(y, z) = \frac{1}{2}(y - z)^2$, and policy $\pi(u) = \mathbb{1}_{|u| < \kappa^-(\gamma, \psi)}$. Different curves correspond to different total budgets $\gamma \in (0, 1)$. Solid lines: theoretical predictions. Dots: numerical experiments in dimension $d = 2000$; error bars represent one standard deviation over 20 trials. Dashed line: test error of the base classifier. Inset: for $\gamma = 0.7$, the dashed line represents the test error when *only the newly selected subset* $\mathcal{S} \setminus \mathcal{S}_0$ is used for the second ERM, with the base training set $\mathcal{S}_0$ being held-out.

label cost. In real settings however, such a dataset would instead need to be carved out from the pool of available data, and labeled at the expense of a fraction of the budget. Furthermore, these works do not consider re-using the base training set, in addition to the selected data, when training the final classifier — thereby misspending already labeled data. The eager reader may take an anticipated look at Fig. 1 (inset), which shows how holding out the base training set (dashed lines), instead of re-employing it (solid lines), may result in a significant loss in accuracy for the final model. Theorem 1.1 enables the removal of those assumptions, and the treatment of the fully budget-constrained problem.

**Margin-based active learning with linear classifiers —** Let us mention that outside of the pool-based setting investigated in the present work, the problem of margin-based methods for binary classification with linear classifiers has traditionally constituted a central question in uncertainty-based active learning (Dasgupta, 2004; Balcan et al., 2007; Balcan & Long, 2013), with a particular focus on streaming (online) active learning algorithms (Dasgupta et al., 2005; Cesa-Bianchi et al., 2009; Dekel et al., 2010; Orabona et al., 2011; Raj & Bach, 2022; Wang & Singh, 2016).

## 2.3. Tradeoff in budget allocation

An attentive reader might have already realized that the budget-constrained active binary classification task described in Subsection 2.1 in fact constitutes a special instantiation of the generic class of iterated ERM problems (1), (2) described in Section 1, with first loss $\ell_0(u, s, c, \epsilon) = c\tilde{\ell}_0(u, \text{sign}(s))$ and second loss $\ell(r, u, s, c, \epsilon) = (\pi(u) + c)\tilde{\ell}(r, \text{sign}(s))$. As such, the test error of the final classifier is sharply described in the high-dimensional regime

$n \asymp d$ by Theorem 1.1, in the generic case with data re-use and without making sample-splitting or oracle assumptions, thereby lifting a core limitation of (Sorscher et al., 2022; Kolossov et al., 2023; Feng et al., 2024; Dohmatob et al., 2025; Askari-Hemmat et al., 2025). In this Subsection, we explore the rich implications of this characterization.

Fig. 1 presents the theoretical predictions of Theorem 1.1 for the active learning example of Section 2, and contrast them to numerical experiments in large but finite dimension $d = 1000$, revealing good agreement. The illustrated setting corresponds to quadratic losses $\ell(y, z), \ell_0(y, z) = 1/2(y - z)^2$, regularizations $\lambda = \lambda_0 = 0.01$, and a margin-based selection policy $\pi(u) = \mathbb{1}_{|u| < \kappa^-(\gamma, \psi)}$, with a threshold $\kappa^-(\gamma, \psi) = \sqrt{2q_0}\mathrm{erf}^{-1}\left(\gamma - \psi/1 - \psi\right)$ chosen so as to saturate the budget constraint with high probability. Fig. 1 represents the test error $\mathcal{E}_{\mathrm{gen}}$ of the final classifier, as a function of fraction of the budget $\psi$ allocated to training the base classifier, with different curves corresponding to different total budget $\gamma$. In particular, the rightmost point $\psi = \gamma$ of each curve corresponds to the performance of a classifier trained on a passively (i.e. randomly) selected subset $\mathcal{S}$ of size $\gamma n$, and exhibits higher test error than most cases where $\mathcal{S}$ is in contrast actively chosen ($\psi < \gamma$).

**A tradeoff in budget allocation** — A first striking feature of the learning curves is their non-monotonicity, revealing the existence of an optimal budget allocation $\psi^\star$ (red star in Fig. 1) for which the test error $\mathcal{E}_{\mathrm{gen}}$ is minimized. This U-shape results from a fundamental tradeoff in the allocation $\psi, \gamma - \psi$ of the labeling budget $\gamma$ across the two ERMs (7) and (8):

- If $\psi$ is chosen too small, the base classifier will yield poor predictions, which results in turn in the selection of scarcely informative samples to include in $\mathcal{S}$.
- Conversely, if too much budget $\psi$ is spent on constituting the training set $\mathcal{S}_0$ of the base classifier, while the accurate base classifier does allow to correctly identify the most valuable samples among the remaining unlabeled samples, only a few of those can be labeled and included in $\mathcal{S}$, due to the scarce $\gamma - \psi$ budget remaining. As a result, the selected subset $\mathcal{S}$ will be in majority composed of the randomly subsampled $\mathcal{S}_0$, with only a few actually selected samples, eventually yielding marginally better-than-passive performance.

This fundamental tension between these two effects was absent from prior works (Kolossov et al., 2023; Sorscher et al., 2022; Feng et al., 2024; Askari-Hemmat et al., 2025; Dohmatob et al., 2025).

**A selection-driven double-descent** — The learning curves in Fig.1 also exhibit a characteristic double-descent shape, where two successive phases of decrease of the test error $\mathcal{E}_{\mathrm{gen}}$ are separated by an interpolation peak. Double-descent

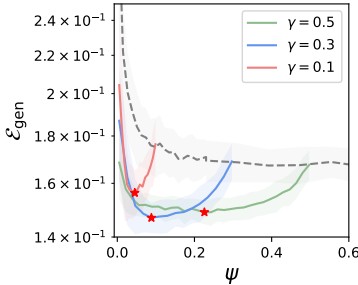

*Figure 2.* Test error as a function of the fraction $\psi \in (0, \gamma)$ allocated to the first stage of ERM, for the pneumonia diagnosis on the chest X-ray dataset (Kermany et al., 2018), pre-processed through a scattering transform with scale 4 and 6 angles (Andreux et al., 2020). The logistic loss $\ell(z, y) = \ell_0(z, y) = \log(1 + \exp(-yz))$ was employed, with $\alpha = 5.41, \lambda = 0.01$. The samples closest to the decision boundary of the first classifier were retained, until the budget was saturated. Different curves represent different total budget $\gamma$. Shades represent one standard deviation over 100 trials.

phenomena have been the object of renewed interest in recent years, and extensively described in a rich body of literature, see e.g. (Belkin et al., 2019; Geiger et al., 2020; Nakkiran, 2019). Such double-descent are however classically observed as a function of the number of parameters of the classifier, or the number of training samples. In stark contrast, in the present setting, both the number of model parameters ($d$ for linear methods) and number of training samples ($|\mathcal{S}| = \gamma n$) are *fixed*, and only the *composition* of the training set $\mathcal{S} \subset [n]$ varies along the curves of Fig. 1. The double-descent is here in fact driven by the way the selected subset $\mathcal{S}$ is constructed. Close to $\psi = 1/\alpha$, the base classifier overfits, causing in turn a biased selection of samples, eventually resulting in a poor accuracy.

**Real data** — While Theorem 1.1 describes the precision-budget tradeoff in simple synthetic settings for Gaussian data, the phenomenon can also be observed for real datasets. We illustrate this point in Fig. 2, for the task of pneumonia diagnosis on a dataset of chest X-ray images (Kermany et al., 2018). This dataset furthermore illustrates in timely fashion an application case where the labeling of the samples was in fact expensive and carried out by several human clinicians, and for which active learning is therefore directly relevant. The images were processed by a scattering transform (Andreux et al., 2020), and a subset selected using the same procedure as the one described in Section 2, using logistic regression. A final classifier was fitted on the selected set using logistic regression. The code for this experiment is provided in this online repository. The resulting learning curve display a clear U-shape, strongly suggesting the presence of the tradeoff described above.

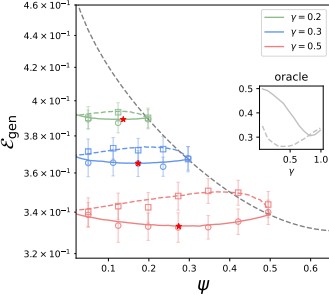

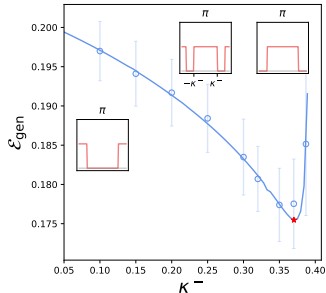

*Figure 3.* Same setting as Fig. 1, with $\alpha = 1$. Dashed curves represent the selection policy $\pi(u) = \mathbb{1}_{|u|>\kappa^+(\gamma,\psi)}$, where $\kappa^+(\gamma,\psi) = \sqrt{2q_0}\mathrm{erf}^{-1}\left(1-\gamma/1-\psi\right)$, corresponding to selecting the samples with higher margin. Solid lines still represent the small-margin selection policy $\pi(u) = \mathbb{1}_{|u|<\kappa^-(\gamma,\psi)}$. The inset represents the case where oracle access of $\beta$ is available, so that no budget $\psi = 0$ needs to be used to train the base classifier, and the data can be selected directly using the true margins as $\pi(\langle\beta, x_i\rangle)$.

*Figure 4.* Test error $\mathcal{E}_{\mathrm{gen}}$ achieved by the second classifier, with $\alpha = 8, \gamma = 0.3, \psi = 0.1, \lambda_0 = \lambda = 0.001$, and square loss. A selection policy $\pi(u) = \mathbb{1}_{|u|<\kappa^-} + \mathbb{1}_{|u|>\kappa^+}$ was used, for $\kappa^- \in (0, \kappa^-(\gamma,\psi))$, and $\kappa^+$ determined from $\kappa^-$ by requiring that the budget constraint should be saturated. Solid lines: characterization of Theorem 1.1. Dots: numerical experiments in dimension $d = 3000$. Error bars represent one standard deviation over 50 trials.

## 2.4. On the selection policy

Having explored the effect of the choice of the budget allocation $\psi$ for the training of the base estimator, we now discuss the second aspect of the active learning algorithm, namely the choice of the selection policy $\pi$. In the likeness of many prior works, (Kolossov et al., 2023; Sorscher et al., 2022), the previous section focused on the particular case of margin-based selection, where samples whose label prediction by the base classifier were smallest in absolute value were retained. This natural choice for linear classifiers corresponds to the intuition that labeling samples closest to the base estimator's decision boundary, and thus a priori hardest and most uncertain, lead to greater information gain when labeled.

While (Dohmatob et al., 2025) prove for linear regression in the population case $\alpha \to \infty$ that selecting small-margin samples is optimal, the answer is more nuanced at finite sample complexity. (Sorscher et al., 2022) have shown that there exist regimes where conversely, selecting *larger* margin samples far from the decision boundary can prove beneficial. This was found to be the case in particular for small budget regimes with small values of $\alpha\gamma$, provided a good estimate $\hat{w}_0$ remains available from a held-out dataset, or from an oracle. In such settings, the large margin samples indeed provide a clearer sketch of the geometry of the target function, and aid the alignment of the final weights $\hat{w}$ with $\hat{w}_0$ and $\beta$. These conclusions, which also echo the findings of (Hacohen et al., 2022) for a related problem, however need to be nuanced in truly budget-constrained settings, where a small budget rules out the constitution of a large held-out set on which a base estimator may be trained up to satisfactory accuracy. In such a case, selecting large-margin samples might reversely *detrimentally bias* the final classifier towards learning an erroneous direction $\hat{w}_0$, rather than the ground-truth

$\beta$. This is illustrated in Fig. 3, where the large-margin policy (dashed) consistently leads to worse accuracy than the small-margin policy (solid lines), despite performing better in the oracle case where a good base estimator is available despite the limited data budget (inset). The latter case corresponds to the observations of (Sorscher et al., 2022; Kolossov et al., 2023; Hacohen et al., 2022). This comparison highlights fundamental discrepancies between the fully budget-constrained setting characterized by Theorem 1.1, and cases where an oracle or an additional held-out set is available.

These observations naturally prompt the question : for given models $\tilde{\ell}_0, \lambda_0, \tilde{\ell}, \lambda$ and budgets $\gamma, \psi$, *what is then the optimal selection policy $\pi$?* The answer to this question is generically complex. Fig. 4 evidences that for certain choices of $\gamma, \psi$, a mixed strategy of the form $\pi(u) = \mathbb{1}_{|u|<\kappa^-} + \mathbb{1}_{|u|>\kappa^+}$ that retains a mixture of samples both far and close to the decision boundary can achieve a better accuracy than naively selecting only the farthest or closest samples. This finding echoes similar observations from (Kolossov et al., 2023; Hacohen & Weinshall, 2023).

Looking ahead, the theoretical characterization of Theorem 1.1 provides an explicit expression of the test error $\mathcal{E}_{\mathrm{gen}}$ as a functional of the selection policy $\pi$, which can in principle be optimized over. While this analytical objective represents a first step forward since it prescinds the need for expensive empirical search, this optimization remains very challenging, since it involves a functional minimization over a discrete-valued function. Such an optimization problem should prove on the other hand more tractable in the closely related setting of *subsampling*, where the policy $\pi$ assigns a continuous subsampling probability, and is thus amenable to easier analysis. Solving this problem constitutes an open question of significant interest for future research.

## 2.5. Data pruning

For completeness, and in an effort to build further connections with prior works (Kolossov et al., 2023; Askari-Hemmat et al., 2025; Dohmatob et al., 2025; Hacohen et al., 2022) we briefly discuss in this last subsection the case of *data pruning*. In this paragraph only, we assume all labels are available, and the base classifier is trained on the whole dataset, namely $\mathcal{S}_0 = [n]$, then used to select a final and *smaller* training subset $\mathcal{S} \subset [n]$ through a selection (pruning) policy $\pi$. Several of the above works report settings in which training a model on the smaller subset $\mathcal{S} \subset [n]$ can yield *better* accuracy than training on the full dataset, when an oracle base classifier is used for selection. In Appendix C, we show by instantiating Theorem 1.1 in two examples (see Fig. 5 and Fig. 6) that this phenomenon can still be observed in the more realistic setting when the base classifier is used to prune the *same* dataset it was trained on.

## Conclusions

We study two iterated ERMs on the same dataset, where the predictions of the first ERM estimator enter the loss function of the second ERM. The dependence of the loss on the predictions and the re-use of the data introduce intricate statistical correlations across samples which fundamentally set the problem apart from the now mature study of single-stage ERM. For Gaussian mixture data and a large class of convex losses, in the asymptotic regime where the number of samples $n$ and the dimension of the covariates $d$ diverge at proportional rate $n \asymp d$, we prove a sharp asymptotic characterization of the statistics of the second-stage ERM estimator. Our proof builds on a sequential version of the standard leave-one-out method (Karoui, 2013). These results find a notable application in the theory of pool-based active learning. Our analysis lets us revisit a well-studied problem in active binary classification (Sorscher et al., 2022; Kolossov et al., 2023; Feng et al., 2024; Askari-Hemmat et al., 2025; Dohmatob et al., 2025), and fully remove sample splitting and oracle assumptions made in prior work. The analysis of the budget-constrained case further uncovers a tradeoff in the allocation of the data acquisition budget, which is also observed in real-data experiments. It also unveils a double-descent behavior of the test error as a function of the budget allocation, which in contrast to prior studies occurs at fixed sample and model size.

## Acknowledgements

The work of Y.M.L. is supported by a Harvard College Professorship, by the Harvard FAS Dean's Fund for Promising Scholarship, and by DARPA under grant DIAL-FP-038.

## Impact statement

The present work aims at advancing the theoretical understanding of machine learning algorithms. There exist a number of possible impacts, none of which we feel needs to be highlighted.

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

# A. Proof of Theorem 1.1

## A.1. Notations and assumptions

### Notations

- For two random variables $a, b$ and a integer $k \in \mathbb{N}$, we write $a = O_{L_k}(b)$ if $\mathbb{E}\left[|a|^k\right] \leq \mathbb{E}\left[|b|^k\right]$. If $k$ is not specified in the statement, the latter is understood to hold for any $k \in \mathbb{N}$.

- In the following, for $a, b \in \mathbb{R}$, $(a, b)$ denotes the unordered interval.

- For $i \in [n]$, we denote $X_{\setminus i} \in \mathbb{R}^{(n-1) \times d}$ the leave-one-row-out matrix constructed by removing the $i-$th row from the full data matrix $X$.

*Remark* A.1 (Properties of $O_{L_k}$). We remind for convenience the following properties of the $O_{L_k}$ notation.

- If $\alpha = O_{L_{2k}}(a_n), \beta = O_{L_{2k}}(b_n)$, for $a_n, b_n$ two positive sequences, it follows from Minkowski's inequality that $\alpha\beta = O_{L_k}(a_n b_n)$.

- Let $\alpha, \beta$ be two positive random variables bounded as $\alpha = O_{L_k}(a_n), \beta = O_{L_k}(b_n)$, for $a_n, b_n$ two positive sequences. Then it follows from Hölder's inequality that $\alpha + \beta = 2O_{L_k}(a_n \vee b_n)$. As a consequence, from the identity $\sqrt{x} \leq 1 + x$ for all $x \geq 0$, it follows that $\sqrt{\alpha} = 2O_{L_k}(\sqrt{a_n})$.

**Assumption A.2** (Base loss $\ell_0$). Let $\ell_0 : \mathbb{R} \times \mathbb{R} \times \mathcal{V} \times \mathcal{E} \rightarrow \mathbb{R}_+$ denote the loss used for the base estimator. We assume,

(A.1) $\ell_0$ is everywhere three times differentiable in its first argument. We denote by $\partial_u$ the derivative with respect to the latter.

(A.2) $\ell_0$ is convex in its first argument, namely $\partial_u^2 \ell_0$ is everywhere positive.

(A.3) The derivatives $\partial_u \ell_0, \partial_u^2 \ell_0, \partial_u^3 \ell_0$ are everywhere well defined and bounded by $O(\mathrm{polylog}(n))$.

(A.4) The reduced function $\tilde{\ell}_0 : s, c, \epsilon \rightarrow \ell_0(0, s, c, \epsilon)$ is bounded by an $O(\mathrm{polylog}(n))$.

**Assumption A.3** (Reweighted loss $\ell$). Let $\ell : \mathbb{R} \times \mathbb{R} \times \mathbb{R} \times \mathcal{V} \times \mathcal{E} \rightarrow \mathbb{R}_+$ denote the loss used for the reweighted problem.

(B.1) The derivatives $\partial_r \ell, \partial_r^2 \ell, \partial_r^3 \ell, \partial_r^4 \ell, \partial_r \partial_u \ell, \partial_r \partial_u^2 \ell, \partial_r^2 \partial_u \ell, \partial_r^3 \partial_u \ell, \partial_r^2 \partial_u^2 \ell$ are everywhere well defined, and bounded by $O(\mathrm{polylog}(n))$. We denoted by $\partial_r, \partial_u$ the derivatives with respect to the first and second arguments of $\ell$.

(B.2) $\ell$ is convex in its first argument, namely $\partial_r^2 \ell$ is everywhere positive.

(B.3) The reduced function $\tilde{\ell} : u, s, c, \epsilon \rightarrow \ell(0, u, s, c, \epsilon)$ is bounded by $O(\mathrm{polylog}(n))$.

In the following, for conciseness, we adopt the shorthands $\ell_i(r_i, u_i) = \ell(r_i, u_i, s_i, c_i, \epsilon_i)$ and $\ell_{0,i}(u_i) = \ell_0(u_i, s_i, c_i, \epsilon_i)$.

**Assumption A.4** (Data distribution). We make the following assumptions on the sample distribution.

(C.1) The latent classes $\{c_i\}_{i \in [n]}$ are drawn i.i.d from an arbitrary distribution on a finite set $\mathcal{V}$.

(C.2) Likewise, the stochastic noises $\{\epsilon_i\}_{i \in [n]}$ follow a distribution on $\mathcal{E} \subset \mathbb{R}$, with all moments being finite.

(C.3) Conditional on $\{c_i\}_{i \in [n]}$, the samples $z_i = x_i - \mu_{c_i}$ have i.i.d unit variance Gaussian entries.

(C.4) We further assume that the inner products $\rho_{c,k} = \langle \mu_c, \mu_k \rangle, \nu_c = \langle \mu_c, \beta \rangle, \varrho = \|\beta\|^2$ are fixed absolute constants. In particular, there thus exist absolute constants $M, B$ such that $\|\mu_c\| \leq M, \|\beta\| \leq B$ for all $c \in \mathcal{V}$ and for all $n$. Furthermore, the Gram matrix

$$A = \left[\begin{array}{c|c} \varrho & \nu \\ \hline \nu & \rho \end{array}\right].$$

has a minimal eigenvalue bounded away from $0$ by an absolute constant. It will also prove convenient to introduce the matrix $\mathcal{M} \in \mathbb{R}^{|\mathcal{V}| \times d}$ with rows $\mu_1, ..., \mu_{|\mathcal{V}|}$.

Finally, we assume throughout the proof that the regularizations are strictly positive, namely $\lambda, \lambda_0 > 0$.

### A.1.1. BASE ERM

We remind the base ERM

$$\hat{w}_0 = \underset{w}{\operatorname{argmin}} \, \hat{\mathcal{R}}_0(w), \qquad \hat{\mathcal{R}}_0(w) = \frac{1}{n} \sum_{j \in [n]} \ell_{0,j}(\langle w, x_j \rangle) + \frac{\lambda_0}{2} \|w\|_2^2$$

and for any $i \in [n]$ the leave-one out surrogate

$$\hat{w}_{0,\backslash i} = \underset{w}{\operatorname{argmin}} \, \hat{\mathcal{R}}_{0,\backslash i}(w), \qquad \hat{\mathcal{R}}_{0,\backslash i}(w) = \frac{1}{n} \sum_{j \neq i} \ell_{0,j}(\langle w, x_j \rangle) + \frac{\lambda_0}{2} \|w\|_2^2.$$

We define the full and leave-one-out residuals

$$u_j = \langle \hat{w}_0, x_j \rangle, \qquad u_{j,\backslash i} = \langle \hat{w}_{0,\backslash i}, x_j \rangle,$$

as well as the surrogate estimator

$$\tilde{w}_{0,i} = \hat{w}_{0,\backslash i} - \frac{1}{n} \partial_u \ell_{0,i}(\tilde{u}_{i,i}) H_{0,\backslash i}^{-1} x_i$$

where we introduced the leave-$i$ out Hessian $H_{0,\backslash i}$ as

$$H_{0,\backslash i} = \frac{1}{n} \sum_{j \neq i} \partial_u^2 \ell_{0,j}(u_{j,\backslash i}) x_j x_j^\top + \lambda_0 I_d$$

and denoted

$$\tilde{u}_{i,i} = \operatorname{prox}_{\gamma_{0,i} \ell_{0,i}(\cdot)}(u_{i,\backslash i}), \qquad \gamma_{0,i} = \frac{1}{n} x_i^\top H_{0,\backslash i}^{-1} x_i.$$

Finally, for $j \neq i$, we similarly define the surrogate residual

$$\tilde{u}_{j,i} = \langle \tilde{w}_{0,i}, x_j \rangle.$$

For simplicity, we will also adopt the shorthand

$$\xi_i = -\frac{1}{n} \partial_u \ell_{0,i}(\tilde{u}_{i,i}) H_{0,\backslash i}^{-1} x_i,$$

so that $\tilde{w}_{0,i}$ may be more compactly rewritten as $\tilde{w}_{0,i} = \hat{w}_{0,\backslash i} + \xi_i$. It will finally also prove convenient to introduce the surrogate Hessian

$$\tilde{H}_{0,i} = \frac{1}{n} \sum_{j \in [n]} \partial_u^2 \ell_{0,j}(\tilde{u}_{j,i}) x_j x_j^\top + \lambda I_d.$$

### A.1.2. REWEIGHTED ERM

By the same token, for the reweighted problem

$$\hat{w} = \underset{w}{\operatorname{argmin}} \, \hat{\mathcal{R}}(w), \qquad \hat{\mathcal{R}}(w) = \frac{1}{n} \sum_{j \in [n]} \ell_j(\langle w, x_j \rangle, u_j) + \frac{\lambda}{2} \|w\|_2^2$$

we define the leave-one out problem

$$\hat{w}_{\backslash i} = \underset{w}{\operatorname{argmin}} \, \hat{\mathcal{R}}_{\backslash i}(w), \qquad \hat{\mathcal{R}}_{\backslash i}(w) = \frac{1}{n} \sum_{j \neq i} \ell_j(\langle w, x_j \rangle, u_{j,\backslash i}) + \frac{\lambda}{2} \|w\|_2^2.$$

Let us also introduce the surrogate risk

$$\tilde{\mathcal{R}}_i(w) = \frac{1}{n} \sum_{j \in [n]} \ell_j(\langle w, x_j \rangle, \tilde{u}_j) + \frac{\lambda}{2} \|w\|_2^2.$$

Note that despite the close notations, $\tilde{w}_i$ is *not* the minimizer of $\tilde{\mathcal{R}}_i$. We define the full and leave-one-out residuals

$$r_j = \langle \hat{w}, x_j \rangle, \qquad\qquad r_{j,\backslash i} = \langle \hat{w}_{\backslash i}, x_j \rangle,$$

as well as the surrogate estimator

$$\tilde{w}_i = \hat{w}_{\backslash i} - \frac{1}{n} \partial_r \ell_i(\tilde{r}_{i,i}, \tilde{u}_{i,i}) H_{\backslash i}^{-1} x_i + \frac{1}{n} \partial_u \ell_{0,i}(\tilde{u}_{i,i}) H_{\backslash i}^{-1} G_{\backslash i} H_{0,\backslash i}^{-1} x_i$$

where we introduced the leave-$i$ out Hessians $H_{\backslash i}, G_{\backslash i}$ as

$$H_{\backslash i} = \frac{1}{n} \sum_{j \neq i} \partial_r^2 \ell_j(r_{j,\backslash i}, u_{j,\backslash i}) x_j x_j^\top + \lambda I_d,$$

$$G_{\backslash i} = \frac{1}{n} \sum_{j \neq i} \partial_u \partial_r \ell_j(r_{j,\backslash i}, u_{j,\backslash i}) x_j x_j^\top$$

and denoted

$$\tilde{r}_{i,i} = \text{prox}_{\gamma_i \ell_i(\cdot, \tilde{u}_{i,i})} \left( r_{i,\backslash i} + \partial_u \ell_{0,i}(\tilde{u}_{i,i}) v_i \right),$$

$$\gamma_i = \frac{1}{n} x_i^\top H_{\backslash i}^{-1} x_i,$$

$$v_i = \frac{1}{n} x_i^\top H_{\backslash i}^{-1} G_{\backslash i} H_{0,\backslash i}^{-1} x_i.$$

We also introduce the full Hessian

$$H = \frac{1}{n} \sum_{j \in [n]} \partial_r^2 \ell_j(r_j, u_j) x_j x_j^\top + \lambda I_d,$$

and the surrogate Hessians

$$\tilde{H}_i = \frac{1}{n} \sum_{j \neq i} \partial_r^2 \ell_j(\tilde{r}_{j,i}, \tilde{u}_{j,i}) x_j x_j^\top + \lambda I_d, \qquad\qquad \tilde{G}_i = \frac{1}{n} \sum_{j \neq i} \partial_r \partial_u \ell_j(\tilde{r}_{j,i}, \tilde{u}_{j,i}) x_j x_j^\top$$

In the following, we adopt the shorthands

$$\eta_i = -\frac{1}{n} \partial_r \ell_i(\tilde{r}_{i,i}, \tilde{u}_{i,i}) H_{\backslash i}^{-1} x_i,$$

$$\zeta_i = \frac{1}{n} \partial_u \ell_{0,i}(\tilde{u}_{i,i}) H_{\backslash i}^{-1} G_{\backslash i} H_{0,\backslash i}^{-1} x_i,$$

so that $\tilde{w}_i$ may be more compactly written as $\tilde{w}_i = \hat{w}_{\backslash i} + \eta_i + \zeta_i$. Finally, for $j \neq i$, we similarly define the surrogate residual

$$\tilde{r}_{j,i} = \langle \tilde{w}_i, x_j \rangle.$$

## A.2. Approximation results

In this first part of the proof, we show that the surrogates $\tilde{w}_i$ (resp. $\tilde{w}_{0,i}$) provide $O(\text{polylog}(n)/n)$ approximations of the full minimizers $\hat{w}$ (resp. $\hat{w}_0$) in Euclidean norm. We start from the following approximation results for the base estimator:

**Proposition A.5.** *(Approximations of the base estimator) For any $k \in \mathbb{N}$,*

$$\sup_{i \in [n]} \|\hat{w}_0 - \tilde{w}_{0,i}\| = O_{L_k}\left(\frac{\text{polylog}(n)}{n}\right).$$

*We also have*

$$\sup_{i \in [n]} \|\tilde{w}_{0,i} - \hat{w}_{0,\backslash i}\| = \sup_{i \in [n]} \|\xi_i\| = O_{L_k}\left(\frac{\text{polylog}(n)}{\sqrt{n}}\right).$$

*Furthermore, at the level of the residuals,*

$$\sup_{i \in [n]} \sup_{j \neq i} |u_{j,\backslash i} - u_j| = O_{L_k}\left(\frac{\text{polylog}(n)}{\sqrt{n}}\right)$$

*and*

$$\sup_{i \in [n]} |u_i - \tilde{u}_{i,i}| = O_{L_k}\left(\frac{\text{polylog}(n)}{\sqrt{n}}\right),$$

*and finally*

$$\sup_{i \in [n]} \sup_{j \neq i} |\tilde{u}_{j,i} - u_{j,\backslash i}| = O_{L_k}\left(\frac{\text{polylog}(n)}{\sqrt{n}}\right).$$

The following Lemma is a direct consequence of Proposition A.5, and complements it for the cross terms

**Lemma A.6.** *We have*

$$\sup_{i \in [n]} \sum_{j \neq i} (u_j - \tilde{u}_{j,i})^2 = O_{L_k}\left(\frac{\text{polylog}(n)}{n}\right).$$

*Proof.* Note that we can rewrite more compactly

$$\sqrt{\sup_{i \in [n]} \sum_{j \neq i} (u_j - \tilde{u}_{j,i})^2} = \sup_{i \in [n]} \left\|X_{\backslash i}(\hat{w}_0 - \tilde{w}_{0,i})\right\| \leq \sup_{i \in [n]} \frac{\|X_{\backslash i}\|}{\sqrt{n}} \sqrt{n} \sup_{i \in [n]} \|\hat{w}_0 - \tilde{w}_{0,i}\|$$

$$= \sup_{i \in [n]} \frac{\|X_{\backslash i}\|}{\sqrt{n}} O_{L_k}\left(\frac{\text{polylog}(n)}{\sqrt{n}}\right)$$

using Proposition A.5 for the last step. The control of the remaining term follows from Lemma A.40 and yields $\sup_{i \in [n]} \|X_{\backslash i}\|/\sqrt{n} = O_{L_k}(\text{polylog}(n))$, completing the proof. $\qquad\square$

Note that Proposition A.5 does not immediately follow from (El Karoui, 2018), as the latter in particular assumes the samples to have zero mean, and a simpler structure of the losses.

*Remark* A.7. It is an important observation that the proof of Proposition A.5 in fact follows from the one reported below for the reweighted ERM. To see this, formally introduce an additional trivial ERM with zero loss $\ell_{-1,j}(\cdot) = 0$ and $\lambda_{-1} = 1$. In particular, this loss satisfies the set of assumptions A.2. The resulting weight minimizer is trivially zero $\hat{w}_{-1} = 0_d$. The base ERM can then be viewed as a ERM reweighted by the (vanishing) predictions of this trivial ERM, and the proof below applies, exchanging the roles of the base / reweighted ERM sequence with the trivial/base ERM sequence. Note that there is no circularity in this argument, as Proposition A.5 trivially holds for $\ell_{-1,j}(\cdot) = 0$ and $\lambda_{-1} = 1$. Therefore the following proof can be applied first to prove Proposition A.5 for the base estimator, which can then in turn be used as a starting point for the reweighted estimator, following a direct re-application of the proof.

**Lemma A.8.** *The distance between the full and surrogate estimators admits the upper bound*

$$\sup_{i \in [n]} \|\hat{w} - \tilde{w}_i\| \leq \frac{1}{\lambda} \sup_{i \in [n]} \left\|\nabla \tilde{\mathcal{R}}_i(\tilde{w}_i)\right\| + \frac{1}{\lambda} \|\partial_r \partial_u \ell\|_\infty O_{L_k}\left(\frac{\text{polylog}(n)}{n}\right)$$

*Proof.* We have

$$
\nabla\tilde{\mathcal{R}}_i(\tilde{w}_i) = \nabla\tilde{\mathcal{R}}_i(\tilde{w}_i) - \nabla\hat{\mathcal{R}}_i(\hat{w})
$$

$$
= \frac{1}{n}\sum_{j\in[n]}\left[\partial_r^2\ell_j(\check{r}_{j,i},\check{u}_{j,i})\langle\tilde{w}_i-\hat{w},x_j\rangle + \partial_r\partial_u\ell_j(\check{r}_{j,i},\check{u}_{j,i})(\tilde{u}_{j,i}-u_j)\right]x_j + \lambda(\tilde{w}_i-\hat{w}),
$$

for some $(\check{r}_{j,i},\check{u}_{j,i})$ lying on the segment connecting $(\tilde{r}_{j,i},\tilde{u}_{j,i})$ and $(r_{j,\backslash i},u_{j,\backslash i})$. Introducing

$$
\check{H} = \frac{1}{n}\sum_{j\in[n]}\partial_r^2\ell_j(\check{r}_{j,i},\check{u}_{j,i})x_jx_j^\top + \lambda I_d
$$

one can write

$$
\tilde{w}_i-\hat{w} = \check{H}^{-1}\left[\nabla\tilde{\mathcal{R}}_i(\tilde{w}_i) + \frac{1}{n}\partial_r\partial_u\ell_j(\check{r}_{i,i},\check{u}_{i,i})(\tilde{u}_{i,i}-u_i)x_i + \frac{1}{n}\sum_{j\neq i}\partial_r\partial_u\ell_j(\check{r}_{j,i},\check{u}_{j,i})(\tilde{u}_{j,i}-u_j)x_j\right]
$$

The first term can be simply controlled as

$$
\sup_{i\in[n]}\left\|\check{H}^{-1}\nabla\tilde{\mathcal{R}}_i(\tilde{w}_i)\right\| \leq \frac{1}{\lambda}\sup_{i\in[n]}\left\|\nabla\tilde{\mathcal{R}}_i(\tilde{w}_i)\right\|.
$$

The second term can be upper-bounded as

$$
\sup_{i\in[n]}\left\|\frac{1}{n}\partial_r\partial_u\ell_j(\check{r}_{i,i},\check{u}_{i,i})(\tilde{u}_{i,i}-u_i)\check{H}^{-1}x_j\right\| \leq \frac{1}{\sqrt{n}\lambda}\|\partial_r\partial_u\ell\|_\infty\sup_{i\in[n]}\frac{\|x_i\|}{\sqrt{n}}O_{L_k}\left(\frac{\mathrm{polylog}(n)}{\sqrt{n}}\right),
$$

using Proposition A.5. Using Lemma A.40, $\sup_{i\in[n]}\|x_i\|/\sqrt{n} = O_{L_k}(\mathrm{polylog}(n))$. Thus,

$$
\sup_{i\in[n]}\left\|\frac{1}{n}\partial_r\partial_u\ell_j(\check{r}_{i,i},\check{u}_{i,i})(\tilde{u}_{i,i}-u_i)\check{H}^{-1}x_j\right\| = \frac{1}{\lambda}\|\partial_r\partial_u\ell\|_\infty O_{L_k}\left(\frac{\mathrm{polylog}(n)}{n}\right).
$$

Finally, defining $h_i\in\mathbb{R}^{(n-1)\times d}$ with entries $h_{i,j} = \partial_r\partial_u\ell_j(\check{r}_{j,i},\check{u}_{j,i})(\tilde{u}_{j,i}-u_j)$, the third term can be written as

$$
\sup_{i\in[n]}\left\|\frac{1}{n}\sum_{j\neq i}\partial_r\partial_u\ell_j(\check{r}_{j,i},\check{u}_{j,i})(\tilde{u}_{j,i}-u_j)\check{H}^{-1}x_j\right\| = \frac{1}{n}\left\|\check{H}^{-1}X_{\backslash i}^\top h_i\right\| \leq \frac{1}{\sqrt{n}\lambda}\sup_{i\in[n]}\frac{\|X_{\backslash i}\|}{\sqrt{n}}\sup_{i\in[n]}\|h_i\|.
$$

From Lemma A.40, $\sup_{i\in[n]}\|X_{\backslash i}\|/\sqrt{n} = O_{L_k}(\mathrm{polylog}(n))$. Furthermore,

$$
\sup_{i\in[n]}\|h_i\| \leq \|\partial_r\partial_u\ell\|_\infty\sqrt{\sup_{i\in[n]}\sum_{j\neq i}(\tilde{u}_{j,i}-u_j)^2} = \|\partial_r\partial_u\ell\|_\infty O_{L_k}\left(\frac{\mathrm{polylog}(n)}{\sqrt{n}}\right).
$$

Thus,

$$
\left\|\frac{1}{n}\sum_{j\neq i}\partial_r\partial_u\ell_j(\check{r}_{j,i},\check{u}_{j,i})(\tilde{u}_{j,i}-u_j)\check{H}^{-1}x_j\right\| = \frac{1}{\lambda}\|\partial_r\partial_u\ell\|_\infty O_{L_k}\left(\frac{\mathrm{polylog}(n)}{n}\right),
$$

which concludes the proof. $\square$

**Lemma A.9.** *We have* $\nabla\tilde{\mathcal{R}}(\tilde{w}_i) = \Delta_i^{(1)} + \Delta_i^{(2)}$ *where*

$$
\Delta_i^{(1)} = \frac{1}{n}\sum_{j\neq i}\left[\partial_r^2\ell_j(\check{r}_{j,i},\check{u}_{j,i}) - \partial_r^2\ell_j(r_{j,\backslash i},u_{j,\backslash i})\right]x_jx_j^\top(\eta_i+\zeta_i),
$$

$$
\Delta_i^{(2)} = \frac{1}{n}\sum_{j\neq i}\left[\partial_r\partial_u\ell_j(\check{r}_{j,i},\check{u}_{j,i}) - \partial_r\partial_u\ell_j(r_{j,\backslash i},u_{j,\backslash i})\right](\tilde{u}_{j,i}-u_{j,\backslash i})x_j,
$$

*where* $\check{r}_{j,i}$ *(resp.* $\check{u}_{j,i}$*) belongs to the unordered interval* $(\tilde{r}_{j,i},r_{j,\backslash i})$ *(resp.* $(\tilde{u}_{j,i},u_{j,\backslash i})$*).*

*Proof.* Starting from the identity $\nabla \hat{\mathcal{R}}_{\backslash i}(\hat{w}_{\backslash i})$, one reaches

$$
\begin{aligned}
\nabla \hat{\mathcal{R}}(\tilde{w}_i) &= \nabla \hat{\mathcal{R}}(\tilde{w}_i) - \nabla \hat{\mathcal{R}}_{\backslash i}(\hat{w}_{\backslash i}) \\
&= \frac{1}{n} \partial_r \ell_i(\tilde{r}_{i,i}, \tilde{u}_{i,i}) x_i + \frac{1}{n} \sum_{j \neq i} \left[ \partial_r \ell_j(\tilde{r}_{j,i}, \tilde{u}_{j,i}) - \partial_r \ell_j(r_{j,\backslash i}, u_{j,\backslash i}) \right] x_j + \lambda(\eta_i + \zeta_i)
\end{aligned}
$$

From a Taylor expansion, the second term reads

$$
\begin{aligned}
&\frac{1}{n} \sum_{j \neq i} \left[ \partial_r \ell_j(\tilde{r}_{j,i}, \tilde{u}_{j,i}) - \partial_r \ell_j(r_{j,\backslash i}, u_{j,\backslash i}) \right] x_j \\
&= \frac{1}{n} \sum_{j \neq i} \left[ \partial_r^2 \ell_j(\check{r}_{j,i}, \check{u}_{j,i}) \langle x_j, \eta_i + \zeta_i \rangle + \partial_r \partial_u \ell_j(\check{r}_{j,i}, \check{u}_{j,i})(\tilde{u}_{j,i} - u_{j,\backslash i}) \right] x_j \\
&= \frac{1}{n} \sum_{j \neq i} \left[ \partial_r^2 \ell_j(r_{j,\backslash i}, u_{j,\backslash i}) \langle x_j, \eta_i + \zeta_i \rangle + \partial_r \partial_u \ell_j(r_{j,\backslash i}, u_{j,\backslash i})(\tilde{u}_{j,i} - u_{j,\backslash i}) \right] x_j + \Delta_i^{(1)} + \Delta_i^{(2)},
\end{aligned}
$$

with $(\check{r}_{j,i}, \check{u}_{j,i})$ lying on the segment connecting $(\tilde{r}_{j,i}, \tilde{u}_{j,i})$ and $(r_{j,\backslash i}, u_{j,\backslash i})$. Thus

$$
\begin{aligned}
\nabla \hat{\mathcal{R}}(\tilde{w}_i) &= \frac{1}{n} \partial_r \ell_i(\tilde{r}_{i,i}, \tilde{u}_{i,i}) x_i + \Delta_i^{(1)} + \Delta_i^{(2)} + H_{\backslash i}(\eta_i + \zeta_i) - \frac{1}{n} \partial_u \ell_{0,i}(\tilde{u}_{i,i}) G_{\backslash i} H_{0,\backslash i}^{-1} x_i \\
&= \Delta_i^{(1)} + \Delta_i^{(2)},
\end{aligned}
$$

using the definitions of $\eta_i, \zeta_i$. $\qquad\square$

**Control of** $\sup_{i \in [n]} \left\| \Delta_i^{(1)} \right\|$ **—** We first aim to bound the first correction $\sup_{i \in [n]} \left\| \Delta_i^{(1)} \right\|$. Introducing the diagonal matrix $\Lambda \in \mathbb{R}^{(n-1) \times (n-1)}$ with elements $\Lambda_{jj} = \partial_r^2 \ell_j(\check{r}_{j,i}, \check{u}_{j,i}) - \partial_r^2 \ell_j(r_{j,\backslash i}, u_{j,\backslash i})$, one has

$$
\begin{aligned}
&\sup_{i \in [n]} \left\| \Delta_i^{(1)} \right\| \\
&\leq \sup_{i \in [n]} \frac{1}{n} \left\| X_{\backslash i}^\top \Lambda X_{\backslash i} \right\| (\|\eta_i\| + \|\zeta_i\|) \\
&\leq \sup_{i \in [n]} \frac{\|X_{\backslash i}\|^2}{n} \sup_{i \in [n]} \sup_{j \neq i} \left| \partial_r^2 \ell_j(\check{r}_{j,i}, \check{u}_{j,i}) - \partial_r^2 \ell_j(r_{j,\backslash i}, u_{j,\backslash i}) \right| \left( \sup_{i \in [n]} \|\eta_i\| + \sup_{i \in [n]} \|\zeta_i\| \right) \\
&\leq \left( \left\| \partial_r^3 \ell \right\|_\infty \vee \left\| \partial_r^2 \partial_u \ell \right\|_\infty \right) \sup_{i \in [n]} \frac{\|X_{\backslash i}\|^2}{n} \left( \sup_{i \in [n]} \sup_{j \neq i} |r_{j,\backslash i} - \tilde{r}_{j,i}| + \sup_{i \in [n]} \sup_{j \neq i} |u_{j,\backslash i} - \tilde{u}_{j,i}| \right) \left( \sup_{i \in [n]} \|\eta_i\| + \sup_{i \in [n]} \|\zeta_i\| \right) \quad (9)
\end{aligned}
$$

Lemma A.40 provides control of the first term as $\sup_{i \in [n]} \|X_{\backslash i}\|^2 / n = O_{L_k}(\mathrm{polylog}(n))$. The control of the last follows from the following lemma.

**Lemma A.10.** *We have*

$$
\sup_{i \in [n]} \|\eta_i\| \leq \frac{\|\partial_r \ell\|_\infty}{\lambda} O_{L_k}\left( \frac{\mathrm{polylog}(n)}{\sqrt{n}} \right),
$$

*and*

$$
\sup_{i \in [n]} \|\zeta_i\| \leq \frac{\|\partial_u \ell_0\|_\infty \|\partial_r \partial_u \ell\|_\infty}{\lambda_0 \lambda} O_{L_k}\left( \frac{\mathrm{polylog}(n)}{\sqrt{n}} \right).
$$

*Proof.* One has

$$
\sup_{i \in [n]} \|\eta_i\| \leq \frac{1}{\sqrt{n}\lambda} \|\partial_r \ell\|_\infty \sup_{i \in [n]} \frac{\|x_i\|}{\sqrt{n}} = \frac{\|\partial_r \ell\|_\infty}{\lambda} O_{L_k}\left( \frac{\mathrm{polylog}(n)}{\sqrt{n}} \right),
$$

using Lemma A.40. By the same token, $\|\zeta_i\|$ can be controlled as

$$\sup_{i\in[n]} \|\zeta_i\| \leq \frac{1}{\sqrt{n}} \|\partial_u \ell_0\|_\infty \frac{1}{\lambda_0 \lambda} \sup_{i\in[n]} \|G_{\setminus i}\| \sup_{i\in[n]} \frac{\|x_i\|}{\sqrt{n}}.$$

But

$$\sup_{i\in[n]} \|G_{\setminus i}\| = \sup_{i\in[n]} \left\| \frac{1}{n} \sum_{j\neq i} \partial_r \partial_u \ell(r_{j,\setminus i}, u_{j,\setminus i}) x_j x_j^\top \right\|$$

$$\leq \|\partial_r \partial_u \ell\|_\infty \sup_{i\in[n]} \frac{\|X_{\setminus i}\|^2}{n} = \|\partial_r \partial_u \ell\|_\infty O_{L_k}(\text{polylog}(n)),$$

using again Lemma A.40. $\qquad \square$

Finally, one needs control of the middle term in (9). The control of $\sup_{i\in[n]} \sup_{j\neq i} |u_{j,\setminus i} - \tilde{u}_{j,i}|$ follows from Lemma C.4 of (El Karoui, 2018), or equivalently the following proof on a surrogate problem with zero loss, as remarked above.

**Lemma A.11.** *For any* $k \in [\lceil 2\log(n) \rceil]$,

$$\sup_{i\in[n]} \sup_{j\neq i} |r_{j,\setminus i} - \tilde{r}_{j,i}| = O_{L_k}\left( \frac{\text{polylog}(n)}{\sqrt{n}} \right).$$

*Proof.* Let us first note that

$$\sup_{i\in[n]} \sup_{j\neq i} |r_{j,\setminus i} - \tilde{r}_{j,i}| \leq \sup_{i\in[n]} \sup_{j\neq i} |\langle x_j, \eta_i \rangle| + \sup_{i\in[n]} \sup_{j\neq i} |\langle x_j, \zeta_i \rangle|$$

$$\leq \|\partial_r \ell\|_\infty \sup_{i\in[n]} \sup_{j\neq i} \left| \frac{1}{n} \left\langle x_j, H_{\setminus i}^{-1} x_i \right\rangle \right| + \|\partial_u \ell_0\|_\infty \sup_{i\in[n]} \sup_{j\neq i} \left| \frac{1}{n} \left\langle x_j, H_{\setminus i}^{-1} G_{\setminus i} H_{0,\setminus i}^{-1} x_i \right\rangle \right| \quad (10)$$

It follows from Lemma A.41 that

$$\sup_{i\in[n]} \sup_{j\neq i} \left| \frac{1}{n} \left\langle x_j, H_{\setminus i}^{-1} x_i \right\rangle \right| = \frac{1}{\lambda} O_{L_k}\left( \frac{\text{polylog}(n)}{\sqrt{n}} \right)$$

The second term of (10) admits the same upper bound, since from Lemma A.40 and the proof of Lemma A.10, on again has $\left\| H_{\setminus i}^{-1} G_{\setminus i} H_{0,\setminus i}^{-1} \right\| = O_{L_k}(\text{polylog}(n))$. Hence returning to (10),

$$\sup_{i\in[n]} \sup_{j\neq i} |r_{j,\setminus i} - \tilde{r}_{j,i}| = O_{L_k}\left( \frac{\text{polylog}(n)}{\sqrt{n}} \right).$$

This concludes the proof. $\qquad \square$

Returning to (9) and assembling these intermediary results finally allows to control $\sup_{i\in[n]} \left\| \Delta_i^{(1)} \right\|$. This is summarized in the following lemma.

**Lemma A.12.** *For any* $k \in [\lceil 1/2 \log(n) \rceil]$,

$$\sup_{i\in[n]} \left\| \Delta_i^{(1)} \right\| = O_{L_k}\left( \frac{\text{polylog}(n)}{n^2} \right).$$

**Control of** $\sup_{i\in[n]} \left\| \Delta_i^{(2)} \right\|$ — We now turn to the second correction $\sup_{i\in[n]} \left\| \Delta_i^{(2)} \right\|$. It can similarly be bounded along the same steps as $\sup_{i\in[n]} \left\| \Delta_i^{(1)} \right\|$.

**Lemma A.13.** *For any $k \in [\lceil 1/2 \log(n) \rceil]$,*

$$\sup_{i \in [n]} \left\| \Delta_i^{(2)} \right\| = O_{L_k} \left( \frac{\mathrm{polylog}(n)}{n} \right).$$

*Proof.* Introducing the diagonal matrix $\Lambda \in \mathbb{R}^{(n-1) \times (n-1)}$ with elements $\Lambda_{jj} = \partial_r \partial_u \ell_j(\check{r}_{j,i}, \check{u}_{j,i}) - \partial_r \partial_u \ell_j(r_{j,\backslash i}, u_{j,\backslash i})$,

$$
\begin{aligned}
\sup_{i \in [n]} \left\| \Delta_i^{(2)} \right\| &\leq \sup_{i \in [n]} \frac{1}{n} \left\| X_{\backslash i}^\top \Lambda X_{\backslash i} \right\| \left\| \partial_u \ell_0 \right\|_\infty \left\| H_{0,\backslash i}^{-1} x_i \right\| \\
&\leq \sup_{i \in [n]} \frac{\left\| X_{\backslash i} \right\|^2}{n} \sup_{i \in [n]} \sup_{j \neq i} \left| \partial_r \partial_u \ell_j(\check{r}_{j,i}, \check{u}_{j,i}) - \partial_r \partial_u \ell_j(r_{j,\backslash i}, u_{j,\backslash i}) \right| \sup_{i \in [n]} \left\| \partial_u \ell_0 \right\|_\infty \frac{1}{n} \left\| H_{0,\backslash i}^{-1} x_i \right\| \\
&\leq \left( \left\| \partial_r^2 \partial_u \ell \right\|_\infty \vee \left\| \partial_r \partial_u^2 \ell \right\|_\infty \right) \sup_{i \in [n]} \frac{\left\| X_{\backslash i} \right\|^2}{n} \left( \sup_{i \in [n]} \sup_{j \neq i} \left| r_{j,\backslash i} - \check{r}_{j,i} \right| + \sup_{i \in [n]} \sup_{j \neq i} \left| u_{j,\backslash i} - \check{u}_{j,i} \right| \right) \\
&\quad \left\| \partial_u \ell_0 \right\|_\infty \sup_{i \in [n]} \frac{1}{n} \left\| H_{0,\backslash i}^{-1} x_i \right\|
\end{aligned}
$$

From Lemma A.40,

$$\sup_{i \in [n]} \frac{1}{n} \left\| H_{0,\backslash i}^{-1} x_i \right\| = O_{L_k} \left( \frac{\mathrm{polylog}(n)}{\sqrt{n}} \right), \qquad \sup_{i \in [n]} \frac{\left\| X_{\backslash i} \right\|^2}{n} = O_{L_k} \left( \mathrm{polylog}(n) \sqrt{n} \right)$$

From the proof of Lemma A.12, for any $k \in [\lceil 2 \log(n) \rceil]$

$$\left( \sup_{i \in [n]} \sup_{j \neq i} \left| r_{j,\backslash i} - \check{r}_{j,i} \right| + \sup_{i \in [n]} \sup_{j \neq i} \left| u_{j,\backslash i} - \check{u}_{j,i} \right| \right) = O_{L_k} \left( \frac{\mathrm{polylog}(n)}{\sqrt{n}} \right).$$

The result follows from combining these bounds. $\qquad \square$

One is finally in a position to write the equivalent of Proposition A.5 for the residuals of the reweighted problem.

**Proposition A.14** (Approximations of the reweighted estimator). *For any $k \in \mathbb{N}$,*

$$\sup_{i \in [n]} \left\| \hat{w} - \tilde{w}_i \right\| = O_{L_k} \left( \frac{\mathrm{polylog}(n)}{n} \right).$$

*Furthermore, at the level of the residuals,*

$$\sup_{i \in [n]} \sup_{j \neq i} \left| r_{j,\backslash i} - r_j \right| = O_{L_k} \left( \frac{\mathrm{polylog}(n)}{\sqrt{n}} \right)$$

*and*

$$\sup_{i \in [n]} \left| r_i - \tilde{r}_{i,i} \right| = O_{L_k} \left( \frac{\mathrm{polylog}(n)}{\sqrt{n}} \right).$$

*Proof.* The first point follows from combining Lemma A.6, Lemma A.9, Lemma A.12 and Lemma A.13. The second point follows from Lemma A.40 and the Cauchy-Schwartz inequality as

$$
\begin{aligned}
\sup_{i \in [n]} \sup_{j \neq i} \left| r_{j,\backslash i} - r_j \right| &\leq \sup_{j \in [n]} \left\| x_j \right\| \sup_{i \in [n]} \left\| \hat{w} - \tilde{w}_i \right\| + \sup_{i \in [n]} \sup_{j \neq i} \left| \langle x_j, \eta_i + \zeta_i \rangle \right| \\
&= O_{L_k} \left( \frac{\mathrm{polylog}(n)}{\sqrt{n}} \right).
\end{aligned}
$$

We remind that the term $\sup_{\substack{i \in [n]}} \sup_{j \neq i} |\langle x_j, \eta_i + \zeta_i \rangle|$ was bounded in Lemma A.11. The last point follows from Lemma A.40 and the Cauchy-Schwartz inequality:

$$\sup_{i \in [n]} |r_i - \tilde{r}_{i,i}| \leq \sup_{i \in [n]} \|x_i\| \sup_{i \in [n]} \|\hat{w} - \tilde{w}_i\| = O_{L_k}\left(\frac{\text{polylog}(n)}{\sqrt{n}}\right)$$

$\square$

*Remark* A.15. One also has

$$\sup_{i \in [n]} \|\tilde{w}_i - \hat{w}_{\setminus i}\| = O_{L_k}\left(\frac{\text{polylog}(n)}{\sqrt{n}}\right).$$

and

$$\sup_{i \in [n]} \|\hat{w} - \hat{w}_{\setminus i}\| = O_{L_k}\left(\frac{\text{polylog}(n)}{\sqrt{n}}\right).$$

*Proof.* The first point follows from $\|\tilde{w}_i - \hat{w}_{\setminus i}\| \leq \|\eta_i\| + \|\zeta_i\|$ and Lemma A.10. The second point follows from the first and Proposition A.14 with the triangle inequality. $\square$

As for the base case, the following Lemma follows from Proposition A.14 as a corollary.

**Lemma A.16.** *We have*

$$\sup_{i \in [n]} \sum_{j \neq i} (r_j - \tilde{r}_{j,i})^2 = O_{L_k}\left(\frac{\text{polylog}(n)}{n}\right).$$

*Proof.* The proof is identical to that of Lemma A.6. $\square$

### A.3. Concentration results

We now show the concentration of the summary statistics $\mathfrak{m}_c = \langle \hat{w}, \mu_c \rangle$, $\mathfrak{h} = \langle \hat{w}, \beta \rangle$, $\mathfrak{t} = \langle \hat{w}, \hat{w}_0 \rangle$, $\mathfrak{q} = \|\hat{w}\|^2$ and $\gamma_i = 1/n x_i^\top H_{\setminus i}^{-1} x_i$, $\mathfrak{V} = 1/n \operatorname{tr}\left[H_{\setminus i}^{-1}\right]$, $v_i = 1/n H_{\setminus i}^{-1} G_{\setminus i} H_{0, \setminus i}^{-1} x_i$, $\mathfrak{X} = 1/n \operatorname{tr}\left[H^{-1} G H_0^{-1}\right]$. We reserve the gothic letters to denote random variables, and will later adopt normal font for their respective deterministic equivalents in $L_2$.

**Lemma A.17** ($\mathfrak{m}_c$ concentrates). *For any $c \in \mathcal{V}$, $\mathfrak{m}_c = \langle \hat{w}, \mu_c \rangle$ concentrates in $L_2$, namely*

$$\operatorname{Var}[\mathfrak{m}_c] = O\left(\frac{\text{polylog}(n)}{n}\right).$$

*Proof.* From the Efron-Stein lemma (Efron & Stein, 1981),

$$\operatorname{Var}[\mathfrak{m}_c] \leq \sum_{i \in [n]} \mathbb{E}\left[\left(\langle \hat{w} - \hat{w}_{\setminus i}, \mu_c \rangle\right)^2\right]$$

$$\leq 2 \sum_{i \in [n]} \mathbb{E}\left[\left(\langle \hat{w} - \tilde{w}_i, \mu_c \rangle\right)^2\right] + 2 \sum_{i \in [n]} \mathbb{E}\left[\left(\langle \tilde{w}_i - \hat{w}_{\setminus i}, \mu_c \rangle\right)^2\right]$$

The first term can be controlled using the Cauchy-Schwartz inequality and Proposition A.14.

$$\sum_{i \in [n]} \mathbb{E}\left[\left(\langle \hat{w} - \tilde{w}_i, \mu_c \rangle\right)^2\right] \leq \sum_{i \in [n]} \mathbb{E}\left[M \|\hat{w} - \tilde{w}_i\|^2\right] = O\left(\frac{\text{polylog}(n)}{n}\right).$$

Now recalling that

$$\left(\langle \tilde{w}_i - \hat{w}_{\setminus i}, \mu_c \rangle\right)^2 \leq 2 \langle \eta_i, \mu_c \rangle^2 + 2 \langle \zeta_i, \mu_c \rangle^2,$$

we focus on bounding $\langle \eta_i, \mu_c \rangle$, $\langle \zeta_i, \mu_c \rangle$. Since

$$\langle \eta_i, \mu_c \rangle^2 \leq \|\partial_r \ell\|_\infty^2 \frac{1}{n^2} \mu_c^\top H_{\backslash i}^{-1} x_i x_i^\top H_{\backslash i}^{-1} \mu_c,$$

one has

$$\mathbb{E}\left[ \langle \eta_i, \mu_c \rangle^2 \right] \leq \|\partial_r \ell\|_\infty^2 \frac{1}{n^2} \mathbb{E}\left[ \left\| H_{\backslash i}^{-1} \mu_c \right\|^2 \right] \leq \|\partial_r \ell\|_\infty^2 \frac{M}{\lambda^2 n^2}.$$

By the same token,

$$\mathbb{E}\left[ \langle \zeta_i, \mu_c \rangle^2 \right] \leq \|\partial_u \ell_0\|_\infty^2 \frac{1}{n^2} \mathbb{E}\left[ \left\| H_{0\backslash i}^{-1} G_{\backslash i} H_{\backslash i}^{-1} \mu_c \right\|^2 \right] \leq \frac{M \|\partial_u \ell_0\|_\infty^2}{\lambda^2 \lambda_0^2 n_2} \mathbb{E}\left[ \|G_{\backslash i}\|^2 \right] = O\left( \frac{\mathrm{polylog}(n)}{n^2} \right),$$

since we showed in Lemma A.10 that $\left\| G_{\backslash i} \right\| = O_{L_k}(\mathrm{polylog}(n))$. Thus,

$$\sum_{i \in [n]} \mathbb{E}\left[ \langle \tilde{w}_i - \hat{w}_{\backslash i}, \mu_c \rangle^2 \right] = O\left( \frac{\mathrm{polylog}(n)}{n} \right),$$

completing the proof. $\qquad\square$

The following concentration result for $\mathfrak{h}$ can be established using an identical proof, substituting $\mu_c$ for $\beta$.

**Lemma A.18** ($\mathfrak{h}$ concentrates)**.** $\mathfrak{h} = \langle \hat{w}, \beta \rangle$ *concentrates in* $L_2$*, namely*

$$\mathrm{Var}\left[ \mathfrak{h} \right] = O\left( \frac{\mathrm{polylog}(n)}{n} \right).$$

Establishing a similar result for $\mathfrak{t}$ requires a more careful analysis, since $\hat{w}_0$ is a stochastic vector that depends on the samples.

**Lemma A.19** ($\mathfrak{t}$ concentrates)**.** $\mathfrak{t} = \langle \hat{w}, \hat{w}_0 \rangle$ *concentrates in* $L_2$*, namely*

$$\mathrm{Var}\left[ \mathfrak{t} \right] = O\left( \frac{\mathrm{polylog}(n)}{n} \right).$$

*Proof.* From the Efron-Stein lemma,

$$\mathrm{Var}\left[ \mathfrak{t} \right] \leq \sum_{i \in [n]} \mathbb{E}\left[ \left( \langle \hat{w}, \hat{w}_0 \rangle - \langle \hat{w}_{\backslash i}, \hat{w}_{0, \backslash i} \rangle \right)^2 \right]$$

$$\leq 5 \sum_{i \in [n]} \mathbb{E}\left[ \langle \hat{w} - \tilde{w}_i, \hat{w}_{0,\backslash i} \rangle^2 + \langle \hat{w}_{\backslash i} - \tilde{w}_i, \hat{w}_{0,\backslash i} \rangle^2 + \langle \hat{w}_{\backslash i}, \hat{w}_0 - \tilde{w}_{0,i} \rangle^2 + \langle \hat{w}_{\backslash i}, \hat{w}_{0,\backslash i} - \tilde{w}_{0,i} \rangle^2 + \langle \hat{w}_{0,\backslash i} - \hat{w}_0, \hat{w}_{\backslash i} - \hat{w} \rangle^2 \right]$$

From the Cauchy-Schwartz inequality, Proposition A.14 and Lemma A.37,

$$\sum_{i \in [n]} \mathbb{E}\left[ \left( \langle \hat{w} - \tilde{w}_i, \hat{w}_{0,\backslash i} \rangle \right)^2 \right] \leq \sum_{i \in [n]} \mathbb{E}\left[ \|\hat{w}_{0,\backslash i}\|^2 \|\hat{w} - \tilde{w}_i\|^2 \right] = O\left( \frac{\mathrm{polylog}(n)}{n} \right).$$

From the Cauchy-Schwartz inequality, Proposition A.5 and Lemma A.38,

$$\sum_{i \in [n]} \mathbb{E}\left[ \left( \langle \hat{w}_0 - \tilde{w}_{0,i}, \hat{w}_{\backslash i} \rangle \right)^2 \right] \leq \sum_{i \in [n]} \mathbb{E}\left[ \|\hat{w}_{,\backslash i}\|^2 \|\hat{w}_0 - \tilde{w}_{0,i}\|^2 \right] = O\left( \frac{\mathrm{polylog}(n)}{n} \right).$$

The last term can simply be controlled from the Cauchy-Schwartz inequality, and Remark A.15:

$$\sum_{i \in [n]} \mathbb{E}\left[ \langle \hat{w}_{0,\backslash i} - \hat{w}_0, \hat{w}_{\backslash i} - \hat{w} \rangle^2 \right] \leq \sum_{i \in [n]} \mathbb{E}\left[ \|\hat{w}_{0,\backslash i} - \hat{w}_0\|^2 \|\hat{w}_{\backslash i} - \hat{w}\|^2 \right] = O\left( \frac{1}{n} \right)$$

Now recalling that

$$\left\langle \tilde{w}_i - \hat{w}_{\backslash i}, \hat{w}_{0, \backslash i} \right\rangle^2 \leq 2 \left\langle \eta_i, \hat{w}_{0, \backslash i} \right\rangle^2 + 2 \left\langle \zeta_i, \hat{w}_{0, \backslash i} \right\rangle^2,$$

we focus on bounding $\left\langle \eta_i, \hat{w}_{0, \backslash i} \right\rangle, \left\langle \zeta_i, \hat{w}_{0, \backslash i} \right\rangle$. Since

$$\left\langle \eta_i, \hat{w}_{0, \backslash i} \right\rangle^2 \leq \|\partial_r \ell\|_\infty^2 \frac{1}{n^2} \hat{w}_{0, \backslash i}^\top H_{\backslash i}^{-1} x_i x_i^\top H_{\backslash i}^{-1} \hat{w}_{0, \backslash i},$$

one has from Lemma A.37

$$\mathbb{E}\left[ \left\langle \eta_i, \hat{w}_{0, \backslash i} \right\rangle^2 \right] \leq \|\partial_r \ell\|_\infty^2 \frac{1}{n^2} \mathbb{E}\left[ \left\| H_{\backslash i}^{-1} \hat{w}_{0, \backslash i} \right\|^2 \right] \leq \|\partial_r \ell\|_\infty^2 \frac{1}{\lambda^2 n^2} O(\mathrm{polylog}(n)).$$

By the same token,

$$\mathbb{E}\left[ \left\langle \zeta_i, \hat{w}_{0, \backslash i} \right\rangle^2 \right] \leq \|\partial_u \ell_0\|_\infty^2 \frac{1}{n^2} \mathbb{E}\left[ \left\| H_{0 \backslash i}^{-1} G_{\backslash i} H_{\backslash i}^{-1} \hat{w}_{0, \backslash i} \right\|^2 \right] \leq \frac{\|\partial_u \ell_0\|_\infty^2}{\lambda^2 \lambda_0^2 n_2} \mathbb{E}\left[ \|G_{\backslash i}\|^2 \|\hat{w}_{0, \backslash i}\|^2 \right]$$

$$= O\left( \frac{\mathrm{polylog}(n)}{n^2} \right),$$

since we showed in Lemma A.10 that $\left\| G_{\backslash i} \right\| = O_{L_k}(\mathrm{polylog}(n))$. Thus,

$$\sum_{i \in [n]} \mathbb{E}\left[ \left\langle \tilde{w}_i - \hat{w}_{\backslash i}, \hat{w}_{0, \backslash i} \right\rangle^2 \right] = O\left( \frac{\mathrm{polylog}(n)}{n} \right).$$

Finally, let us analyze similarly the last term

$$\left\langle \hat{w}_{\backslash i}, \hat{w}_0 - \tilde{w}_{0,i} \right\rangle^2 = \left\langle \hat{w}_{\backslash i}, \frac{1}{n} \partial_u \ell_0(\tilde{u}_{i,i}) H_{0, \backslash i}^{-1} x_i \right\rangle^2$$

Thus,

$$\mathbb{E}\left[ \left\langle \hat{w}_{\backslash i}, \hat{w}_0 - \tilde{w}_{0,i} \right\rangle^2 \right] \leq \|\partial_u \ell_0\|_\infty^2 \frac{1}{n^2} \mathbb{E}\left[ \left\| H_{0, \backslash i}^{-1} \hat{w}_{\backslash i} \right\|^2 \right] = O\left( \frac{\mathrm{polylog}(n)}{n^2} \right).$$

Therefore, finally,

$$\sum_{i \in [n]} \mathbb{E}\left[ \left\langle \hat{w}_{\backslash i}, \hat{w}_0 - \tilde{w}_{0,i} \right\rangle^2 \right] = O\left( \frac{\mathrm{polylog}(n)}{n} \right),$$

completing the proof. $\square$

**Lemma A.20** (q concentrates). $q = \|\hat{w}\|^2$ *concentrates in* $L_2$, *namely*

$$\mathrm{Var}\left[ q \right] = O\left( \frac{\mathrm{polylog}(n)}{n} \right).$$

*Proof.* Once more appealing to the Efron-Stein lemma,

$$\mathrm{Var}\left[ q \right] \leq \sum_{i \in [n]} \mathbb{E}\left[ \left( \|\hat{w}\|^2 - \|\hat{w}_{\backslash i}\|^2 \right)^2 \right]$$

$$\leq 2 \sum_{i \in [n]} \mathbb{E}\left[ \left( \|\hat{w}\|^2 - \|\tilde{w}_i\|^2 \right)^2 \right] + \mathbb{E}\left[ \left( \|\tilde{w}_i\|^2 - \|\hat{w}_{\backslash i}\|^2 \right)^2 \right].$$

But

$$\left(\|\hat{w}\|^2 - \|\tilde{w}_i\|^2\right)^2 \leq \|\hat{w} - \tilde{w}_i\|^2 \left(\|\hat{w}\| + \|\tilde{w}_i\|\right)^2 = O_{L_k}\left(\frac{\text{polylog}(n)}{n^2}\right),$$

using Proposition A.14 and Lemma A.38. By the same token,

$$\|\tilde{w}_i\|^2 - \|\hat{w}_{\backslash i}\|^2 \leq 2\|\eta_i\|^2 + 2\|\zeta_i\|^2 + 2\left\langle \hat{w}_{\backslash i}, \eta_i + \zeta_i \right\rangle = 2\left\langle \hat{w}_{\backslash i}, \eta_i + \zeta_i \right\rangle + O_{L_k}\left(\frac{\text{polylog}(n)}{n}\right),$$

using Lemma A.10. Furthermore, one has

$$\mathbb{E}\left[\left\langle \eta_i, \hat{w}_{\backslash i}\right\rangle^2\right] \leq \|\partial_r \ell\|_\infty^2 \frac{1}{n^2}\mathbb{E}\left[\left\|H_{\backslash i}^{-1}\hat{w}_{\backslash i}\right\|^2\right] \leq \|\partial_r \ell\|_\infty^2 \frac{1}{\lambda^2 n^2}O_{L_k}\left(\text{polylog}(n)\right)$$

and By the same token,

$$\mathbb{E}\left[\left\langle \zeta_i, \hat{w}_{\backslash i}\right\rangle^2\right] \leq \|\partial_u \ell_0\|_\infty^2 \frac{1}{n^2}\mathbb{E}\left[\left\|H_{0\backslash i}^{-1}G_{\backslash i}H_{\backslash i}^{-1}\hat{w}_{\backslash i}\right\|^2\right] \leq \frac{\|\partial_u \ell_0\|_\infty^2}{\lambda^2 \lambda_0^2 n_2}\mathbb{E}\left[\|G_{\backslash i}\|^2\|\hat{w}_{\backslash i}\|^2\right]$$
$$= O\left(\frac{\text{polylog}(n)}{n^2}\right),$$

Thus,

$$\mathbb{E}\left[\left(\|\tilde{w}_i\|^2 - \|\hat{w}_{\backslash i}\|^2\right)^2\right] = O\left(\frac{\text{polylog}(n)}{n^2}\right),$$

concluding the proof. $\square$

**Lemma A.21** ($\mathfrak{V}$ concentrates). *We remind $\gamma_i = {}^1\!/n\, x_i^\top H_{\backslash i}^{-1} x_i$, $\mathfrak{V} = \frac{1}{n}\operatorname{tr}[H^{-1}]$. $\gamma_i$ admits an equivalent*

$$\sup_{i \in [n]} |\gamma_i - \mathfrak{V}| = O_{L_k}\left(\frac{\text{polylog}(n)}{\lambda\sqrt{n}}\right).$$

*Furthermore, $\mathfrak{V}$ concentrates, with*

$$\operatorname{Var}[\mathfrak{V}] = O\left(\frac{\text{polylog}(n)}{n}\right).$$

*Proof.* We begin by proving the first point. First note that the means bring negligible contributions to $\gamma_i$:

$$\gamma_i = \frac{\mu_{c_i}^\top H_{\backslash i}^{-1}\mu_{c_i}}{n} + \frac{z_i^\top H_{\backslash i}^{-1}\mu_{c_i}}{n} + \frac{z_i^\top H_{\backslash i}^{-1}z_i}{n} = \frac{z_i^\top H_{\backslash i}^{-1}z_i}{n} + O_{L_k}\left(\frac{\text{polylog}(n)}{\sqrt{n}}\right),$$

using Cauchy-Schwartz bounds and the bound on $\sup_{i\in[n]}\|x_i\|/\sqrt{n}$ of Lemma A.40. From Lemma G.2 of (El Karoui, 2018), defining $\mathfrak{V}_{\backslash i} = \frac{1}{n}\operatorname{tr}\left[H_{\backslash i}^{-1}\right]$, one has the bound

$$\sup_{i \in [n]} \left|\frac{z_i^\top H_{\backslash i}^{-1}z_i}{n} - \mathfrak{V}_{\backslash i}\right| = O_{L_k}\left(\frac{\text{polylog}(n)}{\lambda\sqrt{n}}\right).$$

It now remains to control $|\mathfrak{V}_{\backslash i} - \mathfrak{V}|$. We start from the identity

$$\mathfrak{V} - \mathfrak{V}_{\backslash i} = \frac{1}{n}\operatorname{tr}\left[H^{-1}\left(H_{\backslash i}^{-1} - H^{-1}\right)H_{\backslash i}^{-1}\right]$$
$$= -\frac{1}{n}\partial_r^2\ell_i(r_i, u_i)\frac{x_i^\top H_{\backslash i}^{-1}H^{-1}x_i}{n} - \frac{1}{n}\sum_{j\neq i}\left(\partial_r^2\ell_j(r_j, u_j) - \partial_r^2\ell_j(r_{j,\backslash i}, u_{j,\backslash i})\right)\frac{x_j^\top H_{\backslash i}^{-1}H^{-1}x_j}{n}.$$

The first term can be simply controlled as

$$\left| \frac{1}{n} \partial_r^2 \ell_i(r_i, u_i) \frac{x_i^\top H_{\backslash i}^{-1} H^{-1} x_i}{n} \right| \leq \left\| \partial_r^2 \ell \right\|_\infty \frac{1}{\lambda^2} O_{L_k} \left( \frac{\text{polylog}(n)}{n} \right),$$

using Lemma A.40. The second term can be bounded as

$$\left| \frac{1}{n} \sum_{j \neq i} \left( \partial_r^2 \ell_j(r_j, u_j) - \partial_r^2 \ell_j(r_{j,\backslash i}, u_{j,\backslash i}) \right) \frac{x_j^\top H_{\backslash i}^{-1} H^{-1} x_j}{n} \right|$$

$$\leq \left( \left\| \partial_r^3 \ell \right\|_\infty \vee \left\| \partial_r^2 \partial_u \ell \right\|_\infty \right) \left( \sup_{i \in [n]} \sup_{j \neq i} |r_{j,\backslash i} - r_j| + \sup_{i \in [n]} \sup_{j \neq i} |u_{j,\backslash i} - u_j| \right) \frac{1}{\lambda^2} \sup_{j \in [n]} \frac{\|x_j\|^2}{n} = O_{L_k} \left( \frac{\text{polylog}(n)}{\sqrt{n}} \right),$$

using in the last step Proposition A.5, Proposition A.14 and Lemma A.40. This concludes the proof of the first point. Establishing the second point requires to reach a finer control of $|\mathfrak{V}_{\backslash i} - \mathfrak{V}|$. The reasoning follows closely that of Lemma 11 in (Barnfield et al., 2025). Starting again from the Efron-Stein lemma,

$$\text{Var}\left[ \mathfrak{V} \right] \leq \sum_{i \in [n]} \mathbb{E}\left[ (\mathfrak{V} - \mathfrak{V}_{\backslash i})^2 \right] \leq 2 \sum_{i \in [n]} \left( \mathbb{E}\left[ (\mathfrak{V} - \tilde{\mathfrak{V}}_i)^2 \right] + \mathbb{E}\left[ (\tilde{\mathfrak{V}}_i - \mathfrak{V}_{\backslash i})^2 \right] \right),$$

we introduce $\tilde{\mathfrak{V}}_i = 1/n \, \text{tr}\left[ \tilde{H}_i \right]$ as the normalized trace of the surrogate Hessian. We first focus on $\mathbb{E}\left[ (\mathfrak{V} - \tilde{\mathfrak{V}}_i)^2 \right]$.

$$\mathfrak{V} - \tilde{\mathfrak{V}}_i = \frac{1}{n} \text{tr}\left[ H^{-1} \left( \tilde{H}_i^{-1} - H^{-1} \right) \tilde{H}_i^{-1} \right]$$

$$= \frac{1}{n} \sum_{j \neq i} \left( \partial_r^2 \ell_j(r_j, u_j) - \partial_r^2 \ell_j(\tilde{r}_{j,i}, \tilde{u}_{j,i}) \right) \frac{x_j^\top \tilde{H}_i^{-1} H^{-1} x_j}{n} + \partial_r^2 \ell(r_j, u_j) \frac{x_i^\top \tilde{H}_i^{-1} H^{-1} x_i}{n^2}$$

$$= \frac{1}{n} \sum_{j \in [n]} \left( \partial_r^3 \ell_j(\check{r}_{j,i}, \check{u}_{j,i})(r_j - \tilde{r}_{j,i}) + \partial_r^2 \partial_u \ell_j(\check{r}_{j,i}, \check{u}_{j,i})(u_j - \tilde{u}_{j,i}) \right) \frac{x_j^\top \tilde{H}_i^{-1} H^{-1} x_j}{n} + \partial_r^2 \ell(r_j, u_j) \frac{x_i^\top \tilde{H}_i^{-1} H^{-1} x_i}{n^2}$$

for some $\check{r}_{j,i}, \check{u}_{j,i}$ lying on the segment connecting $\tilde{r}_{j,i}, \tilde{u}_{j,i}$ and $r_j, u_j$, from the mean value theorem. The last term is $O_{L_k}(\text{polylog}(n)/n)$. Let us introduce the vectors $h^r \in \mathbb{R}^n, h^u \in \mathbb{R}^n$, with entries $h_j^r = \partial_r^3 \ell_j(\check{r}_{j,i}, \check{u}_{j,i}) x_j^\top \tilde{H}_i^{-1} H^{-1} x_j / n$ and $h_j^u = \partial_r^2 \partial_u \ell_j(\check{r}_{j,i}, \check{u}_{j,i}) x_j^\top \tilde{H}_i^{-1} H^{-1} x_j / n$. Similarly, let $\delta_r, \delta_u \in \mathbb{R}^n$ with entries $\delta_{r,j} = (r_j - \tilde{r}_{j,i})$ and $\delta_{u,j} = (u_j - \tilde{u}_{j,i})$. Then $\mathfrak{V} - \tilde{\mathfrak{V}}_i$ can be more compactly rewritten as

$$|\mathfrak{V} - \tilde{\mathfrak{V}}_i| \leq \frac{1}{n} |\langle h^r, \delta_r \rangle| + \frac{1}{n} |\langle h^u, \delta_u \rangle| + O_{L_k}(\frac{\text{polylog}(n)}{n}) \leq \frac{1}{n} \left( \|h^r\| \|\delta^r\| + \|h^u\| \|\delta^u\| \right) + O_{L_k}(\frac{\text{polylog}(n)}{n}).$$

From Lemma A.6 and Lemma A.16, $\|\delta^r\|, \|\delta^u\| = O_{L_k}(\text{polylog}(n)/\sqrt{n})$. In parallel,

$$\|h^r\| \leq \sqrt{n} \frac{\left\| \partial_r^3 \ell \right\|_\infty}{\lambda^2} \sup_{j \in [n]} \frac{\|x_j\|}{\sqrt{n}} = O_{L_k}\left( \sqrt{n} \text{polylog}(n) \right),$$

$$\|h^u\| \leq \sqrt{n} \frac{\left\| \partial_r^2 \partial_u \ell \right\|_\infty}{\lambda^2} \sup_{j \in [n]} \frac{\|x_j\|}{\sqrt{n}} = O_{L_k}\left( \sqrt{n} \text{polylog}(n) \right).$$

Thus $|\mathfrak{V} - \tilde{\mathfrak{V}}_i| = O_{L_k}(\text{polylog}(n)/n)$ and

$$2 \sum_{i \in [n]} \mathbb{E}\left[ (\mathfrak{V} - \tilde{\mathfrak{V}}_i)^2 \right] = O_{L_k}\left( \frac{\text{polylog}(n)}{n} \right).$$

We now turn to the term $\mathbb{E}\left[(\tilde{\mathfrak{V}}_i - \mathfrak{V}_{\setminus i})^2\right]$. $\tilde{\mathfrak{V}}_i - \mathfrak{V}_{\setminus i}$ can be decomposed as

$$\tilde{\mathfrak{V}}_i - \mathfrak{V}_{\setminus i} = \frac{1}{n}\sum_{j\neq i}\left(\partial_r^3\ell_j(r_{j,\setminus i}, u_{j,\setminus i})(\tilde{r}_{j,i} - r_{j,\setminus i}) + \partial_r^2\partial_u\ell_j(r_{j,\setminus i}, u_{j,\setminus i})(\tilde{u}_{j,i} - u_{j,\setminus i})\right)\frac{x_j^\top \tilde{H}_i^{-1} H_{\setminus i}^{-1} x_j}{n}$$

$$+ \frac{1}{2n}\sum_{j\neq i}\left(\partial_r^4\ell_j(\check{r}_{j,i}, \check{u}_{j,i})^2(\tilde{r}_{j,i} - r_{j,\setminus i})^2 + \partial_r^2\partial_u^2\ell_j(\check{r}_{j,i}, \check{u}_{j,i})(\tilde{u}_{j,i} - u_{j,\setminus i})^2\right.$$

$$\left. + 2\partial_r^3\partial_u\ell_j(\check{r}_{j,i}, \check{u}_{j,i})(\tilde{u}_{j,i} - u_{j,\setminus i})(\tilde{r}_{j,i} - r_{j,\setminus i})\right)\frac{x_j^\top \tilde{H}_i^{-1} H_{\setminus i}^{-1} x_j}{n}. \tag{11}$$

using a second-order Taylor expansion, for $(\check{r}_{j,i}, \check{u}_{j,i})$ lying on the line connecting $\tilde{r}_{j,i}, \tilde{u}_{j,i}$ and $r_{j,\setminus i}, u_{j,\setminus i}$. The control of the last term follows from Propositions A.5 and A.14 and Lemma 10:

$$\left|\frac{1}{2n}\sum_{j\neq i}\partial_r^4\ell_j(r_{j,\setminus i}, u_{j,\setminus i})^2(\tilde{r}_{j,i} - r_{j,\setminus i})^2\frac{x_j^\top \tilde{H}_i^{-1} H_{\setminus i}^{-1} x_j}{n}\right| \leq \left\|\partial_r^4\ell\right\|_\infty \frac{1}{2\lambda^2}\left(\sup_{j\neq i}\frac{\|x_j\|}{\sqrt{n}}\sup_{j\neq i}|\tilde{r}_{j,i} - r_{j,\setminus i}|\right)^2$$

$$= O_{L_k}\left(\frac{\operatorname{polylog}(n)}{n}\right)$$

and

$$\left|\frac{1}{2n}\sum_{j\neq i}\partial_r^2\partial_u^2\ell_j(r_{j,\setminus i}, u_{j,\setminus i})^2(\tilde{u}_{j,i} - u_{j,\setminus i})^2\frac{x_j^\top \tilde{H}_i^{-1} H_{\setminus i}^{-1} x_j}{n}\right| \leq \left\|\partial_r^2\partial_u^2\ell\right\|_\infty \frac{1}{2\lambda^2}\left(\sup_{j\neq i}\frac{\|x_j\|}{\sqrt{n}}\sup_{j\neq i}|\tilde{u}_{j,i} - u_{j,\setminus i}|\right)^2$$

$$= O_{L_k}\left(\frac{\operatorname{polylog}(n)}{n}\right)$$

and

$$\left|\frac{1}{2n}\sum_{j\neq i}\partial_r^3\partial_u\ell_j(r_{j,\setminus i}, u_{j,\setminus i})^2(\tilde{u}_{j,i} - u_{j,\setminus i})(\tilde{r}_{j,i} - r_{j,\setminus i})\frac{x_j^\top \tilde{H}_i^{-1} H_{\setminus i}^{-1} x_j}{n}\right|$$

$$\leq \left\|\partial_r^2\partial_u^2\ell\right\|_\infty \frac{1}{2\lambda^2}\left(\sup_{j\neq i}\frac{\|x_j\|}{\sqrt{n}}\sup_{j\neq i}|\tilde{u}_{j,i} - u_{j,\setminus i}|\sup_{j\neq i}|\tilde{r}_{j,i} - r_{j,\setminus i}|\right) = O_{L_k}\left(\frac{\operatorname{polylog}(n)}{n}\right).$$

We finally focus on the second term in (11). Following (Barnfield et al., 2025), we first approximate $\tilde{H}_i$ by the $x_i-$independent Hessian $H_{\setminus i}$ to make explicit all statistical dependencies on $x_i$, which we subsequently carefully analyze. The corresponding correction term reads

$$\left|\frac{1}{n}\sum_{j\neq i}\left(\partial_r^3\ell_j(r_{j,\setminus i}, u_{j,\setminus i})(\tilde{r}_{j,i} - r_{j,\setminus i}) + \partial_r^2\partial_u\ell_j(r_{j,\setminus i}, u_{j,\setminus i})(\tilde{u}_{j,i} - u_{j,\setminus i})\right)\frac{x_j^\top(\tilde{H}_i^{-1} - H_{\setminus i}^{-1}) H_{\setminus i}^{-1} x_j}{n}\right|$$

$$\leq \left(\left\|\partial_r^3\ell\right\|_\infty \vee \left\|\partial_r^2\partial_u\ell\right\|_\infty\right)\left(\sup_{i\in[n]}\sup_{j\neq i}|r_{j,\setminus i} - \tilde{r}_{j,i}| + \sup_{i\in[n]}\sup_{j\neq i}|u_{j,\setminus i} - \tilde{u}_{j,i}|\right)\frac{1}{\lambda}\sup_{j\in[n]}\frac{\|x_j\|^2}{n}\left\|\tilde{H}_i^{-1} - H_{\setminus i}^{-1}\right\|$$

$$\leq O_{L_k}\left(\frac{\operatorname{polylog}(n)}{\sqrt{n}}\right)\frac{1}{\lambda^2}\left\|\tilde{H}_i - H_{\setminus i}\right\|.$$

The difference of the Hessians can further be bounded as

$$
\begin{aligned}
\left\|\tilde{H}_i - H_{\backslash i}\right\| &= \left\|\frac{1}{n}\sum_{j\neq i}\left(\partial_r^2\ell_j(r_{j,\backslash i}, u_{j,\backslash i}) - \partial_r^2\ell_j(\tilde{r}_{j,i}, \tilde{u}_{j,i})\right)x_j x_j^\top\right\| \\
&\leq \left(\left\|\partial_r^3\ell\right\|_\infty \vee \left\|\partial_r^2\partial_u\ell\right\|_\infty\right)\left(\sup_{i\in[n]}\sup_{j\neq i}|r_{j,\backslash i} - \tilde{r}_{j,i}| + \sup_{i\in[n]}\sup_{j\neq i}|u_{j,\backslash i} - \tilde{u}_{j,i}|\right)\frac{\|X\|^2}{n} \\
&= O_{L_k}\left(\frac{\mathrm{polylog}(n)}{\sqrt{n}}\right).
\end{aligned}
$$

Therefore, in summary, at the price of a correction of order $\mathrm{polylog}(n)/n$ in $L_k$, the surrogate Hessian $\tilde{H}_i$ in the second term of (11) may be replaced by the leave-$i$ out Hessian $H_{\backslash i}$. It thus suffices to study the quantity

$$
\begin{aligned}
\mathbb{E}&\left[\left(\frac{1}{n}\sum_{j\neq i}\left(\partial_r^3\ell_j(r_{j,\backslash i}, u_{j,\backslash i})(\tilde{r}_{j,i} - r_{j,\backslash i}) + \partial_r^2\partial_u\ell_j(r_{j,\backslash i}, u_{j,\backslash i})(\tilde{u}_{j,i} - u_{j,\backslash i})\right)\frac{x_j^\top H_{\backslash i}^{-2}x_j}{n}\right)^2\right] \\
&\leq 3\mathbb{E}\left[\left(\frac{1}{n}\sum_{j\neq i}\partial_r^3\ell_j(r_{j,\backslash i}, u_{j,\backslash i})\langle x_j, \eta_i\rangle\frac{x_j^\top H_{\backslash i}^{-2}x_j}{n}\right)^2\right] + 3\mathbb{E}\left[\left(\frac{1}{n}\sum_{j\neq i}\partial_r^3\ell_j(r_{j,\backslash i}, u_{j,\backslash i})\langle x_j, \zeta_i\rangle\frac{x_j^\top H_{\backslash i}^{-2}x_j}{n}\right)^2\right] \\
&\quad + 3\mathbb{E}\left[\left(\frac{1}{n}\sum_{j\neq i}\partial_r^2\partial_u\ell_j(r_{j,\backslash i}, u_{j,\backslash i})\langle x_j, \xi_i\rangle\frac{x_j^\top H_{\backslash i}^{-2}x_j}{n}\right)^2\right]
\end{aligned}
\tag{12}
$$

with the aim of establishing $O(1/n^2)$ control. We successively examine the three terms, leveraging the explicit expressions of $\eta_i, \zeta_i, \xi_i$.

$$
\begin{aligned}
\mathbb{E}\left[\left(\tfrac{1}{n}\sum_{j\neq i}\partial_r^3\ell_j(r_{j,\backslash i}, u_{j,\backslash i})\langle x_j, \eta_i\rangle\tfrac{x_j^\top H_{\backslash i}^{-2}x_j}{n}\right)^2\right] &= \mathbb{E}\left[\left(\frac{1}{n}\sum_{j\neq i}\partial_r\ell(\tilde{r}_{i,i})\partial_r^3\ell_j(r_{j,\backslash i}, u_{j,\backslash i})\frac{x_j^\top H_{\backslash i}^{-2}x_j x_j^\top H_{\backslash i}^{-1}}{n^2}x_i\right)^2\right] \\
&\leq \|\partial_r\ell\|_\infty^2\mathbb{E}\left[\left\|\frac{1}{n}\sum_{j\neq i}\partial_r^3\ell_j(r_{j,\backslash i}, u_{j,\backslash i})\frac{x_j^\top H_{\backslash i}^{-2}x_j H_{\backslash i}^{-1}x_j}{n^2}\right\|^2\right],
\end{aligned}
$$

where we took the expectation over $x_i$ only. The norm can be controlled as

$$
\left\|\frac{1}{n}\sum_{j\neq i}\partial_r^3\ell_j(r_{j,\backslash i}, u_{j,\backslash i})\frac{x_j^\top H_{\backslash i}^{-2}x_j H_{\backslash i}^{-1}x_j}{n^2}\right\| = \left\|\frac{1}{n^2}h^\top X_{\backslash i}H_{\backslash i}^{-1}\right\| \leq \frac{1}{n\lambda}\frac{\|X_{\backslash i}\|}{\sqrt{n}}\frac{\|h\|}{\sqrt{n}}.
$$

We have introduced the vector $h \in \mathbb{R}^{n-1}$ with entries $h_j = \partial_r^3\ell_j(r_{j,\backslash i}, u_{j,\backslash i})x_j^\top H_{\backslash i}^{-2}x_j/n$. From Lemma A.40, $\frac{\|X_{\backslash i}\|}{\sqrt{n}} = O_{L_k}(\mathrm{polylog}(n))$. Finally,

$$
\|h\| \leq \sqrt{n}\|\partial_r^3\ell\|_\infty\frac{1}{\lambda^2}\sup_{j\in[n]}\frac{\|x_j\|^2}{n} = O_{L_k}(\sqrt{n}\,\mathrm{polylog}(n)).
$$

Thus,

$$
\mathbb{E}\left[\left(\frac{1}{n}\sum_{j\neq i}\partial_r^3\ell_j(r_{j,\backslash i}, u_{j,\backslash i})\langle x_j, \eta_i\rangle\frac{x_j^\top H_{\backslash i}^{-2}x_j}{n}\right)^2\right] = O_{L_k}\left(\frac{\mathrm{polylog}(n)}{n^2}\right).
$$

We follow the same lines for the remaining terms of (12). On the one hand,

$$
\mathbb{E}\left[\left(\tfrac{1}{n}\sum_{j\neq i}\partial_r^3\ell_j(r_{j,\backslash i},u_{j,\backslash i})\langle x_j,\zeta_i\rangle\tfrac{x_j^\top H_{\backslash i}^{-2}x_j}{n}\right)^2\right] \leq \|\partial_u\ell_0\|_\infty^2\mathbb{E}\left[\left\|\frac{1}{n}\sum_{j\neq i}\partial_r^3\ell_j(r_{j,\backslash i},u_{j,\backslash i})\frac{x_j^\top H_{\backslash i}^{-2}x_j H_{0,\backslash i}^{-1}G_{\backslash i}H_{\backslash i}^{-1}x_j}{n^2}\right\|^2\right]
$$

$$
\leq \|\partial_u\ell_0\|_\infty^2\frac{1}{\lambda^6\lambda_0^2}\|\partial_r^3\ell\|_\infty^2\|\partial_r\partial_u\ell\|_\infty^2 O\left(\frac{\mathrm{polylog}(n)}{n^2}\right).
$$

On the other hand

$$
\mathbb{E}\left[\left(\tfrac{1}{n}\sum_{j\neq i}\partial_r^2\partial_u\ell_j(r_{j,\backslash i},u_{j,\backslash i})\langle x_j,\xi_i\rangle\tfrac{x_j^\top H_{\backslash i}^{-2}x_j}{n}\right)^2\right] \leq \|\partial_u\ell_0\|_\infty^2\mathbb{E}\left[\left\|\frac{1}{n}\sum_{j\neq i}\partial_r^2\partial_u\ell_j(r_{j,\backslash i},u_{j,\backslash i})\frac{x_j^\top H_{\backslash i}^{-2}x_j H_{0,\backslash i}^{-1}x_j}{n^2}\right\|^2\right]
$$

$$
\leq \|\partial_u\ell_0\|_\infty^2\frac{1}{\lambda^4\lambda_0^2}\|\partial_r^2\partial_u\ell\|_\infty^2 O\left(\frac{\mathrm{polylog}(n)}{n^2}\right).
$$

Returning to (12) finally allows one to reach the bound

$$
\mathbb{E}\left[\left(\frac{1}{n}\sum_{j\neq i}\left(\partial_r^3\ell_j(r_{j,\backslash i},u_{j,\backslash i})(\tilde{r}_{j,i}-r_{j,\backslash i})+\partial_r^2\partial_u\ell_j(r_{j,\backslash i},u_{j,\backslash i})(\tilde{u}_{j,i}-u_{j,\backslash i})\right)\frac{x_j^\top H_{\backslash i}^{-2}x_j}{n}\right)^2\right]=O\left(\frac{\mathrm{polylog}(n)}{n^2}\right)
$$

Thus, putting all the intermediary results together,

$$
2\sum_{i\in[n]}\mathbb{E}\left[(\mathfrak{V}_{\backslash i}-\tilde{\mathfrak{V}}_i)^2\right]=O_{L_k}\left(\frac{\mathrm{polylog}(n)}{n}\right).
$$

The Efron-Stein lemma then guarantees that

$$
\mathrm{Var}\left[\mathfrak{V}\right]=O\left(\frac{\mathrm{polylog}(n)}{n}\right),
$$

thereby concluding the proof. $\qquad\square$

We now prove a similar concentration inequality for $\upsilon_i,\mathfrak{X}$.

**Lemma A.22** ($\mathfrak{X}$ concentrates). *We remind $\upsilon_i=\tfrac{1}{n}x_i^\top H_{\backslash i}^{-1}G_{\backslash i}H_{0,\backslash i}^{-1}x_i$, $\mathfrak{X}=\frac{1}{n}\mathrm{tr}\left[H^{-1}GH_0^{-1}\right]$. $\upsilon_i$ admits an equivalent*

$$
\sup_{i\in[n]}|\upsilon_i-\mathfrak{X}|=O_{L_k}\left(\frac{\mathrm{polylog}(n)}{\lambda\sqrt{n}}\right).
$$

*Furthermore, $\mathfrak{X}$ concentrates, with*

$$
\mathrm{Var}\left[\mathfrak{X}\right]=O\left(\frac{\mathrm{polylog}(n)}{n}\right).
$$

*Proof.* The proof proceeds similarly to that for $\mathfrak{V}$. From Lemma G.2 of (El Karoui, 2018) applied to the symmetrized matrix $\tfrac{1}{2}(H_{\backslash i}^{-1}G_{\backslash i}H_{0,\backslash i}^{-1}+H_{0,\backslash i}^{-1}G_{\backslash i}H_{\backslash i}^{-1})$,

$$
\sup_{i\in[n]}\left|\upsilon_i-\mathfrak{X}_{\backslash i}\right|=O_{L_k}\left(\frac{\mathrm{polylog}(n)}{\lambda\sqrt{n}}\right),
$$

where $\mathfrak{X}_{\backslash i} = {}^1\!/n \operatorname{tr}\left[H_{\backslash i}^{-1} G_{\backslash i} H_{0,\backslash i}^{-1}\right]$. To complete the derivation of the first point, one now needs to bound $|\mathfrak{X}_{\backslash i} - \mathfrak{X}|$. Developing the difference,

$$
\begin{aligned}
\mathfrak{X}_{\backslash i} - \mathfrak{X} =& \frac{1}{n}\operatorname{tr}\left[H^{-1}(H_{\backslash i} - H)H_{\backslash i}^{-1}G_{\backslash i}H_{0,\backslash i}^{-1}\right] + \frac{1}{n}\operatorname{tr}\left[H^{-1}(G - G_{\backslash i})H_{0,\backslash i}^{-1}\right] \\
&+ \frac{1}{n}\operatorname{tr}\left[H^{-1}G_{\backslash i}H_0^{-1}(H_{0,\backslash i} - H_0)H_{0,\backslash i}^{-1}\right] + \frac{1}{n}\operatorname{tr}\left[H^{-1}(H_{\backslash i} - H)H_{\backslash i}^{-1}G_{\backslash i}H_0^{-1}(H_{0,\backslash i} - H_0)H_{0,\backslash i}^{-1}\right] \\
&+ \frac{1}{n}\operatorname{tr}\left[H^{-1}(G - G_{\backslash i})H_0^{-1}(H_{0,\backslash i} - H_0)H_{0,\backslash i}^{-1}\right] + \frac{1}{n}\operatorname{tr}\left[H^{-1}(H_{\backslash i} - H)H_{\backslash i}^{-1}(G - G_{\backslash i})H_{0,\backslash i}^{-1}\right] \\
&+ \frac{1}{n}\operatorname{tr}\left[H^{-1}(H_{\backslash i} - H)H_{\backslash i}^{-1}(G - G_{\backslash i})H_0^{-1}(H_{0,\backslash i} - H_0)H_{0,\backslash i}^{-1}\right]
\end{aligned} \tag{13}
$$

The control of each of these seven terms proceeds in much the same way. We illustrate the reasoning on the more complex last term.

$$
\left|\frac{1}{n}\operatorname{tr}\left[H^{-1}(H_{\backslash i} - H)H_{\backslash i}^{-1}(G - G_{\backslash i})H_0^{-1}(H_{0,\backslash i} - H_0)H_{0,\backslash i}^{-1}\right]\right| \tag{14}
$$

$$
\leq a_1 \frac{\|x_i\|^6}{\lambda^2\lambda_0^2 n^4} + a_2 \sup_{j\neq i}|u_j - u_{j,\backslash i}|\sup_{j\neq i}\frac{\|x_i\|^4\|x_j\|^2}{\lambda^2\lambda_0^2 n^3} + a_3(\sup_{j\neq i}|u_j - u_{j,\backslash i}| + \sup_{j\neq i}|r_j - r_{j,\backslash i}|)\sup_{j\neq i}\frac{\|x_i\|^4\|x_j\|^2}{\lambda^2\lambda_0^2 n^3}
$$

$$
+ a_4(\sup_{j\neq i}|u_j - u_{j,\backslash i}| + \sup_{j\neq i}|r_j - r_{j,\backslash i}|)\sup_{j\neq i}\sup_{j\neq i}\frac{\|x_i\|^4\|x_j\|^2}{\lambda^2\lambda_0^2 n^3}
$$

$$
+ a_5(\sup_{j\neq i}|u_j - u_{j,\backslash i}| + \sup_{j\neq i}|r_j - r_{j,\backslash i}|)\sup_{j\neq i}|u_j - u_{j,\backslash i}|
$$

$$
\left[\sup_{j\neq i}\sup_{k\neq i,j}\frac{|x_i^\top H_{\backslash i}^{-1}x_j x_j^\top H_0^{-1}x_k x_k^\top H_{0,\backslash i}^{-1}H^{-1}x_i|}{n^2} + \frac{1}{n^3}\sup_{j\neq i}\frac{\|x_i\|^2\|x_j\|^4}{\lambda^2\lambda_0^2}\right]
$$

$$
+ a_6(\sup_{j\neq i}|u_j - u_{j,\backslash i}| + \sup_{j\neq i}|r_j - r_{j,\backslash i}|)\sup_{j\neq i}|u_j - u_{j,\backslash i}|
$$

$$
\left[\sup_{j\neq i}\sup_{k\neq i,j}\frac{|x_j^\top H_{\backslash i}^{-1}x_i x_i^\top H_0^{-1}x_k x_k^\top H_{0,\backslash i}^{-1}H^{-1}x_j|}{n^2} + \frac{1}{n^3}\sup_{j\neq i}\frac{\|x_i\|^2\|x_j\|^4}{\lambda^2\lambda_0^2}\right]
$$

$$
+ a_7(\sup_{j\neq i}|u_j - u_{j,\backslash i}| + \sup_{j\neq i}|r_j - r_{j,\backslash i}|)^2\left[\sup_{j\neq i}\sup_{k\neq i,j}\frac{|x_j^\top H_{\backslash i}^{-1}x_k x_k^\top H_0^{-1}x_i x_i^\top H_{0,\backslash i}^{-1}H^{-1}x_j|}{n^2} + \frac{1}{n^3}\sup_{j\neq i}\frac{\|x_i\|^2\|x_j\|^4}{\lambda^2\lambda_0^2}\right]
$$

$$
+ a_8(\sup_{j\neq i}|u_j - u_{j,\backslash i}| + \sup_{j\neq i}|r_j - r_{j,\backslash i}|)^2\sup_{j\neq i}|u_j - u_{j,\backslash i}|
$$

$$
\left[\sup_{j\neq i}\sup_{k\neq i,j}\sup_{l\neq i,j,k}\frac{|x_j^\top H_{\backslash i}^{-1}x_k x_k^\top H_0^{-1}x_l x_l^\top H_{0,\backslash i}^{-1}H^{-1}x_j|}{n} + \frac{3}{n^2}\sup_{j\neq i}\sup_{k\neq i,j}\frac{\|x_k\|^2\|x_j\|^4}{\lambda^2\lambda_0^2} + \frac{1}{n^3}\sup_{j\neq i}\frac{\|x_j\|^6}{\lambda^2\lambda_0^2}\right]
$$

The absolute constants $a_m$ are simple products or maxima of the $\ell_\infty$ norms $\left\|\partial_r^2\ell\right\|_\infty, \left\|\partial_u^2\ell_0\right\|_\infty, \left\|\partial_r^3\ell\right\|_\infty, \left\|\partial_r\partial_u\ell\right\|_\infty,$ $\left\|\partial_r^2\partial_u\ell\right\|_\infty, \left\|\partial_r\partial_u^2\ell\right\|_\infty, \left\|\partial_u^3\ell_0\right\|_\infty.$ In the above, we resorted to a coarse Cauchy-Schwartz bound wherever it proved sufficient. It follows from Lemma A.41 that when $j \neq i$,

$$
\sup_{j\neq i}|x_i^\top H_{\backslash i}^{-1}x_j| \leq O_{L_k}\left(\sqrt{n}\,\mathrm{polylog}(n)\right).
$$

We now turn to the other terms. Let $i \neq j$, and let us focus on bounding the quadratic term $|x_i^\top H_0^{-1}x_j|$. We first note that

$$
\left\|H_0^{-1} - \left(H_{0,\backslash i} + \frac{\partial_u^2\ell_{0,i}(u_i)}{n}x_i x_i^\top\right)^{-1}\right\| \leq \frac{1}{\lambda_0^2}\left\|\partial_u^3\ell_0\right\|_\infty\sup_{j\neq i}|u_j - u_{j,\backslash i}|\frac{\|X_{\backslash i}\|^2}{n} = O_{L_k}\left(\frac{\mathrm{polylog}(n)}{\sqrt{n}}\right),
$$

using Lemma A.40 and Proposition A.5. Using Woodbury's inversion lemma to make $H_{0,\backslash i}$ appear,

$$
\begin{aligned}
\sup_{j \neq i} \left| x_i^\top H_0^{-1} x_j \right| &= \sup_{j \neq i} \left| x_i^\top \left( H_{0,\backslash i}^{-1} - \frac{\partial_u^2 \ell_{0,i}(u_i)}{n} \frac{1}{1 + \frac{\partial_u^2 \ell_{0,i}(u_i)}{n} x_i^\top H_{0,\backslash i}^{-1} x_i} H_{0,\backslash i}^{-1} x_i x_i^\top H_{0,\backslash i}^{-1} \right) x_j \right| \\
&\quad + O_{L_k} \left( \sqrt{n} \mathrm{polylog}(n) \right) \\
&\leq \sup_{j \neq i} \left| x_i^\top H_{0,\backslash i}^{-1} x_j \right| + \frac{\left\| \partial_u^2 \ell_0 \right\|_\infty}{\lambda_0} \frac{\|x_i\|^2}{n} \sup_{j \neq i} \left| x_i^\top H_{0,\backslash i}^{-1} x_j \right| + O_{L_k} \left( \sqrt{n} \mathrm{polylog}(n) \right) \\
&= O_{L_k} \left( \sqrt{n} \mathrm{polylog}(n) \right),
\end{aligned}
$$

again using Lemma A.41. Similarly,

$$
\left\| H^{-1} - \left( H_{\backslash i} + \frac{\partial_r^2 \ell_i(r_i, u_i)}{n} x_i x_i^\top \right)^{-1} \right\| = O_{L_k} \left( \frac{\mathrm{polylog}(n)}{\sqrt{n}} \right),
$$

from which it follows that for $j \neq i$,

$$
\begin{aligned}
\sup_{j \neq i} \left| x_j^\top H_{0,\backslash i}^{-1} H^{-1} x_i \right| &= \sup_{j \neq i} \left| x_j^\top H_{0,\backslash i}^{-1} \left( H_{\backslash i}^{-1} - \frac{\partial_r^2 \ell_i(r_i, u_i)}{n} \frac{1}{1 + \frac{\partial_r^2 \ell_i(r_i, u_i)}{n} x_i^\top H_{\backslash i}^{-1} x_i} H_{\backslash i}^{-1} x_i x_i^\top H_{\backslash i}^{-1} \right) x_j \right| \\
&\quad + O_{L_k} \left( \sqrt{n} \mathrm{polylog}(n) \right) \\
&\leq \sup_{j \neq i} \left| x_i^\top H_{0,\backslash i}^{-1} x_j \right| + \frac{\left\| \partial_u^2 \ell_0 \right\|_\infty}{\lambda_0} \frac{\|x_i\|^2}{n} \sup_{j \neq i} \left| x_i^\top H_{0,\backslash i}^{-1} x_j \right| + O_{L_k} \left( \sqrt{n} \mathrm{polylog}(n) \right) \\
&= O_{L_k} \left( \sqrt{n} \mathrm{polylog}(n) \right).
\end{aligned}
$$

Finally, one needs to control $\sup_{j \neq i} \sup_{k \neq i,j} \left| x_j^\top H_{0,\backslash i}^{-1} H^{-1} x_k \right|$ for $k \neq j$ and $k, j \neq i$. To this end, we will approximate both $H_{0,\backslash j}^{-1}$ and $H^{-1}$ by their leave-$j$ out versions. We have already controlled the difference between $H^{-1}$ and $\left( H_{\backslash i} + \partial_r^2 \ell_j(r_j, u_j)/n x_j x_j^\top \right)^{-1}$. By the same token,

$$
\begin{aligned}
\sup_{j \neq i} \left\| H_{0,\backslash i}^{-1} - \left( H_{0,\backslash i,\backslash j} + \frac{\partial_u^2 \ell_{0,j}(u_{j,\backslash i})}{n} x_j x_j^\top \right)^{-1} \right\| &\leq \frac{1}{\lambda_0^2} \left\| \partial_u^3 \ell_0 \right\|_\infty \sup_{k \neq i,j} |u_{k,\backslash i} - u_{k,\backslash i,\backslash j}| \sup_{j \neq i} \frac{\left\| X_{\backslash i,\backslash j} \right\|^2}{n} \\
&= O_{L_k} \left( \frac{\mathrm{polylog}(n)}{\sqrt{n}} \right),
\end{aligned}
$$

We denoted $X_{\backslash i,\backslash j} \in \mathbb{R}^{(n-2) \times d}$ the matrix with rows $\{x_k\}_{k \neq i,j}$ and

$$
H_{0,\backslash i,\backslash j} = \frac{1}{n} \sum_{k \neq i,j} \partial_u^2 \ell_0(u_{k,\backslash i,\backslash j}) x_k x_k^\top + \lambda_0 I_d,
$$

using the fact that Proposition A.5 and Lemma A.40 directly apply to $X_{\backslash i}$, yielding control over the leave-two-out quantities.

Thus,

$$\sup_{j\neq i}\sup_{k\neq i,j}\left|x_j^\top H_{0,\backslash i}^{-1}H^{-1}x_k\right|$$

$$= \sup_{j\neq i}\sup_{k\neq i,j}\left|x_j^\top\left(H_{\backslash j}^{-1} - \frac{\partial_r^2\ell_j(r_j,u_j)}{n}\frac{1}{1+\frac{\partial_r^2\ell_j(r_j,u_j)}{n}x_j^\top H_{\backslash j}^{-1}x_j}H_{\backslash j}^{-1}x_jx_j^\top H_{\backslash j}^{-1}\right)\right.$$

$$\left.\left(H_{0,\backslash i,\backslash j}^{-1} - \frac{\partial_u^2\ell_{0,j}(u_{j,\backslash i})}{n}\frac{1}{1+\frac{\partial_u^2\ell_{0,j}(u_{j,\backslash i})}{n}x_j^\top H_{0,\backslash i,\backslash j}^{-1}x_j}H_{0,\backslash i,\backslash j}^{-1}x_jx_j^\top H_{0,\backslash i,\backslash j}^{-1}\right)x_k\right|$$

$$+ O_{L_k}\left(\mathrm{polylog}(n)\sqrt{n}\right)$$

$$= \sup_{j\neq i}\sup_{k\neq i,j}\left|x_j^\top H_{\backslash j}^{-1}H_{0,\backslash i,\backslash j}^{-1}x_k\right| + \left\|\partial_u^2\ell_0\right\|_\infty\sup_{j\neq i}\frac{\|x_j\|^2}{\lambda\lambda_0 n}\sup_{j\neq i}\sup_{k\neq i,j}\left|x_j^\top H_{0,\backslash i,\backslash j}^{-1}x_k\right|$$

$$+ \left\|\partial_r^2\ell\right\|_\infty\sup_{j\neq i}\frac{\|x_j\|^2}{\lambda n}\sup_{j\neq i}\sup_{k\neq i,j}\left|x_j^\top H_{\backslash j}^{-1}H_{0,\backslash i,\backslash j}^{-1}x_k\right|$$

$$+ \left\|\partial_r^2\ell\right\|_\infty\left\|\partial_u^2\ell_0\right\|_\infty\sup_{j\neq i}\frac{\|x_j\|^4}{\lambda^2\lambda_0 n^2}\sup_{j\neq i}\sup_{k\neq i,j}\left|x_j^\top H_{0,\backslash i,\backslash j}^{-1}x_k\right| + O_{L_k}\left(\mathrm{polylog}(n)\sqrt{n}\right)$$

$$= O_{L_k}\left(\mathrm{polylog}(n)\sqrt{n}\right),$$

using repeatedly the bound offered by Lemma A.41 in the last step. The same strategy shows

$$\sup_{j\neq i}\sup_{k\neq i,j}\left|x_j^\top H_{\backslash i}^{-1}x_k\right| = O_{L_k}\left(\mathrm{polylog}(n)\sqrt{n}\right).$$

Putting all these results together finally leads to the bounds

$$\sup_{j\neq i}\sup_{k\neq i,j}\frac{|x_i^\top H_{\backslash i}^{-1}x_jx_j^\top H_0^{-1}x_kx_k^\top H_{0,\backslash i}^{-1}H^{-1}x_i|}{n^2} = O_{L_k}\left(\frac{\mathrm{polylog}(n)}{\sqrt{n}}\right)$$

$$\sup_{j\neq i}\sup_{k\neq i,j}\frac{|x_j^\top H_{\backslash i}^{-1}x_kx_k^\top H_0^{-1}x_ix_i^\top H_{0,\backslash i}^{-1}H^{-1}x_j|}{n^2} = O_{L_k}\left(\frac{\mathrm{polylog}(n)}{\sqrt{n}}\right)$$

$$\sup_{j\neq i}\sup_{k\neq i,j}\frac{|x_j^\top H_{\backslash i}^{-1}x_ix_i^\top H_0^{-1}x_kx_k^\top H_{0,\backslash i}^{-1}H^{-1}x_j|}{n^2} = O_{L_k}\left(\frac{\mathrm{polylog}(n)}{\sqrt{n}}\right)$$

$$\sup_{j\neq i}\sup_{k\neq i,j}\sup_{l\neq i,j,k}\frac{|x_j^\top H_{\backslash i}^{-1}x_kx_k^\top H_0^{-1}x_lx_l^\top H_{0,\backslash i}^{-1}H^{-1}x_j|}{n} = O_{L_k}\left(\mathrm{polylog}(n)\sqrt{n}\right).$$

Incorporating these bounds into (14), and using Proposition A.5 and Proposition A.14 to bound the differences of residuals, allows one to reach

$$\left|\frac{1}{n}\mathrm{tr}\left[H^{-1}(H_{\backslash i}-H)H_{\backslash i}^{-1}(G-G_{\backslash i})H_0^{-1}(H_{0,\backslash i}-H_0)H_{0,\backslash i}^{-1}\right]\right| = O_{L_k}\left(\frac{\mathrm{polylog}(n)}{\sqrt{n}}\right).$$

An identical proof may be repeated for each of the other terms of (13), and finally yield

$$|\mathfrak{X}_{\backslash i}-\mathfrak{X}| = O_{L_k}\left(\frac{\mathrm{polylog}(n)}{\sqrt{n}}\right),$$

implying the first claim.

We now turn to the second claim. Starting again from the Efron-Stein lemma,

$$\mathrm{Var}\left[\mathfrak{X}\right] \leq \sum_{i\in[n]}\mathbb{E}\left[(\mathfrak{X}-\mathfrak{X}_{\backslash i})^2\right] \leq 2\sum_{i\in[n]}\left(\mathbb{E}\left[(\mathfrak{X}-\tilde{\mathfrak{X}}_i)^2\right] + \mathbb{E}\left[(\tilde{\mathfrak{X}}_i-\mathfrak{X}_{\backslash i})^2\right]\right),$$

We defined $\tilde{\mathfrak{X}}_i = {}^1\!/n \operatorname{tr}\!\left[\tilde{H}_i^{-1}\tilde{G}_i\tilde{H}_{0,i}^{-1}\right]$. We focus on the first term $\mathbb{E}\left[(\mathfrak{X}-\tilde{\mathfrak{X}}_i)^2\right]$. We can decompose

$$\mathfrak{X}-\tilde{\mathfrak{X}}_i = \frac{1}{n}\operatorname{tr}\!\left[(H^{-1}-\tilde{H}_i^{-1})\tilde{G}_i\tilde{H}_{0,i}^{-1}\right] + \frac{1}{n}\operatorname{tr}\!\left[H^{-1}(G-\tilde{G}_i)\tilde{H}_{0,i}^{-1}\right] + \frac{1}{n}\operatorname{tr}\!\left[H^{-1}G(H_0^{-1}-\tilde{H}_{0,i}^{-1})\right].$$

The first term can be explicitly written as

$$\left|\frac{1}{n}\operatorname{tr}\!\left[(H^{-1}-\tilde{H}_i^{-1})\tilde{G}_i\tilde{H}_{0,i}^{-1}\right]\right|$$

$$= \left|\frac{1}{n}\sum_{j\neq i}\left[\partial_r^3\ell_j(\check{r}_{j,i},\check{u}_{j,i})(r_j-\tilde{r}_{j,i})+\partial_r^2\partial_u\ell_j(\check{r}_{j,i},\check{u}_{j,i})(u_j-\tilde{u}_{j,i})\right]\frac{x_j^\top\tilde{H}_i^{-1}\tilde{G}_i\tilde{H}_{0,i}^{-1}H^{-1}x_j}{n} + \partial_r^2\ell_i(r_i,u_i)\frac{x_i^\top\tilde{H}_i^{-1}\tilde{G}_i\tilde{H}_{0,i}^{-1}H^{-1}x_i}{n^2}\right|$$

$$\leq \frac{1}{n}\left(\|h^r\|\|\delta^r\|+\|h^u\|\|\delta^u\|\right) + O_{L_k}\left(\frac{\operatorname{polylog}(n)}{n}\right)$$

for some $\check{r}_{j,i},\check{u}_{j,i}$ lying on the segment connecting $\tilde{r}_{j,i},\tilde{u}_{j,i}$ and $r_j,u_j$, from the mean value theorem. We introduced the vectors $h^r\in\mathbb{R}^n, h^u\in\mathbb{R}^n$, with entries $h_j^r = \partial_r^3\ell_j(\check{r}_{j,i},\check{u}_{j,i})x_j^\top\tilde{H}_i^{-1}\tilde{G}_i\tilde{H}_{0,i}^{-1}H^{-1}x_j/n$ and $h_j^u = \partial_r^2\partial_u\ell_j(\check{r}_{j,i},\check{u}_{j,i})x_j^\top\tilde{G}_i\tilde{H}_{0,i}^{-1}x_j/n$, alongside $\delta_r,\delta_u\in\mathbb{R}^n$ with entries $\delta_{r,j}=(r_j-\tilde{r}_{j,i})$ and $\delta_{u,j}=(u_j-\tilde{u}_{j,i})$. Lemma A.6 and Lemma A.16 ensure that $\|\delta_r\|,\|\delta_u\| = O_{L_k}(\operatorname{polylog}(n)/\sqrt{n})$. We now aim to establish $O_{L_k}(\operatorname{polylog}(n)\sqrt{n})$ control of $\|h^u\|,\|h^r\|$. These follow from the bounds

$$\|h^r\| \leq \sqrt{n}\|\partial_r^3\ell\|_\infty \sup_{j\in[n]}\frac{\|x_j\|^2}{\lambda_0\lambda^2 n}\left\|\tilde{G}_i\right\|,$$

$$\|h^u\| \leq \sqrt{n}\|\partial_r^2\partial_u\ell\|_\infty \sup_{j\in[n]}\frac{\|x_j\|^2}{\lambda_0\lambda^2 n}\left\|\tilde{G}_i\right\|,$$

Lemma A.40 and the bound $\left\|\tilde{G}_i\right\| \leq \|\partial_r\partial_u\ell\|_\infty\|X\|^2/n = O_{L_k}(\operatorname{polylog}(n))$. Similarly, one can establish

$$\frac{1}{n}\operatorname{tr}\!\left[(H^{-1}-\tilde{H}_i^{-1})\tilde{G}_i\tilde{H}_{0,i}^{-1}\right], \frac{1}{n}\operatorname{tr}\!\left[H^{-1}(G-\tilde{G}_i)\tilde{H}_{0,i}^{-1}\right], \frac{1}{n}\operatorname{tr}\!\left[H^{-1}G(H_0^{-1}-\tilde{H}_{0,i}^{-1})\right] = O_{L_k}\left(\frac{\operatorname{polylog}(n)}{n}\right),$$

from which it follows that

$$\mathbb{E}\left[(\mathfrak{X}-\tilde{\mathfrak{X}}_i)^2\right] = O\left(\frac{\operatorname{polylog}(n)}{n^2}\right).$$

We now turn to controlling the second term $\mathbb{E}\left[(\mathfrak{X}_{\backslash i}-\tilde{\mathfrak{X}}_i)^2\right]$ in the Efron-Stein upper bound. We start from a similar decomposition

$$(\mathfrak{X}_{\backslash i}-\tilde{\mathfrak{X}}_i)^2 \leq \frac{3}{n^2}\operatorname{tr}\!\left[(H_{\backslash i}^{-1}-\tilde{H}_i^{-1})\tilde{G}_i\tilde{H}_{0,i}^{-1}\right]^2 + \frac{3}{n}\operatorname{tr}\!\left[H_{\backslash i}^{-1}(G_{\backslash i}-\tilde{G}_i)\tilde{H}_{0,i}^{-1}\right]^2 + \frac{3}{n}\operatorname{tr}\!\left[H_{\backslash i}^{-1}G_{\backslash i}(H_{0,\backslash i}^{-1}-\tilde{H}_{0,i}^{-1})\right]^2 \tag{15}$$

Let us examine the first term; the two other terms can be controlled using identical steps. The strategy follows very closely that of the corresponding step in the proof of the concentration of $\mathfrak{V}$. Using a second-order Taylor expansion:

$$\frac{\operatorname{tr}\!\left[(H_{\backslash i}^{-1}-\tilde{H}_i^{-1})\tilde{G}_i\tilde{H}_{0,i}^{-1}\right]}{n}$$

$$= \frac{1}{n}\sum_{j\neq i}\left(\partial_r^3\ell_j(r_{j,\backslash i},u_{j,\backslash i})(\tilde{r}_{j,i}-r_{j,\backslash i})+\partial_r^2\partial_u\ell_j(r_{j,\backslash i},u_{j,\backslash i})(\tilde{u}_{j,i}-u_{j,\backslash i})\right)\frac{x_j^\top\tilde{H}_i^{-1}\tilde{G}_i\tilde{H}_{0,i}^{-1}H_{\backslash i}^{-1}x_j}{n}$$

$$+ \frac{1}{2n}\sum_{j\neq i}\left(\partial_r^4\ell_j(r_{j,\backslash i},u_{j,\backslash i})^2(\tilde{r}_{j,i}-r_{j,\backslash i})^2 + \partial_r^2\partial_u^2\ell_j(r_{j,\backslash i},u_{j,\backslash i})(\tilde{u}_{j,i}-u_{j,\backslash i})^2\right.$$

$$\left. + 2\partial_r^3\partial_u\ell_j(r_{j,\backslash i},u_{j,\backslash i})(\tilde{u}_{j,i}-u_{j,\backslash i})(\tilde{r}_{j,i}-r_{j,\backslash i})\right)\frac{x_j^\top\tilde{H}_i^{-1}\tilde{G}_i\tilde{H}_{0,i}^{-1}H_{\backslash i}^{-1}x_j}{n}. \tag{16}$$

The first term can be bounded as

$$\left| \frac{1}{n} \partial_r^2 \ell_i(\tilde{r}_{i,i}, \tilde{u}_{i,i}) \frac{x_i^\top \tilde{H}_i^{-1} \tilde{G}_i \tilde{H}_{0,i}^{-1} H_{\backslash i}^{-1} x_i}{n} \right| = \frac{\left\| \partial_r^2 \ell \right\|_\infty \left\| \partial_r \partial_u \ell \right\|_\infty}{\lambda^2 \lambda_0} O_{L_k}\left( \frac{\mathrm{polylog}(n)}{n} \right).$$

The control of the last term follows from Propositions A.5 and A.14 and Lemma 10:

$$\left| \frac{1}{2n} \sum_{j \neq i} \partial_r^4 \ell_j(r_{j,\backslash i}, u_{j,\backslash i})^2 (\tilde{r}_{j,i} - r_{j,\backslash i})^2 \frac{x_j^\top \tilde{H}_i^{-1} \tilde{G}_i \tilde{H}_{0,i}^{-1} H_{\backslash i}^{-1} x_j}{n} \right| \leq \frac{\left\| \partial_r^4 \ell \right\|_\infty \left\| \partial_r \partial_u \ell \right\|_\infty}{2\lambda^2 \lambda_0} \left( \sup_{j \neq i} \frac{\|x_j\|}{\sqrt{n}} \sup_{j \neq i} |\tilde{r}_{j,i} - r_{j,\backslash i}| \right)^2$$

$$= O_{L_k}\left( \frac{\mathrm{polylog}(n)}{n} \right)$$

and

$$\left| \frac{1}{2n} \sum_{j \neq i} \partial_r^2 \partial_u^2 \ell_j(r_{j,\backslash i}, u_{j,\backslash i})^2 (\tilde{u}_{j,i} - u_{j,\backslash i})^2 \frac{x_j^\top \tilde{H}_i^{-1} \tilde{G}_i \tilde{H}_{0,i}^{-1} H_{\backslash i}^{-1} x_j}{n} \right| \leq \frac{\left\| \partial_r^2 \partial_u^2 \ell \right\|_\infty \left\| \partial_r \partial_u \ell \right\|_\infty}{2\lambda^2 \lambda_0} \left( \sup_{j \neq i} \frac{\|x_j\|}{\sqrt{n}} \sup_{j \neq i} |\tilde{u}_{j,i} - u_{j,\backslash i}| \right)$$

$$= O_{L_k}\left( \frac{\mathrm{polylog}(n)}{n} \right)$$

and

$$\left| \frac{1}{2n} \sum_{j \neq i} \partial_r^3 \partial_u \ell_j(r_{j,\backslash i}, u_{j,\backslash i})^2 (\tilde{u}_{j,i} - u_{j,\backslash i})(\tilde{r}_{j,i} - r_{j,\backslash i}) \frac{x_j^\top \tilde{H}_i^{-1} \tilde{G}_i \tilde{H}_{0,i}^{-1} H_{\backslash i}^{-1} x_j}{n} \right|$$

$$\leq \frac{\left\| \partial_r \partial_u \ell \right\|_\infty \left\| \partial_r^2 \partial_u^2 \ell \right\|_\infty}{2\lambda^2 \lambda_0} \left( \sup_{j \neq i} \frac{\|x_j\|}{\sqrt{n}} \sup_{j \neq i} |\tilde{u}_{j,i} - u_{j,\backslash i}| \sup_{j \neq i} |\tilde{r}_{j,i} - r_{j,\backslash i}| \right) = O_{L_k}\left( \frac{\mathrm{polylog}(n)}{n} \right).$$

The control of the second term of (16) requires a finer analysis of the statistical dependencies on $x_i$. As for the proof of the concentration of $\mathfrak{V}$, the key step is to approximate the matrices $\tilde{G}_i, \tilde{H}_{0,i}^{-1}$ by their leave-$i$ out approximations, in order to unravel all statistical correlations with $x_i$. On the one hand,

$$\left| \frac{1}{n} \sum_{j \neq i} \left( \partial_r^3 \ell_j(r_{j,\backslash i}, u_{j,\backslash i})(\tilde{r}_{j,i} - r_{j,\backslash i}) + \partial_r^2 \partial_u \ell_j(r_{j,\backslash i}, u_{j,\backslash i})(\tilde{u}_{j,i} - u_{j,\backslash i}) \right) \frac{x_j^\top \left( \tilde{H}_i^{-1} \tilde{G}_i \tilde{H}_{0,i}^{-1} - H_{\backslash i}^{-1} G_{\backslash i} H_{0,\backslash i}^{-1} \right) H_{\backslash i}^{-1} x_j}{n} \right|$$

$$\leq a_1 \left( \sup_{i \in [n]} \sup_{j \neq i} |r_{j,\backslash i} - \tilde{r}_{j,i}| + \sup_{i \in [n]} \sup_{j \neq i} |u_{j,\backslash i} - \tilde{u}_{j,i}| \right) \frac{1}{\lambda \lambda_0} \sup_{j \in [n]} \frac{\|x_j\|^2}{n}$$

$$\left[ b_1 \sup_{i \in [n]} \left\| \tilde{H}_i^{-1} - H_{\backslash i}^{-1} \right\| + b_2 \sup_{i \in [n]} \left\| \tilde{G}_i - G_{\backslash i} \right\| + b_3 \sup_{i \in [n]} \left\| \tilde{H}_{0,i}^{-1} - H_{0,\backslash i}^{-1} \right\| + b_4 \sup_{i \in [n]} \left\| \tilde{H}_i^{-1} - H_{\backslash i}^{-1} \right\| \left\| \tilde{H}_{0,i}^{-1} - H_{0,\backslash i}^{-1} \right\| \right.$$

$$\left. + b_5 \sup_{i \in [n]} \left\| \tilde{H}_i^{-1} - H_{\backslash i}^{-1} \right\| \left\| \tilde{G}_i - G_{\backslash i} \right\| + b_6 \sup_{i \in [n]} \left\| \tilde{G}_i - G_{\backslash i} \right\| \left\| \tilde{H}_{0,i}^{-1} - H_{0,\backslash i}^{-1} \right\| + b_7 \sup_{i \in [n]} \left\| \tilde{H}_i^{-1} - H_{\backslash i}^{-1} \right\| \left\| \tilde{G}_i - G_{\backslash i} \right\| \left\| \tilde{H}_{0,i}^{-1} - H_{0,\backslash i}^{-1} \right\| \right]$$

with $a_1 = (\left\| \partial_r^3 \ell \right\|_\infty \vee \left\| \partial_r^2 \partial_u \ell \right\|_\infty) \left\| \partial_r \partial_u \ell \right\|_\infty$, and $b_1, b_2, b_3, b_4, b_5, b_6, b_7 = O_{L_k}(\mathrm{polylog}(n))$. We recall from the proof of Lemma A.21 that

$$\left\| \tilde{H}_i - H_{\backslash i} \right\| = O_{L_k}\left( \frac{\mathrm{polylog}(n)}{\sqrt{n}} \right).$$

Similarly,

$$\left\| \tilde{H}_{0,i} - H_{0,\backslash i} \right\| = \left\| \frac{1}{n} \sum_{j \neq i} \left( \partial_u^2 \ell_{0,j}(u_{j,\backslash i}) - \partial_u^2 \ell_{0,j}(\tilde{u}_{j,i}) \right) x_j x_j^\top \right\|$$

$$\leq \left\| \partial_u^3 \ell_0 \right\|_\infty \sup_{i \in [n]} \sup_{j \neq i} |u_{j,\backslash i} - \tilde{u}_{j,i}| \frac{\|X\|^2}{n}$$

$$= O_{L_k}\left( \frac{\mathrm{polylog}(n)}{\sqrt{n}} \right).$$

and

$$\left\| \tilde{G}_i - G_{\backslash i} \right\| = \left\| \frac{1}{n} \sum_{j \neq i} \left( \partial_r \partial_u \ell_j (r_{j,\backslash i}, u_{j,\backslash i}) - \partial_r \partial_u \ell_j (\tilde{r}_{j,i}, \tilde{u}_{j,i}) \right) \frac{x_j x_j^\top}{n} \right\|$$

$$\leq \left( \left\| \partial_r^2 \partial_u \ell \right\|_\infty \vee \left\| \partial_r \partial_u^2 \ell \right\|_\infty \right) \left( \sup_{i \in [n]} \sup_{j \neq i} |r_{j,\backslash i} - \tilde{r}_{j,i}| + \sup_{i \in [n]} \sup_{j \neq i} |u_{j,\backslash i} - \tilde{u}_{j,i}| \right) \frac{\|X\|^2}{n}$$

$$= O_{L_k} \left( \frac{\mathrm{polylog}(n)}{\sqrt{n}} \right)$$

Thus,

$$\left| \frac{1}{n} \sum_{j \neq i} \left( \partial_r^3 \ell_j (r_{j,\backslash i}, u_{j,\backslash i})(\tilde{r}_{j,i} - r_{j,\backslash i}) + \partial_r^2 \partial_u \ell_j (r_{j,\backslash i}, u_{j,\backslash i})(\tilde{u}_{j,i} - u_{j,\backslash i}) \right) \frac{x_j^\top H_{\backslash i}^{-1} \left( \tilde{H}_i^{-1} \tilde{G}_i \tilde{H}_{0,i}^{-1} - H_{\backslash i}^{-1} G_{\backslash i} H_{0,\backslash i}^{-1} \right) x_j}{n} \right| = O_{L_k} \left( \frac{\mathrm{polylog}(n)}{n} \right),$$

and one can replace the product $\tilde{H}_i^{-1} \tilde{G}_i \tilde{H}_{0,i}^{-1}$ in the second term of (16) by the leave-one-out approximation $H_{\backslash i}^{-1} G_{\backslash i} H_{0,\backslash i}^{-1}$, at the price of a $O_{L_k} (\mathrm{polylog}(n)/n)$ correction. One can thus focus on studying

$$\mathbb{E} \left[ \left( \frac{1}{n} \sum_{j \neq i} \left( \partial_r^3 \ell_j (r_{j,\backslash i}, u_{j,\backslash i})(\tilde{r}_{j,i} - r_{j,\backslash i}) + \partial_r^2 \partial_u \ell_j (r_{j,\backslash i}, u_{j,\backslash i})(\tilde{u}_{j,i} - u_{j,\backslash i}) \right) \frac{x_j^\top H_{\backslash i}^{-2} G_{\backslash i} H_{0,\backslash i}^{-1} x_j}{n} \right)^2 \right]$$

$$\leq 3\mathbb{E} \left[ \left( \frac{1}{n} \sum_{j \neq i} \partial_r^3 \ell_j (r_{j,\backslash i}, u_{j,\backslash i}) \langle x_j, \eta_i \rangle \frac{x_j^\top H_{\backslash i}^{-2} G_{\backslash i} H_{0,\backslash i}^{-1} x_j}{n} \right)^2 \right]$$

$$+ 3\mathbb{E} \left[ \left( \frac{1}{n} \sum_{j \neq i} \partial_r^3 \ell_j (r_{j,\backslash i}, u_{j,\backslash i}) \langle x_j, \zeta_i \rangle \frac{x_j^\top H_{\backslash i}^{-2} G_{\backslash i} H_{0,\backslash i}^{-1} x_j}{n} \right)^2 \right]$$

$$+ 3\mathbb{E} \left[ \left( \frac{1}{n} \sum_{j \neq i} \partial_r^2 \partial_u \ell_j (r_{j,\backslash i}, u_{j,\backslash i}) \langle x_j, \xi_i \rangle \frac{x_j^\top H_{\backslash i}^{-2} G_{\backslash i} H_{0,\backslash i}^{-1} x_j}{n} \right)^2 \right] \tag{17}$$

with the aim of establishing $O(1/n^2)$ control. The remainder of the reasoning proceeds in exaclty the same way as the corresponding step in the proof of Lemma A.21, where the corresponding quantity (17) is controlled. We have

$$\mathbb{E} \left[ \left( \frac{1}{n} \sum_{j \neq i} \partial_r^3 \ell_j (r_{j,\backslash i}, u_{j,\backslash i}) \langle x_j, \eta_i \rangle \frac{x_j^\top H_{\backslash i}^{-2} G_{\backslash i} H_{0,\backslash i}^{-1} x_j}{n} \right)^2 \right] = \mathbb{E} \left[ \left( \frac{1}{n} \sum_{j \neq i} \partial_r \ell(\tilde{r}_{i,i}) \partial_r^3 \ell_j (r_{j,\backslash i}, u_{j,\backslash i}) \frac{x_j^\top H_{\backslash i}^{-2} G_{\backslash i} H_{0,\backslash i}^{-1} x_j x_j^\top H_{\backslash i}^{-1}}{n^2} x_i \right)^2 \right]$$

$$\leq \|\partial_r \ell\|_\infty^2 \mathbb{E} \left[ \left\| \frac{1}{n} \sum_{j \neq i} \partial_r^3 \ell_j (r_{j,\backslash i}, u_{j,\backslash i}) \frac{x_j^\top H_{\backslash i}^{-2} G_{\backslash i} H_{0,\backslash i}^{-1} x_j H_{\backslash i}^{-1} x_j}{n^2} \right\|^2 \right]$$

$$\leq \|\partial_r \ell\|_\infty^2 \|\partial_r \partial_u \ell\|_\infty^2 \|\partial_r^3 \ell\|_\infty^2 \frac{1}{\lambda^6 \lambda_0^2} \|\partial_r \ell\|_\infty^2 O_{L_k} \left( \frac{\mathrm{polylog}(n)}{n^2} \right).$$

and

$$\mathbb{E} \left[ \left( \frac{1}{n} \sum_{j \neq i} \partial_r^3 \ell_j (r_{j,\backslash i}, u_{j,\backslash i}) \langle x_j, \zeta_i \rangle \frac{x_j^\top H_{\backslash i}^{-2} G_{\backslash i} H_{0,\backslash i}^{-1} x_j}{n} \right)^2 \right] \leq \|\partial_u \ell_0\|_\infty^2 \mathbb{E} \left[ \left\| \frac{1}{n} \sum_{j \neq i} \partial_r^3 \ell_j (r_{j,\backslash i}, u_{j,\backslash i}) \frac{x_j^\top H_{\backslash i}^{-2} G_{\backslash i} H_{0,\backslash i}^{-1} x_j H_{0,\backslash i}^{-1} G_{\backslash i} H_{\backslash i}^{-1} x_j}{n^2} \right\|^2 \right]$$

$$\leq \|\partial_u \ell_0\|_\infty^2 \frac{1}{\lambda^6 \lambda_0^2} \|\partial_r^3 \ell\|_\infty^2 \|\partial_r \partial_u \ell\|_\infty^4 O \left( \frac{\mathrm{polylog}(n)}{n^2} \right).$$

and finally

$$\mathbb{E}\left[\left(\frac{1}{n}\sum_{j\neq i}\partial_r^2\partial_u\ell_j(r_{j,\backslash i},u_{j,\backslash i})\langle x_j,\xi_i\rangle\frac{x_j^\top H_{\backslash i}^{-2}G_{\backslash i}H_{0,\backslash i}^{-1}x_j}{n}\right)^2\right] \leq \|\partial_u\ell_0\|_\infty^2\mathbb{E}\left[\left\|\frac{1}{n}\sum_{j\neq i}\partial_r^2\partial_u\ell_j(r_{j,\backslash i},u_{j,\backslash i})\frac{x_j^\top H_{\backslash i}^{-2}G_{\backslash i}H_{0,\backslash i}^{-1}x_jH_{0,\backslash i}^{-1}x_j}{n^2}\right\|^2\right]$$

$$\leq \|\partial_u\ell_0\|_\infty^2\frac{1}{\lambda^4\lambda_0^4}\|\partial_r^2\partial_u\ell\|_\infty^2\|\partial_r\partial_u\ell\|_\infty^2 O\left(\frac{\mathrm{polylog}(n)}{n^2}\right).$$

Returning to (12) finally allows one to reach the bound

$$\mathbb{E}\left[\left(\frac{1}{n}\sum_{j\neq i}\left(\partial_r^3\ell_j(r_{j,\backslash i},u_{j,\backslash i})(\tilde{r}_{j,i}-r_{j,\backslash i})+\partial_r^2\partial_u\ell_j(r_{j,\backslash i},u_{j,\backslash i})(\tilde{u}_{j,i}-u_{j,\backslash i})\right)\frac{x_j^\top H_{\backslash i}^{-2}G_{\backslash i}H_{0,\backslash i}^{-1}x_j}{n}\right)^2\right]=O\left(\frac{\mathrm{polylog}(n)}{n^2}\right)$$

Thus, putting all the intermediary results together,

$$\frac{3}{n^2}\mathbb{E}\left[\mathrm{tr}\left[(H_{\backslash i}^{-1}-\tilde{H}_i^{-1})\tilde{G}_i\tilde{H}_{0,i}^{-1}\right]^2\right]=O_{L_k}\left(\frac{\mathrm{polylog}(n)}{n^2}\right).$$

This takes care of the first term in (15). The remaining two terms can be controlled as

$$\frac{3}{n}\mathbb{E}\left[\mathrm{tr}\left[H_{\backslash i}^{-1}(G_{\backslash i}-\tilde{G}_i)\tilde{H}_{0,i}^{-1}\right]\right]^2=O_{L_k}\left(\frac{\mathrm{polylog}(n)}{n^2}\right),$$

$$\frac{3}{n}\mathbb{E}\left[\mathrm{tr}\left[H_{\backslash i}^{-1}G_{\backslash i}(H_{0,\backslash i}^{-1}-\tilde{H}_{0,i}^{-1})\right]^2\right]=O_{L_k}\left(\frac{\mathrm{polylog}(n)}{n^2}\right),$$

using an identical proof strategy. Returning to (15), one thus has the bound

$$\mathbb{E}\left[(\mathfrak{X}_{\backslash i}-\tilde{\mathfrak{X}}_i)^2\right]=O\left(\frac{\mathrm{polylog}(n)}{n^2}\right).$$

The Efron-Stein lemma finally guarantees that

$$\mathrm{Var}\left[\mathfrak{X}\right]=O\left(\frac{\mathrm{polylog}(n)}{n}\right),$$

which concludes the proof. $\qquad\square$

Before concluding this subsection, we state one last concentration result, regarding the quantity $\omega_i={}^1\!/nx_i^\top G_{\backslash i}H_{0,\backslash i}^{-1}x_i$ and $\mathfrak{W}={}^1\!/n\,\mathrm{Tr}\left[GH_0^{-1}\right]$. Although this quantity does not directly appear in the final characterization, it will prove instrumental in establishing a characterization for $\mathbb{E}\left[\mathfrak{X}\right]$ in the later subsection A.5.

**Lemma A.23** ($\mathfrak{W}$ concentrates). *We remind $\omega_i={}^1\!/nx_i^\top G_{\backslash i}H_{0,\backslash i}^{-1}x_i$, $\mathfrak{W}=\frac{1}{n}\mathrm{tr}\left[GH_0^{-1}\right]$. $\omega_i$ admits an equivalent*

$$\sup_{i\in[n]}|\omega_i-\mathfrak{W}|=O_{L_k}\left(\frac{\mathrm{polylog}(n)}{\lambda\sqrt{n}}\right).$$

*Furthermore, $\mathfrak{W}$ concentrates, with*

$$\mathrm{Var}\left[\mathfrak{W}\right]=O\left(\frac{\mathrm{polylog}(n)}{n}\right).$$

*Proof.* The proof is identical to that of Lemma A.21. One can check that in all steps, the matrix $H^{-1}$ in Lemma A.21 can be replaced by the identity, with all steps remaining valid. $\qquad\square$

## A.4. Distribution of the residuals

We showed in Subsection A.3 that the summary statistics $\mathfrak{m}_c = \langle \hat{w}, \mu_c \rangle$, $\mathfrak{h} = \langle \hat{w}, \beta \rangle$, $\mathfrak{t} = \langle \hat{w}, \hat{w}_0 \rangle$, $\mathfrak{q} = \|\hat{w}\|^2$ and $, \mathfrak{V} = 1/n \operatorname{tr}\left[H_{\backslash i}^{-1}\right]$, $\mathfrak{X} = 1/n \operatorname{tr}\left[H^{-1} G H_0^{-1}\right]$ concentrate in $L_2$, making it hence sufficient to evaluate their expectations, which we denote $m_c, \theta, t, q, V, \chi$. To do so, it remains to ascertain the asymptotic distribution of the residuals $r_i$, $r_{i,\backslash i}$ and $\tilde{r}_{i,i}$, which is the objective of the present subsection. We start with the leave-one-out residuals $r_{i,\backslash i}$. It is possible to show, such as in (Barnfield et al., 2025), that the leave-on-out residual are Gaussian up to vanishing corrections in $L_2$. For later convenience, we rather state the convergence of the expectations of certain functions, which we define in the following, to the corresponding Gaussian averages.

**Definition A.24.** Let us define $\mathcal{F}$ the class of sequences of functions $\varphi : \mathbb{R}^{|\mathcal{V}|+3} \times \mathcal{V} \times \mathcal{E} \to \mathbb{R}$ [2] which are Lipschitz-continuous in its first two variables up to linear functions, and polynomially bounded in the others. Namely, $\varphi \in \mathcal{F}$ if (a) on the one hand there exists $\mathsf{L}_\varphi = O(\operatorname{polylog}(n))$ such that for any $R, R' \in \mathbb{R}^{|\mathcal{V}|+3}$ that coincide everywhere except the first two entries ($R_l = R'_l$ for all $l \geq 3$), and any $\epsilon \in \mathcal{E}, c \in \mathcal{V}$,

$$|\varphi(R, c, \epsilon) - \varphi(R', c, \epsilon)| \leq \mathsf{L}_\varphi(1 + |R_1| + |R'_1| + |R_2| + |R'_2|)\sqrt{(R_1 - R'_1)^2 + (R_2 - R'_2)^2}, \tag{18}$$

and (b) on the other hand there exists a polynomial $Q_\varphi$ in $|\mathcal{V}| + 1$ variables, with $O(\operatorname{polylog}(n))$ coefficients and $O(1)$ degree, such that for all $R \in \mathbb{R}^{|\mathcal{V}|+3}$ with vanishing first two entries $R_1 = R_2 = 0$

$$|\varphi(R, c, \epsilon)| \leq \left|Q_\varphi\left(R_{[3:|\mathcal{V}|+3]}, \epsilon\right)\right|,$$

where $R_{[3:|\mathcal{V}|+3]} = (R_i)_{i \geq 3}$ denotes the entries of $R$ starting from the third.

**Lemma A.25** (Distribution of the leave-one-out residuals). *For any $i \in [n]$, introduce $R_{i,\backslash i} \in \mathbb{R}^{|\mathcal{V}|+3}$ as*

$$R_{i,\backslash i} = \begin{bmatrix} r_{i,\backslash i} \\ u_{i,\backslash i} \\ \langle x_i, \beta \rangle \\ \hline \mathcal{M} x_i \end{bmatrix}.$$

*Then for any function $\varphi : \mathbb{R}^{|\mathcal{V}|+3} \in \mathcal{F}$, one has the convergence*

$$\left| \mathbb{E}\left[\varphi\left(R_{i,\backslash i}, c_i, \epsilon_i\right)\right] - \mathbb{E}\left[\varphi\left(\Psi_c + \Phi^{\frac{1}{2}} g, c, \epsilon\right)\right] \right| = O\left(\frac{\operatorname{polylog}(n)}{n^{\frac{1}{4}}}\right),$$

*with $g \sim \mathcal{N}(0, I_{|\mathcal{V}|+3})$. The first average bears over the training set; the second over $g, c, \epsilon$. We defined the mean and covariance $\Psi \in \mathbb{R}^{|\mathcal{V}|+3}, \Phi \in R^{(|\mathcal{V}|+3) \times (|\mathcal{V}|+3)}$ as*

$$\Psi_c = \begin{bmatrix} m_c \\ m_{0,c} \\ \nu_c \\ \hline \rho_c \end{bmatrix}, \qquad \Phi = \begin{bmatrix} q & t & \theta & m \\ t & q_0 & \theta_0 & m_0 \\ \theta & \theta_0 & \varrho & \nu \\ \hline m & m_0 & \nu & \rho \end{bmatrix},$$

*where we denoted $\rho_c$ the $c$−th row of $\rho$. We recall that the definitions of $\rho \in \mathbb{R}^{|\mathcal{V}| \times |\mathcal{V}|}$, $\nu \in \mathbb{R}^{|\mathcal{V}|}$ can be found in Assumption A.4.*

---

[2] For conciseness of notations, we keep the dependence of $\varphi$ on $n$ implicit.

*Proof.* Let us introduce the stochastic mean and covariance

$$\mathfrak{M}_{\backslash i,c} = \left[ \begin{array}{c} \mathfrak{m}_{c,\backslash i} \\ \mathfrak{m}_{0,c,\backslash i} \\ \hline \nu_c \\ \hline \rho_c \end{array} \right], \qquad \mathfrak{S}_{\backslash i} = \left[ \begin{array}{ccc|c} \mathfrak{q}_{\backslash i} & \mathfrak{t}_{\backslash i} & \mathfrak{h}_{\backslash i} & \mathfrak{m}_{\backslash i} \\ \mathfrak{t}_{\backslash i} & \mathfrak{q}_{0,\backslash i} & \mathfrak{h}_{0,\backslash i} & \mathfrak{m}_{0,\backslash i} \\ \mathfrak{h}_{\backslash i} & \mathfrak{h}_{0,\backslash i} & \varrho & \nu \\ \hline \mathfrak{m}_{\backslash i} & \mathfrak{m}_{0,\backslash i} & \nu & \rho \end{array} \right].$$

We used the shorthands $\mathfrak{m}_{c,\backslash i} = \langle \hat{w}_{\backslash i}, \mu_c \rangle, \mathfrak{h}_{\backslash i} = \langle \hat{w}_{\backslash i}, \beta \rangle, \mathfrak{t}_{\backslash i} = \langle \hat{w}_{\backslash i}, \hat{w}_{0,\backslash i} \rangle, \mathfrak{q}_{\backslash i} = \|\hat{w}_{\backslash i}\|^2$ and $\mathfrak{m}_{0,c,\backslash i} = \langle \hat{w}_{0,\backslash i}, \mu_c \rangle, \mathfrak{h}_{0,\backslash i} = \langle \hat{w}_{0,\backslash i}, \beta \rangle, \mathfrak{t}_{\backslash i} = \langle \hat{w}_{0,\backslash i}, \hat{w}_{0,\backslash i} \rangle, \mathfrak{q}_{0,\backslash i} = \|\hat{w}_{0,\backslash i}\|^2$. We start from the identity

$$\left| \mathbb{E}\left[ \varphi\left( R_{i,\backslash i}, c_i, \epsilon_i \right) \right] - \mathbb{E}\left[ \varphi\left( \Psi_c + \Phi^{\frac{1}{2}} g, c, \epsilon \right) \right] \right| = \left| \mathbb{E}\left[ \varphi\left( \mathfrak{M}_{\backslash i,c} + \mathfrak{S}_{\backslash i}^{\frac{1}{2}} g, c_i, \epsilon_i \right) \right] - \mathbb{E}\left[ \varphi\left( \Psi_c + \Phi^{\frac{1}{2}} g, c, \epsilon \right) \right] \right|, \quad (19)$$

which follows from the independence of $z_i$ from $\hat{w}_i$. To control the right-hand side, we first show that the leave-one-out statistics $\mathfrak{m}_{c,\backslash i}, \mathfrak{t}_{\backslash i}, \mathfrak{q}_{\backslash i}, \mathfrak{m}_{0,c,\backslash i}, \mathfrak{h}_{0,\backslash i}, \mathfrak{t}_{\backslash i}, , \mathfrak{q}_{0,\backslash i}$ are close to the full statistics $\mathfrak{m}_c, \mathfrak{t}, \mathfrak{q}, \mathfrak{m}_{0,c}, \mathfrak{h}_0, \mathfrak{t}, , \mathfrak{q}_0$. We have shown in the proof of Lemma A.20 that

$$\mathfrak{q}_{\backslash i} = \mathfrak{q} + O_{L_2}\left( \frac{\text{polylog}(n)}{n} \right) = q + O_{L_2}\left( \frac{\text{polylog}(n)}{n} \right) + \sqrt{O_{L_2}\left( \frac{\text{polylog}(n)}{n} \right)}$$

$$= q + O_{L_2}\left( \frac{\text{polylog}(n)}{\sqrt{n}} \right),$$

using the concentration result of Lemma A.20 in the last step. By the same token,

$$\mathfrak{t}_{\backslash i} = \mathfrak{t} + \langle \hat{w}_{\backslash i} - \hat{w}, \hat{w}_0 \rangle + \langle \hat{w}, \hat{w}_{0,\backslash i} - \hat{w}_0 \rangle + \langle \hat{w}_{\backslash i} - \hat{w}, \hat{w}_{0,\backslash i} - \hat{w}_0 \rangle$$

$$= \mathfrak{t} + O_{L_k}\left( \frac{\text{polylog}(n)}{\sqrt{n}} \right)$$

$$= t + O_{L_2}\left( \frac{\text{polylog}(n)}{\sqrt{n}} \right).$$

We used Proposition A.5 and Proposition A.14 in the first step, and Lemma A.19 in the last. Similarly,

$$\mathfrak{h}_{\backslash i} = \mathfrak{h} + O_{L_k}\left( \frac{\text{polylog}(n)}{\sqrt{n}} \right) = \theta + O_{L_2}\left( \frac{\text{polylog}(n)}{\sqrt{n}} \right),$$

and for any $c \in \mathcal{V}$,

$$\mathfrak{m}_{c,\backslash i} = \mathfrak{m}_c + O_{L_k}\left( \frac{\text{polylog}(n)}{\sqrt{n}} \right) = m_c + O_{L_2}\left( \frac{\text{polylog}(n)}{\sqrt{n}} \right).$$

The same controls hold for the base statistics $\mathfrak{m}_{0,c,\backslash i}, \mathfrak{h}_{0,\backslash i}, \mathfrak{t}_{\backslash i}, , \mathfrak{q}_{0,\backslash i}$. Thus, the differences $\mathfrak{M}_{\backslash i,c} - \Psi_c$ and $\mathfrak{S}_{\backslash i,c_i} - \Phi_{c_i}$ have $O_{L_2}(\text{polylog}(n)/\sqrt{n})$ entries. We are now in a position to bound

$$\left| \mathbb{E}\left[ \varphi\left( \mathfrak{M}_{\backslash i,c} + \mathfrak{S}_{\backslash i}^{\frac{1}{2}} g, c_i, \epsilon_i \right) \right] - \mathbb{E}\left[ \varphi\left( \Psi_c + \Phi^{\frac{1}{2}} g, c, \epsilon \right) \right] \right|$$

A fine analysis of the different arguments is required, as $\varphi$ is only Lipschitz in its first two variables. Let us denote $g^{\mathfrak{S}} = \mathfrak{S}_{\backslash i}^{\frac{1}{2}} g, g^{\Phi} = \Phi^{\frac{1}{2}} g$ the anisotropic Gaussian variables. Because $\mathfrak{S}_{\backslash i}, \Phi$ coincide in their lower right $(|\mathcal{V}|+1) \times (|\mathcal{V}|+1)$ minor $A$, the entries $g_{[3:]}^{\mathfrak{S}}, g_{[3:]}^{\Phi}$ from the third on are identically distributed. Thus, making explicit the Gaussian conditioning, one can write

$$\mathbb{E}\left[ \varphi\left( \mathfrak{M}_{\backslash i,c} + \mathfrak{S}_{\backslash i}^{\frac{1}{2}} g, c_i, \epsilon_i \right) \right] - \mathbb{E}\left[ \varphi\left( \Psi_c + \Phi^{\frac{1}{2}} g, c, \epsilon \right) \right]$$

$$= \mathbb{E}_{X_{\backslash i}} \mathbb{E}_{g_A}\left[ \mathbb{E}_{g_2}\left[ \varphi\left( \mathfrak{M}_{\backslash i,c} + \left[ \frac{\mathfrak{B} g_A + \mathfrak{C}^{\frac{1}{2}} g_2}{g_A} \right], c_i, \epsilon_i \right) \right] - \mathbb{E}_{g_2}\left[ \varphi\left( \Psi_c + \left[ \frac{B g_A + C^{\frac{1}{2}} g_2}{g_A} \right], \epsilon, c \right) \right] \right],$$

where $g_A \sim \mathcal{N}(0, A)$ is a $(|\mathcal{V}| + 1)-$ dimensional Gaussian vector, and $g_2 \sim \mathcal{N}(0, I_2)$ is an independent variable. We denoted

$$A = \Phi_{[3:],[3:]} = \mathfrak{S}_{[3:],[3:]}, \qquad B = \Phi_{[1:2],[3:]}A^{-1}, \qquad C = \Phi_{[1:2],[1:2]} - BAB^\top,$$

and

$$\mathfrak{B} = \mathfrak{S}_{[1:2],[3:]}A^{-1}, \qquad\qquad \mathfrak{C} = \mathfrak{S}_{[1:2],[1:2]} - \mathfrak{B}A\mathfrak{B}^\top$$

Noting that $\mathfrak{M}_{\backslash i,c,[3:]} = \Psi_{c,[3:]}$, we thus have that the arguments of the two $\varphi$ functions coincide on all entries except the first two. Leveraging the Lipschitzness of $\varphi$ in its first two variables, and further using the independence of $\mathfrak{M}_{\backslash i,c}, \mathfrak{S}_{\backslash i}$ with respect to $\epsilon_i, c_i$,

$$\left| \mathbb{E}\left[ \varphi\left( \mathfrak{M}_{\backslash i,c} + \mathfrak{S}_{\backslash i}^{\frac{1}{2}}g, c_i, \epsilon_i \right) \right] - \mathbb{E}\left[ \varphi\left( \Psi_c + \Phi^{\frac{1}{2}}g, c, \epsilon \right) \right] \right|$$

$$\leq \mathsf{L}_\varphi \mathbb{E}_{X_{\backslash i}} \mathbb{E}_{g_A} \mathbb{E}_{g_2}\left[ \left( 1 + \left\| \mathfrak{M}_{\backslash i,c,[1:2]} \right\|_1 + \left\| \Psi_{c,[1:2]} \right\|_1 + \left\| \mathfrak{B}g_A + \mathfrak{C}^{\frac{1}{2}}g_2 \right\|_1 + \left\| Bg_A + C^{\frac{1}{2}}g_2 \right\|_1 \right) \right.$$

$$\left. \left( \left\| \mathfrak{M}_{\backslash i,c,[1:2]} - \Psi_{c,[1:2]} \right\| + \left\| (\mathfrak{B} - B)g_A + (\mathfrak{C}^{\frac{1}{2}} - C^{\frac{1}{2}})g_2 \right\| \right) \right]$$

$$\leq \mathsf{L}_\varphi \mathbb{E}\left[ \left( 1 + \left\| \mathfrak{M}_{\backslash i,c,[1:2]} \right\|_1 + \left\| \Psi_{c,[1:2]} \right\|_1 + \left\| \mathfrak{B}g_A + \mathfrak{C}^{\frac{1}{2}}g_2 \right\|_1 + \left\| Bg_A + C^{\frac{1}{2}}g_2 \right\|_1 \right)^2 \right]^{\frac{1}{2}}$$

$$\mathbb{E}\left[ \left( \left\| \mathfrak{M}_{\backslash i,c,[1:2]} - \Psi_{c,[1:2]} \right\| + \left\| (\mathfrak{B} - B)g_A + (\mathfrak{C}^{\frac{1}{2}} - C^{\frac{1}{2}})g_2 \right\| \right)^2 \right]^{\frac{1}{2}}$$

But

$$\mathbb{E}\left[ \left( \left\| \mathfrak{M}_{\backslash i,c,[1:2]} - \Psi_{c,[1:2]} \right\| + \left\| (\mathfrak{B} - B)g_A + (\mathfrak{C}^{\frac{1}{2}} - C^{\frac{1}{2}})g_2 \right\| \right)^2 \right]$$

$$\leq 2\mathsf{L}_\varphi \mathbb{E}\left[ \left\| \mathfrak{M}_{\backslash i,c} - \Psi_{c_i} \right\|^2 \right] + 2\mathsf{L}_\varphi \mathbb{E}\left[ \operatorname{tr}\left[ (\mathfrak{B} - B)A(\mathfrak{B} - B)^\top \right] + \operatorname{tr}\left[ (\mathfrak{C}^{\frac{1}{2}} - C^{\frac{1}{2}})^2 \right] \right]$$

From the observation above,

$$\mathbb{E}\left[ \left\| \mathfrak{M}_{\backslash i,c} - \Psi_c \right\|^2 \right] = O\left( \frac{\operatorname{polylog}(n)}{n} \right).$$

Turning to the second term,

$$\mathbb{E}\left[ \operatorname{tr}\left[ (\mathfrak{B} - B)A(\mathfrak{B} - B)^\top \right] \right] \leq 2\mathbb{E}\left[ \operatorname{tr}\left[ (\mathfrak{S}_{[1:2],[3:]} - \Phi_{[1:2],[3:]})A^{-1}(\mathfrak{S}_{[1:2],[3:]} - \Phi_{[1:2],[3:]})^\top \right] \right]$$

$$\leq 4\|A^{-1}\|\mathbb{E}\left[ \left\| \mathfrak{S}_{[1:2],[3:]} - \Phi_{[1:2],[3:]} \right\|^2 \right] = O\left( \frac{\operatorname{polylog}(n)}{n} \right)$$

We can bound the third term using the Powers-Størmer inequality (Powers & Størmer, 1970)

$$\operatorname{tr}\left[ \left( \mathfrak{C}^{\frac{1}{2}} - C^{\frac{1}{2}} \right)^2 \right] \leq \|\mathfrak{C} - C\|_*$$

$$\leq \sqrt{2}\left( \left\| \Phi_{[1:2],[1:2]} - \mathfrak{S}_{[1:2],[1:2]} \right\|_F + \left\| \Phi_{[1:2],[3:]}A^{-1}\Phi_{[1:2],[3:]}^\top - \mathfrak{S}_{[1:2],[3:]}A^{-1}\mathfrak{S}_{[1:2],[3:]}^\top \right\|_F \right)$$

$$\leq \sqrt{2}\left( \left\| \Phi_{[1:2],[1:2]} - \mathfrak{S}_{[1:2],[1:2]} \right\|_F + 2\left\| (\Phi_{[1:2],[3:]} - \mathfrak{S}_{[1:2],[3:]})A^{-1}\Phi_{[1:2],[3:]}^\top \right\|_F \right)$$

$$+ \sqrt{2}\left\| (\Phi_{[1:2],[3:]} - \mathfrak{S}_{[1:2],[3:]})A^{-1}(\Phi_{[1:2],[3:]} - \mathfrak{S}_{[1:2],[3:]}) \right\|_F$$

But

$$\mathbb{E}\left[ \left\| \Phi_{[1:2],[1:2]} - \mathfrak{S}_{[1:2],[1:2]} \right\|_F \right] \leq \mathbb{E}\left[ \left\| \Phi_{[1:2],[1:2]} - \mathfrak{S}_{[1:2],[1:2]} \right\|_F^2 \right]^{\frac{1}{2}} = O\left( \frac{\operatorname{polylog}(n)}{\sqrt{n}} \right)$$

and

$$\mathbb{E}\left[\left\|(\Phi_{[1:2],[3:]} - \mathfrak{S}_{[1:2],[3:]})A^{-1}\Phi_{[1:2],[3:]}^\top\right\|_F\right] \le \left\|\Phi_{[1:2],[3:]}\right\|_F 2\|A^{-1}\|\mathbb{E}\left[\left\|\Phi_{[1:2],[3:]} - \mathfrak{S}_{[1:2],[3:]}\right\|_F\right]$$
$$= O\left(\frac{\text{polylog}(n)}{\sqrt{n}}\right),$$

using Lemma A.38 and Lemma A.37 to bound $\left\|\Phi_{[1:2],[3:]}\right\|_F$. since $\mathfrak{S}_{\backslash i, c_i} - \Phi_{c_i}$ has $O_{L_2}(\text{polylog}(n)/\sqrt{n})$ entries. Finally,

$$\mathbb{E}\left[\left\|(\Phi_{[1:2],[3:]} - \mathfrak{S}_{[1:2],[3:]})A^{-1}\Phi_{[1:2],[3:]} - \mathfrak{S}_{[1:2],[3:]})^\top\right\|_F\right] \le 2\|A^{-1}\|\mathbb{E}\left[\left\|\Phi_{[1:2],[3:]} - \mathfrak{S}_{[1:2],[3:]}\right\|_F^2\right]$$
$$= O\left(\frac{\text{polylog}(n)}{n}\right),$$

Thus,

$$\mathbb{E}\left[\left(\left\|\mathfrak{M}_{\backslash i,c,[1:2]} - \Psi_{c,[1:2]}\right\| + \left\|(\mathfrak{B} - B)g_A + (\mathfrak{C}^{\frac{1}{2}} - C^{\frac{1}{2}})g_2\right\|\right)^2\right]^{\frac{1}{2}} = O\left(\frac{\text{polylog}(n)}{n^{\frac{1}{4}}}\right).$$

It now remains to bound

$$\mathbb{E}\left[\left(1 + \left\|\mathfrak{M}_{\backslash i,c,[1:2]}\right\|_1 + \left\|\Psi_{c,[1:2]}\right\|_1 + \left\|\mathfrak{B}g_A + \mathfrak{C}^{\frac{1}{2}}g_2\right\|_1 + \left\|Bg_A + C^{\frac{1}{2}}g_2\right\|_1\right)^2\right]$$

$$\le 4\mathbb{E}\left[1 + \left\|\mathfrak{M}_{\backslash i,c,[1:2]}\right\|^2 + \left\|\Psi_{c,[1:2]}\right\|^2 + \left\|g_{[1:2]}^{\mathfrak{S}}\right\|^2 + \left\|g_{[1:2]}^{\Phi}\right\|^2\right]$$

$$\le 4 + O\left(\frac{\text{polylog}(n)}{n}\right) + 12\left\|\Psi_{c,[1:2]}\right\|^2 + 4\mathbb{E}\left[\left\|g_{[1:2]}^{\mathfrak{S}}\right\|^2 + \left\|g_{[1:2]}^{\Phi}\right\|^2\right]$$

$$\le 4 + O\left(\frac{\text{polylog}(n)}{n}\right) + 12(m_c^2 + m_{0,c}^2) + 8\mathbb{E}\left[\text{tr}\left[\Phi_{[:2,:2]}\right] + O_{L_2}\left(\frac{\text{polylog}(n)}{\sqrt{n}}\right)\right]$$

$$= 4 + O\left(\frac{\text{polylog}(n)}{\sqrt{n}}\right) + 12(m_c^2 + m_{0,c}^2) + 8(q + q_0).$$

It follows directly from Lemma A.38 and Lemma A.37 and the Cauchy-Schwartz inequality that the sequences $m_c, m_{0,c}, q, q_0$ are bounded $O(\text{polylog}(n))$. Thus, returning to (19),

$$\left|\mathbb{E}\left[\varphi\left(\mathfrak{M}_{\backslash i,c} + \mathfrak{S}_{\backslash i}^{\frac{1}{2}}g, c_i, \epsilon_i\right)\right] - \mathbb{E}\left[\varphi\left(\Psi_{c_i} + \Phi^{\frac{1}{2}}g, c, \epsilon\right)\right]\right| = O\left(\frac{\text{polylog}(n)}{n^{\frac{1}{4}}}\right),$$

which concludes the proof. $\square$

We are now in a position to prove a similar distributional statement for the surrogate residuals

$$\tilde{r}_{i,i} = \text{prox}_{\gamma_i \ell_i(\cdot, \tilde{u}_{i,i})}\left(r_{i,\backslash i} + \partial_u \ell_{0,i}(\tilde{u}_{i,i})v_i\right),$$
$$\tilde{u}_{i,i} = \text{prox}_{\gamma_{0,i}\ell_{0,i}(\cdot)}(u_{i,\backslash i}).$$

The goal is to use the convergence result of Lemma A.25, leveraging the regularity of the proximal operators, and the concentration results of Lemmas A.21 and A.22.

**Lemma A.26** (Distribution of the surrogate residuals). *For any $i \in [n]$, introduce $\tilde{R}_{i,\backslash i} \in \mathbb{R}^{|\mathcal{V}|+3}$ as*

$$\tilde{R}_{i,\backslash i} = \begin{bmatrix} \tilde{r}_{i,\backslash i} \\ \tilde{u}_{i,\backslash i} \\ \langle x_i, \beta \rangle \\ \hline \mathcal{M}x_i \end{bmatrix}.$$

*Then for any function $\varphi : \mathbb{R}^{|\mathcal{V}|+3} \in \mathcal{F}$, one has the convergence*

$$\left| \mathbb{E}\left[ \varphi\left( \tilde{R}_{i,\backslash i}, c_i, \epsilon_i \right) \right] - \mathbb{E}\left[ \varphi\left( G, c, \epsilon \right) \right] \right| = O\left( \frac{\text{polylog}(n)}{n^{\frac{1}{4}}} \right),$$

*where*

$$G = \left[ \begin{array}{c} \text{prox}_{V\ell(\cdot,\text{prox}_{V_0\ell_0(\cdot,g_3,c,\epsilon)}(g_2),g_3,c,\epsilon)}\left( g_1 + \chi\partial_u\ell_0(\text{prox}_{V_0\ell_0(\cdot,g_3,c,\epsilon)}(g_2),g_3,c,\epsilon) \right) \\ \text{prox}_{V_0\ell_0(\cdot,g_3,c,\epsilon)}(g_2) \\ g_3 \\ \hline g_{[4:|\mathcal{V}|+3]} \end{array} \right].$$

*where $g_1, g_2, g_3, g_{[4:|\mathcal{V}|+3]} = (g_i)_{i \geq 4}$ designate the entries of a Gaussian vector $g \in \mathbb{R}^{|\mathcal{V}|+3}$ with law*

$$g \sim \mathcal{N}\left( \Psi_c, \Phi \right),$$

*conditionally on $c$. $\Psi_c, \Phi$ were defined in Lemma A.25.*

*Proof.* To alleviate the notations, we will in this proof use the shorthands

$$f_{v_1,v_2,v_3}(r,u) = \text{prox}_{v_1\ell(\cdot,\text{prox}_{v_2\ell_0(\cdot,\langle x_i,\beta\rangle,c_i,\epsilon_i)}(u),\langle x_i,\beta\rangle,c_i,\epsilon_i)}\left( r + v_3\partial_u\ell_0(\text{prox}_{v_2\ell_0(\cdot,\langle x_i,\beta\rangle,c_i,\epsilon_i)}(u), \langle x_i,\beta\rangle, c_i, \epsilon_i) \right),$$

$$h_{v_2}(u) = \text{prox}_{v_2\ell_0(\cdot,\langle x_i,\beta\rangle,c_i,\epsilon_i)}(u).$$

We can decompose the objective as

$$\left| \mathbb{E}\left[ \varphi\left( \tilde{R}_{i,\backslash i}, c_i, \epsilon_i \right) \right] - \mathbb{E}\left[ \varphi\left( G, c, \epsilon \right) \right] \right|$$

$$= \left| \mathbb{E}\left[ \varphi\left( \left[ \begin{array}{c} f_{\gamma_i,\gamma_{0,i},\upsilon_i}(r_{i,\backslash i}, u_{i,\backslash i}) \\ h_{\gamma_{0,i}}(u_{i,\backslash i}) \\ \langle x_i, \beta \rangle \\ \hline \mathcal{M}x_i \end{array} \right], c_i, \epsilon_i \right) \right] - \mathbb{E}\left[ \varphi\left( G, c, \epsilon \right) \right] \right|$$

$$\leq \left| \mathbb{E}\left[ \varphi\left( \left[ \begin{array}{c} f_{\gamma_i,\gamma_{0,i},\upsilon_i}(r_{i,\backslash i}, u_{i,\backslash i}) \\ h_{\gamma_{0,i}}(u_{i,\backslash i}) \\ \langle x_i, \beta \rangle \\ \hline \mathcal{M}x_i \end{array} \right], c_i, \epsilon_i \right) \right] - \mathbb{E}\left[ \varphi\left( \left[ \begin{array}{c} f_{V,V_0,\chi}(r_{i,\backslash i}, u_{i,\backslash i}) \\ h_{V_0}(u_{i,\backslash i}) \\ \langle x_i, \beta \rangle \\ \hline \mathcal{M}x_i \end{array} \right], c_i, \epsilon_i \right) \right] \right|$$

$$+ \left| \mathbb{E}\left[ \varphi\left( \left[ \begin{array}{c} f_{V,V_0,\chi}(r_{i,\backslash i}, u_{i,\backslash i}) \\ h_{V_0}(u_{i,\backslash i}) \\ \langle x_i, \beta \rangle \\ \hline \mathcal{M}x_i \end{array} \right], c_i, \epsilon_i \right) \right] - \mathbb{E}\left[ \varphi\left( G, c, \epsilon \right) \right] \right|. \tag{20}$$

In words, we first replace the stochastic statistics $\gamma_i, \gamma_{0,i}, \upsilon_i$ by their deterministic equivalents $V, V_0, \chi$. The associated error is controlled by the first term of the right-hand side. Then, the second term bounds the approximation when replacing the expectations over the leave-one-out residuals $r_{i,\backslash i}, u_{i,\backslash i}$ by the associated Gaussian averages. We first examine the latter. One can guarantee that the associated error term vanishes as $n \to \infty$ from Lemma A.25, provided one establishes

beforehand that the coumpound function

$$\tilde{\varphi} : \begin{bmatrix} r \\ u \\ \dfrac{s}{\phantom{.}} \\ \omega \end{bmatrix}, c, \epsilon \to \varphi \left( \begin{bmatrix} f_{V,V_0,\chi}(r,u) \\ h_{V_0}(u) \\ \dfrac{s}{\phantom{.}} \\ \omega \end{bmatrix}, c, \epsilon \right).$$

belongs to $\mathcal{F}$, namely is pseudo-Lipschitz in its first two arguments and admits a polynomial majorant in the others. For that purpose, it is sufficient to show that $h_{V_0}$ and $f_{V,V_0,\chi}$ are pseudo-Lipschitz. The first claim follows from the contractivity of the proximal operator. Examining the second claim, let us take $(u_1, r_1), (u_2, r_2) \in \mathbb{R}^2$ and bound

$$|f_{V,V_0,\chi}(r_1, u_1) - f_{V,V_0,\chi}(r_2, u_2)| \le |f_{V,V_0,\chi}(r_1, u_1) - f_{V,V_0,\chi}(r_2, u_1)| + |f_{V,V_0,\chi}(r_2, u_1) - f_{V,V_0,\chi}(r_2, u_2)|.$$

It follows from the contractivity of the proximal operator that

$$|f_{V,V_0,\chi}(r_1, u_1) - f_{V,V_0,\chi}(r_2, u_1)| \le |r_1 - r_2|.$$

On the other hand,

$$
\begin{aligned}
&f_{V,V_0,\chi}(r_2, u_1) - f_{V,V_0,\chi}(r_2, u_2) \\
&= \chi \left( \partial_u \ell_0(\mathrm{prox}_{V_0 \ell_0(\cdot, s, c, \epsilon)}(u_1), s, c, \epsilon) - \partial_u \ell_0(\mathrm{prox}_{V_0 \ell_0(\cdot, s, c, \epsilon)}(u_2), s, c, \epsilon) \right) \\
&\quad + V \left[ \partial_r \ell \left( f_{V,V_0,\chi}(r_2, u_2), \mathrm{prox}_{V_0 \ell_0(\cdot, s, c, \epsilon)}(u_2), s, c, \epsilon \right) - \partial_r \ell \left( f_{V,V_0,\chi}(r_2, u_1), \mathrm{prox}_{V_0 \ell_0(\cdot, s, c, \epsilon)}(u_1), s, c, \epsilon \right) \right] \\
&= \chi \left( \partial_u \ell_0(\mathrm{prox}_{V_0 \ell_0(\cdot, s, c, \epsilon)}(u_1), s, c, \epsilon) - \partial_u \ell_0(\mathrm{prox}_{V_0 \ell_0(\cdot, s, c, \epsilon)}(u_2), s, c, \epsilon) \right) \\
&\quad + V \Bigg[ \partial_r^2 \ell (\check{r}, \check{u}, s, c, \epsilon) \left( f_{V,V_0,\chi}(r_2, u_2) - f_{V,V_0,\chi}(r_2, u_1) \right) \\
&\qquad\qquad + \partial_r \partial_u \ell (\check{r}, \check{u}, s, c, \epsilon) \left( \mathrm{prox}_{V_0 \ell_0(\cdot, s, c, \epsilon)}(u_2) - \mathrm{prox}_{V_0 \ell_0(\cdot, s, c, \epsilon)}(u_1) \right) \Bigg],
\end{aligned}
$$

for some $(\check{r}, \check{u})$ lying on the line connecting $f_{V,V_0,\chi}(r_2, u_2), \mathrm{prox}_{V_0 \ell_0(\cdot, y, c_i, \epsilon)}(u_2)$ and $f_{V,V_0,\chi}(r_2, u_1), \mathrm{prox}_{V_0 \ell_0(\cdot, y, c_i, \epsilon)}(u_1)$, leveraging the mean-value theorem. Thus,

$$
\begin{aligned}
|f_{V,V_0,\chi}(r_2, u_1) - f_{V,V_0,\chi}(r_2, u_2)| &\le \frac{1}{1 + V \partial_r^2 \ell (\check{r}, \check{u}, y, c_i, \epsilon)} \left[ \left( |\chi| \|\partial_u^2 \ell_0\|_\infty + V \|\partial_r \partial_u \ell\|_\infty \right) |u_2 - u_1| \right] \\
&\le \left( |\chi| \|\partial_u^2 \ell_0\|_\infty + \frac{1}{\lambda} \|\partial_r \partial_u \ell\|_\infty \right) |u_2 - u_1|,
\end{aligned}
$$

using again the contractivity of the proximal operator and the rough bound $V = \frac{1}{n} \mathbb{E}\left[ \mathrm{tr}\left[ H^{-1} \right] \right] \le \mathbb{E}\left[ \|H^{-1}\| \right] \le \frac{1}{\lambda}$. It now remains to bound $\chi = \frac{1}{n} \mathbb{E}\left[ \mathrm{tr}\left[ H^{-1} G H_0^{-1} \right] \right]$. One has

$$
\begin{aligned}
\chi = \frac{1}{n} \mathbb{E}\left[ \mathrm{tr}\left[ H^{-1} G H_0^{-1} \right] \right] &\le \frac{1}{\lambda \lambda_0} \mathbb{E}\left[ \|G\| \right] \\
&\le \frac{1}{\lambda \lambda_0} \|\partial_r \partial_u \ell\|_\infty \mathbb{E}\left[ \|\hat{\Sigma}\| \right] \le O(\mathrm{polylog}(n))
\end{aligned}
$$

using Lemma A.39. Thus,

$$|f_{V,V_0,\chi}(r_1, u_1) - f_{V,V_0,\chi}(r_2, u_2)| \le O(\mathrm{polylog}(n))(|r_1 - r_2| + |u_1 - u_2|),$$

which shows that $f_{V,V_0,\chi}$ is Lipschitz, and thus so is $\tilde{\varphi}$. To complete the demonstration that $\tilde{\varphi} \in \mathcal{F}$, we now need to construct a polynomial majorant in its remaining variables. We note that

$$
\tilde{\varphi}\left(\begin{bmatrix} 0 \\ 0 \\ s \\ \hline \omega \end{bmatrix}, c, \epsilon\right) = \varphi\left(\begin{bmatrix} f_{V,V_0,\chi}(0,0) \\ h_{V_0}(0) \\ s \\ \hline \omega \end{bmatrix}, c, \epsilon\right) \leq \mathsf{L}_\varphi(|f_{V,V_0,\chi}(0,0)| + |h_{V_0}(0)|) + \left|Q_\varphi\left(\begin{bmatrix} s \\ \hline \omega \end{bmatrix}, \epsilon\right)\right|
$$

But on the one hand

$$
|h_{V_0}(0)| \leq \frac{1}{\lambda_0}\|\partial_u \ell_0\|_\infty,
$$

while on the other hand

$$
|f_{V,V_0,\chi}(0,0)| \leq \chi\|\partial_u \ell_0\|_\infty + \frac{1}{\lambda}\|\partial_r \ell\|_\infty.
$$

One can thus bound the test function as

$$
\tilde{\varphi}\left(\begin{bmatrix} 0 \\ 0 \\ s \\ \hline \omega \end{bmatrix}, c, \epsilon\right) \leq O(\text{polylog}(n)) + \left|Q_\varphi\left(\begin{bmatrix} s \\ \hline \omega \end{bmatrix}, c, \epsilon\right)\right|
$$

$$
\leq 1 + O(\text{polylog}(n)) + Q_\varphi\left(\begin{bmatrix} s \\ \hline \omega \end{bmatrix}, \epsilon\right)^2,
$$

yielding a polynomial majorant (recall that we allow coefficients to be $O(\text{polylog}(n))$). Thus, we have established that $\tilde{\varphi} \in \mathcal{F}$. One can thus apply Lemma A.25, and approximate the expectation over leave-one-out residuals by the corresponding Gaussian expectation

$$
\left|\mathbb{E}\left[\varphi\left(\begin{bmatrix} f_{V,V_0,\chi}(r_{i,\backslash i}, u_{i,\backslash i}) \\ h_{V_0}(u_{i,\backslash i}) \\ \langle x_i, \beta \rangle \\ \hline \mathcal{M}x_i \end{bmatrix}, c_i, \epsilon_i\right)\right] - \mathbb{E}\left[\varphi\left(G, c, \epsilon\right)\right]\right| = O\left(\frac{\text{polylog}(n)}{n^{\frac{1}{4}}}\right).
$$

Returning to (20), we now aim to bound the first term in the decomposition (20). We decompose it into three error terms,

which we sequentially control:

$$\left\|\mathbb{E}\left[\varphi\left(\left[\frac{\begin{array}{c}f_{\gamma_i,\gamma_{0,i},v_i}(r_{i,\backslash i},u_{i,\backslash i})\\ h_{\gamma_{0,i}}(u_{i,\backslash i})\\ \langle x_i,\beta\rangle\end{array}}{\mathcal{M}x_i}\right],c_i,\epsilon_i\right)\right]-\mathbb{E}\left[\varphi\left(\left[\frac{\begin{array}{c}f_{V,V_0,\chi}(r_{i,\backslash i},u_{i,\backslash i})\\ h_{V_0}(u_{i,\backslash i})\\ \langle x_i,\beta\rangle\end{array}}{\mathcal{M}x_i}\right],c_i,\epsilon_i\right)\right]\right\|$$

$$\leq\left\|\mathbb{E}\left[\varphi\left(\left[\frac{\begin{array}{c}f_{\gamma_i,\gamma_{0,i},v_i}(r_{i,\backslash i},u_{i,\backslash i})\\ h_{\gamma_{0,i}}(u_{i,\backslash i})\\ \langle x_i,\beta\rangle\end{array}}{\mathcal{M}x_i}\right],c_i,\epsilon_i\right)\right]-\mathbb{E}\left[\varphi\left(\left[\frac{\begin{array}{c}f_{\gamma_i,\gamma_{0,i},\chi}(r_{i,\backslash i},u_{i,\backslash i})\\ h_{\gamma_{0,i}}(u_{i,\backslash i})\\ \langle x_i,\beta\rangle\end{array}}{\mathcal{M}x_i}\right],c_i,\epsilon_i\right)\right]\right\|$$

$$\left\|\mathbb{E}\left[\varphi\left(\left[\frac{\begin{array}{c}f_{\gamma_i,\gamma_{0,i},\chi}(r_{i,\backslash i},u_{i,\backslash i})\\ h_{\gamma_{0,i}}(u_{i,\backslash i})\\ \langle x_i,\beta\rangle\end{array}}{\mathcal{M}x_i}\right],c_i,\epsilon_i\right)\right]-\mathbb{E}\left[\varphi\left(\left[\frac{\begin{array}{c}f_{V,\gamma_{0,i},\chi}(r_{i,\backslash i},u_{i,\backslash i})\\ h_{\gamma_{0,i}}(u_{i,\backslash i})\\ \langle x_i,\beta\rangle\end{array}}{\mathcal{M}x_i}\right],c_i,\epsilon_i\right)\right]\right\|$$

$$\left\|\mathbb{E}\left[\varphi\left(\left[\frac{\begin{array}{c}f_{V,\gamma_{0,i},\chi}(r_{i,\backslash i},u_{i,\backslash i})\\ h_{\gamma_{0,i}}(u_{i,\backslash i})\\ \langle x_i,\beta\rangle\end{array}}{\mathcal{M}x_i}\right],c_i,\epsilon_i\right)\right]-\mathbb{E}\left[\varphi\left(\left[\frac{\begin{array}{c}f_{V,V_0,\chi}(r_{i,\backslash i},u_{i,\backslash i})\\ h_{V_0}(u_{i,\backslash i})\\ \langle x_i,\beta\rangle\end{array}}{\mathcal{M}x_i}\right],c_i,\epsilon_i\right)\right]\right\|.$$

Because $\varphi$ is $\mathsf{L}_\varphi-$ Lipschitz in its first variable up to linear terms, the first error can be controlled by

$$\mathsf{L}_\varphi\mathbb{E}\left[R(r_{i,\backslash i},u_{i,\backslash i})|f_{\gamma_i,\gamma_{0,i},v_i}(r_{i,\backslash i},u_{i,\backslash i})-f_{\gamma_i,\gamma_{0,i},\chi}(r_{i,\backslash i},u_{i,\backslash i})|\right]\leq\mathsf{L}_\varphi\|\partial_u\ell_0\|_\infty\mathbb{E}\left[|\chi-v_i|^2\right]^{\frac12}\mathbb{E}\left[R(r_{i,\backslash i},u_{i,\backslash i})^2\right]^{\frac12}$$

We used Hölder's inequality, and introduced the linear Lipschitz majorant

$$\begin{aligned}R(r_{i,\backslash i},u_{i,\backslash i})=&1+|f_{\gamma_i,\gamma_{0,i},v_i}(r_{i,\backslash i},u_{i,\backslash i})|+|f_{\gamma_i,\gamma_{0,i},\chi}(r_{i,\backslash i},u_{i,\backslash i})|\\&+|f_{V,\gamma_{0,i},\chi}(r_{i,\backslash i},u_{i,\backslash i})|+|f_{V,V_0,\chi}(r_{i,\backslash i},u_{i,\backslash i})|+|h_{\gamma_{0,i}}(u_{i,\backslash i})|+|h_{V_0}(u_{i,\backslash i})|.\end{aligned}$$

So as not to repeat several times the analysis, we chose to include more terms than required by the property (18), so that the bound we establish for $R(r_{i,\backslash i},u_{i,\backslash i})$ can be reemployed for the other two error terms. Since Lemma A.22 guarantees $\mathbb{E}\left[|\chi-v_i|^2\right]^{\frac12}=O\left(\mathrm{polylog}(n)/\sqrt{n}\right)$, we only need to bound $\mathbb{E}\left[R(r_{i,\backslash i},u_{i,\backslash i})^2\right]$ by $O(\mathrm{polylog}(n))$. We have

$$\begin{aligned}\frac17R(r_{i,\backslash i},u_{i,\backslash i})^2\leq&1+12r_{i,\backslash i}^2+3(v_i^2+3\chi^2)\|\partial_u\ell_0\|_\infty^2+3(2\gamma_i^2+2V^2)\|\partial_r\ell\|_\infty^2+4u_{i,\backslash i}^2+2(\gamma_{0,i}^2+V_0^2)\|\partial_u\ell_0^2\|_\infty^2\\ \leq&1+12(\sup_{i\in[n]}|r_{i,\backslash i}|)^2+12\chi^2\|\partial_u\ell_0\|_\infty^2+12V^2\|\partial_r\ell\|_\infty^2+4(\sup_{i\in[n]}|u_{i,\backslash i}|)^2\\ &+4V_0^2\|\partial_u\ell_0^2\|_\infty^2+O_{L_1}\left(\frac{\mathrm{polylog}(n)}{\sqrt{n}}\right)\end{aligned}$$

Lemma A.42 ensures that $\sup_{i\in[n]}|r_{i,\backslash i}|,\sup_{i\in[n]}|u_{i,\backslash i}|+O_{L_k}(\mathrm{polylog}(n))$. Finally, we established above that $\chi,V,V_0$ are bounded by $O(\mathrm{polylog}(n))$. Thus

$$\mathbb{E}\left[R(r_{i,\backslash i},u_{i,\backslash i})^2\right]^{\frac12}=O(\mathrm{polylog}(n)).$$

Hence

$$\mathsf{L}_\varphi\mathbb{E}\left[R(r_{i,\backslash i},u_{i,\backslash i})|f_{\gamma_i,\gamma_{0,i},v_i}(r_{i,\backslash i},u_{i,\backslash i})-f_{\gamma_i,\gamma_{0,i},\chi}(r_{i,\backslash i},u_{i,\backslash i})|\right]\leq\mathsf{L}_\varphi\|\partial_u\ell_0\|_\infty O\left(\frac{\mathrm{polylog}(n)}{\sqrt{n}}\right).$$

The second error term is similarly controlled by

$$\mathsf{L}_\varphi \mathbb{E}\left[ R(r_{i,\backslash i}, u_{i,\backslash i}) | f_{\gamma_i, \gamma_{0,i}, \chi}(r_{i,\backslash i}, u_{i,\backslash i}) - f_{V, \gamma_{0,i}, \chi}(r_{i,\backslash i}, u_{i,\backslash i}) | \right].$$

Since

$$
\begin{aligned}
&f_{\gamma_i, \gamma_{0,i}, \chi}(r_{i,\backslash i}, u_{i,\backslash i}) - f_{V, \gamma_{0,i}, \chi}(r_{i,\backslash i}, u_{i,\backslash i}) \\
&\quad = -\gamma_i \partial_r \ell \left( f_{\gamma_i, \gamma_{0,i}, \chi}(r_{i,\backslash i}, u_{i,\backslash i}), \mathrm{prox}_{\gamma_{0,i}\ell_0(\cdot, \langle \beta, x_i \rangle, c_i, \epsilon)}(u_{i,\backslash i}), \langle \beta, x_i \rangle, c_i, \epsilon_i \right) \\
&\qquad + V \partial_r \ell \left( f_{V, \gamma_{0,i}, \chi}(r_{i,\backslash i}, u_{i,\backslash i}), \mathrm{prox}_{\gamma_{0,i}\ell_0(\cdot, \langle \beta, x_i \rangle, c_i, \epsilon)}(u_{i,\backslash i}), \langle \beta, x_i \rangle, c_i, \epsilon_i \right) \\
&\quad = -V \partial_r^2 \ell \left( \check{r}, \mathrm{prox}_{\gamma_{0,i}\ell_0(\cdot, \langle \beta, x_i \rangle, c_i, \epsilon)}(u_{i,\backslash i}), \langle \beta, x_i \rangle, c_i, \epsilon_i \right) (f_{\gamma_i, \gamma_{0,i}, \chi}(r_{i,\backslash i}, u_{i,\backslash i}) - f_{V, \gamma_{0,i}, \chi}(r_{i,\backslash i}, u_{i,\backslash i})) \\
&\qquad + (V - \gamma_i) \partial_r \ell \left( f_{\gamma_i, \gamma_{0,i}, \chi}(r_{i,\backslash i}, u_{i,\backslash i}), \mathrm{prox}_{\gamma_{0,i}\ell_0(\cdot, \langle \beta, x_i \rangle, c_i, \epsilon)}(u_{i,\backslash i}), \langle \beta, x_i \rangle, c_i, \epsilon_i \right),
\end{aligned}
$$

for some $\check{r}$ in the unordered interval $(f_{\gamma_i, \gamma_{0,i}, \chi}(r_{i,\backslash i}, u_{i,\backslash i}), f_{V, \gamma_{0,i}, \chi}(r_{i,\backslash i}, u_{i,\backslash i}))$, from the mean value theorem. Thus

$$\left| f_{\gamma_i, \gamma_{0,i}, \chi}(r_{i,\backslash i}, u_{i,\backslash i}) - f_{V, \gamma_{0,i}, \chi}(r_{i,\backslash i}, u_{i,\backslash i}) \right| \leq \left| \frac{(V - \gamma_i) \partial_r \ell \left( f_{\gamma_i, \gamma_{0,i}, \chi}(r_{i,\backslash i}, u_{i,\backslash i}), \mathrm{prox}_{\gamma_{0,i}\ell_0(\cdot, \langle \beta, x_i \rangle, c_i, \epsilon)}(u_{i,\backslash i}), \langle \beta, x_i \rangle, c_i, \epsilon_i \right)}{1 + V \partial_r^2 \ell \left( \check{r}, \mathrm{prox}_{\gamma_{0,i}\ell_0(\cdot, \langle \beta, x_i \rangle, c_i, \epsilon)}(u_{i,\backslash i}), \langle \beta, x_i \rangle, c_i, \epsilon_i \right)} \right|,$$

from which the bound follows:

$$
\begin{aligned}
\mathbb{E}\left[ R(r_{i,\backslash i}, u_{i,\backslash i}) | f_{\gamma_i, \gamma_{0,i}, \chi}(r_{i,\backslash i}, u_{i,\backslash i}) - f_{V, \gamma_{0,i}, \chi}(r_{i,\backslash i}, u_{i,\backslash i}) | \right] &\leq \|\partial_r \ell\|_\infty \mathbb{E}\left[ R(r_{i,\backslash i}, u_{i,\backslash i})^2 \right]^{\frac{1}{2}} \mathbb{E}\left[ |V - \gamma_i|^2 \right]^{\frac{1}{2}} \\
&\leq \|\partial_r \ell\|_\infty O\left( \frac{\mathrm{polylog}(n)}{\sqrt{n}} \right),
\end{aligned}
$$

using Hölder's inequality and the concentration result A.21. The last error term is controlled by

$$\mathsf{L}_\varphi \mathbb{E}\left[ R(r_{i,\backslash i}, u_{i,\backslash i}) | f_{V, \gamma_{0,i}, \chi}(r_{i,\backslash i}, u_{i,\backslash i}) - f_{V, V_0, \chi}(r_{i,\backslash i}, u_{i,\backslash i}) | \right] + \mathsf{L}_\varphi \mathbb{E}\left[ R(r_{i,\backslash i}, u_{i,\backslash i}) | h_{\gamma_{0,i}}(u_{i,\backslash i}) - h_{V_0}(u_{i,\backslash i}) | \right].$$

Using the same reasoning as above, there exists $\check{u} \in (h_{\gamma_{0,i}}(u_{i,\backslash i}), h_{V_0}(u_{i,\backslash i}))$ such that

$$h_{\gamma_{0,i}}(u_{i,\backslash i}) - h_{V_0}(u_{i,\backslash i}) = \frac{1}{1 + V_0 \partial_u^2 \ell_0(\check{u}, \langle \beta, x_i \rangle, c_i, \epsilon_i)} \partial_u \ell_0(h_{\gamma_{0,i}}(u_{i,\backslash i}), \langle \beta, x_i \rangle, c_i, \epsilon_i)(V_0 - \gamma_{0,i}).$$

Using Lemma A.21, it follows that

$$\mathbb{E}\left[ R(r_{i,\backslash i}, u_{i,\backslash i}) | h_{\gamma_{0,i}}(u_{i,\backslash i}) - h_{V_0}(u_{i,\backslash i}) | \right] \leq \|\partial_u \ell\|_\infty O\left( \frac{\mathrm{polylog}(n)}{\sqrt{n}} \right).$$

Finally, control of $\mathbb{E}\left[ |f_{V, \gamma_{0,i}, \chi}(r_{i,\backslash i}, u_{i,\backslash i}) - f_{V, V_0, \chi}(r_{i,\backslash i}, u_{i,\backslash i}) | \right]$ can be obtained as follows:

$$
\begin{aligned}
&f_{V, \gamma_{0,i}, \chi}(r_{i,\backslash i}, u_{i,\backslash i}) - f_{V, V_0, \chi}(r_{i,\backslash i}, u_{i,\backslash i}) \\
&\quad = \chi \left( \partial_u \ell_0(h_{\gamma_{0,i}}(u_{i,\backslash i}), \langle x_i, \beta \rangle, c_i, \epsilon_i) - \partial_u \ell_0(h_{V_0}(u_{i,\backslash i}), \langle x_i, \beta \rangle, c_i, \epsilon_i) \right) \\
&\qquad - V \left( \partial_r \ell \left( f_{V, \gamma_{0,i}, \chi}(r_{i,\backslash i}, u_{i,\backslash i}), h_{\gamma_{0,i}}(u_{i,\backslash i}), \langle \beta, x_i \rangle, c_i, \epsilon_i \right) - \partial_r \ell \left( f_{V, V_0, \chi}(r_{i,\backslash i}, u_{i,\backslash i}), h_{\gamma_{0,i}}(u_{i,\backslash i}), \langle \beta, x_i \rangle, c_i, \epsilon_i \right) \right) \\
&\quad = \chi \partial_u^2 \ell_0(\check{u}, \langle \beta, x_i \rangle, c_i, \epsilon_i) \left( h_{\gamma_{0,i}}(u_{i,\backslash i}) - h_{V_0}(u_{i,\backslash i}) \right) \\
&\qquad - V \partial_r^2 \ell (\check{r}, \hat{u}, \langle \beta, x_i \rangle, c_i, \epsilon_i) \left( f_{V, \gamma_{0,i}, \chi}(r_{i,\backslash i}, u_{i,\backslash i}) - f_{V, V_0, \chi}(r_{i,\backslash i}, u_{i,\backslash i}) \right) \\
&\qquad - V \partial_r \partial_u \ell (\check{r}, \hat{u}, \langle \beta, x_i \rangle, c_i, \epsilon_i) \left( h_{\gamma_{0,i}}(u_{i,\backslash i}) - h_{V_0}(u_{i,\backslash i}) \right),
\end{aligned}
$$

for some $\check{u}, \hat{u} \in (h_{\gamma_{0,i}}(u_{i,\backslash i}), h_{V_0}(u_{i,\backslash i}))$ and $\check{r} \in (f_{V, \gamma_{0,i}, \chi}(r_{i,\backslash i}, u_{i,\backslash i}), f_{V, V_0, \chi}(r_{i,\backslash i}, u_{i,\backslash i}))$, applying the mean value theorem. One can thus express $f_{V, \gamma_{0,i}, \chi}(r_{i,\backslash i}, u_{i,\backslash i}) - f_{V, V_0, \chi}(r_{i,\backslash i}, u_{i,\backslash i})$ as

$$f_{V, \gamma_{0,i}, \chi}(r_{i,\backslash i}, u_{i,\backslash i}) - f_{V, V_0, \chi}(r_{i,\backslash i}, u_{i,\backslash i}) = \frac{\left( \chi \partial_u^2 \ell_0(\check{u}, \langle \beta, x_i \rangle, c_i, \epsilon_i) - V \partial_r \partial_u \ell (\check{r}, \hat{u}, \langle \beta, x_i \rangle, c_i, \epsilon_i) \right) \left( h_{\gamma_{0,i}}(u_{i,\backslash i}) - h_{V_0}(u_{i,\backslash i}) \right)}{1 + V \partial_r^2 \ell (\check{r}, \hat{u}, \langle \beta, x_i \rangle, c_i, \epsilon_i)},$$

from which it follows that

$$\left| f_{V,\gamma_{0,i},\chi}(r_{i,\backslash i}, u_{i,\backslash i}) - f_{V,V_0,\chi}(r_{i,\backslash i}, u_{i,\backslash i}) \right| \leq \left[ \chi \|\partial_u^2 \ell_0\|_\infty + \frac{1}{\lambda} \|\partial_r \partial_u \ell\|_\infty \right] |h_{\gamma_{0,i}}(u_{i,\backslash i}) - h_{V_0}(u_{i,\backslash i})|.$$

Thus, from Hölder's inequality and the previous bound on $|h_{\gamma_{0,i}}(u_{i,\backslash i}) - h_{V_0}(u_{i,\backslash i})|$,

$$\mathbb{E}\left[ R(r_{i,\backslash i}, u_{i,\backslash i}) | f_{V,\gamma_{0,i},\chi}(r_{i,\backslash i}, u_{i,\backslash i}) - f_{V,V_0,\chi}(r_{i,\backslash i}, u_{i,\backslash i})| \right] = O\left( \frac{\text{polylog}(n)}{\sqrt{n}} \right).$$

Putting these three bounds together finally yields

$$\left| \mathbb{E}\left[ \varphi\left( \begin{bmatrix} f_{\gamma_i,\gamma_{0,i},\upsilon_i}(r_{i,\backslash i}, u_{i,\backslash i}) \\ h_{\gamma_{0,i}}(u_{i,\backslash i}) \\ y_i \\ \hline \mathcal{M}x_i \end{bmatrix}, c_i, \epsilon_i \right) \right] - \mathbb{E}\left[ \varphi\left( \begin{bmatrix} f_{V,V_0,\chi}(r_{i,\backslash i}, u_{i,\backslash i}) \\ h_{V_0}(u_{i,\backslash i}) \\ y_i \\ \hline \mathcal{M}x_i \end{bmatrix}, c_i, \epsilon_i \right) \right] \right| = O\left( \frac{\text{polylog}(n)}{\sqrt{n}} \right).$$

Returning to the original objective (20), one finally reaches

$$\left| \mathbb{E}\left[ \varphi\left( \tilde{R}_{i,\backslash i}, c_i, \epsilon_i \right) \right] - \mathbb{E}\left[ \varphi\left( G, c, \epsilon \right) \right] \right| = O\left( \frac{\text{polylog}(n)}{n^{\frac{1}{4}}} \right),$$

which completes the proof. □

## A.5. Deterministic equivalents

To summarize the proof so far, we have shown in subsection A.3 that the summary statistics $\mathfrak{m}_c = \langle \hat{w}, \mu_c \rangle$, $\mathfrak{h} = \langle \hat{w}, \beta \rangle$, $\mathfrak{t} = \langle \hat{w}, \hat{w}_0 \rangle$, $\mathfrak{q} = \|\hat{w}\|^2$ and $\mathfrak{V} = 1/n \, \text{tr}\left[ H_{\backslash i}^{-1} \right]$, $\mathfrak{X} = 1/n \, \text{tr}\left[ H^{-1} G H_0^{-1} \right]$ concentrate in $L_2$ to their expectations $m_c, \theta, t, q, V, \chi$. In this subsection, we compute the latter, leveraging the results of subsection A.4 that characterize the distribution of the involved residuals. Throughout this section $g \sim \mathcal{N}(\Psi_c, \Phi)$ denotes a Gaussian variable with mean and covariance

$$\Psi_c = \begin{bmatrix} m_c \\ m_{0,c} \\ \nu_c \\ \hline \rho_c \end{bmatrix}, \qquad \Phi = \begin{bmatrix} q & t & \theta & | & m \\ t & q_0 & \theta_0 & | & m_0 \\ \theta & \theta_0 & \varrho & | & \nu \\ \hline m & m_0 & \nu & | & \rho \end{bmatrix},$$

conditionally on $c$. We will also use the shorthands

$$r = \text{prox}_{V\ell(\cdot, u, g_3, c, \epsilon)}\left( g_1 + \chi \partial_u \ell_0(u, g_3, c, \epsilon), g_3, c, \epsilon \right),$$
$$u = \text{prox}_{V_0 \ell_0(\cdot, g_3, c, \epsilon)}(g_2),$$

where the values of $\epsilon, c$ are understood from the expectations inside which $r, u$ appear. All expectations are understood to bear over $c, \epsilon, g$. We will also employ the shorthand $s_i = \langle x_i, \beta \rangle$, so that $y_i = \phi(s_i, c_i, \epsilon_i)$.

**Lemma A.27** (Characterization of $\theta$)**.** *The summary statistic $\theta = \mathbb{E}[\mathfrak{h}]$ is asymptotically characterized by the self-consistent equation*

$$\left| \theta + \frac{1}{\lambda} \mathbb{E}[\partial_r \ell(r, u, g_3, c, \epsilon) g_3] \right| = O\left( \frac{\text{polylog}(n)}{n^{\frac{1}{4}}} \right).$$

*Proof.* We start from the optimality identity

$$\hat{w} = -\frac{1}{\lambda} \frac{1}{n} \sum_{i \in [n]} \partial_r \ell_i(r_i, u_i) x_i.$$

Taking an inner product with $\beta$,

$$\langle \hat{w}, \beta \rangle = -\frac{1}{\lambda}\frac{1}{n}\sum_{i\in[n]} \partial_r \ell_i(r_i, u_i) s_i. \tag{21}$$

Before taking an expectation, we first approximate the full residuals $r_i, u_i$ by the surrogates $\tilde{r}_{i,i}, \tilde{u}_{i,i}$. The associated correction can be controlled as

$$\left| \frac{1}{n}\sum_{i\in[n]} \left( \partial_r \ell_i(r_i, u_i) - \partial_r \ell_i(\tilde{r}_{i,i}, \tilde{u}_{i,i}) \right) s_i \right|$$

$$\leq \left( \left\| \partial_r^2 \ell \right\|_\infty \vee \left\| \partial_r \partial_u \ell \right\|_\infty \right) \left( \sup_{i\in[n]} |r_i - \tilde{r}_{i,i}| + \sup_{i\in[n]} |u_i - \tilde{u}_{i,i}| \right) \sup_{i\in[n]} |s_i| = O_{L_k}\left( \frac{\text{polylog}(n)}{\sqrt{n}} \right).$$

We used Proposition A.5, Proposition A.14 and Lemma A.42. Thus, taking expectation in (21),

$$\theta = -\frac{1}{\lambda}\frac{1}{n}\sum_{i\in[n]} \mathbb{E}\left[ \partial_r \ell(\tilde{r}_{i,i}, \tilde{u}_{i,i}, y_i, c_i, \epsilon_i) s_i \right] + O\left( \frac{\text{polylog}(n)}{\sqrt{n}} \right)$$

By permutation symmetry of the problem, all expectations are equal, and

$$\theta = -\frac{1}{\lambda}\mathbb{E}\left[ \partial_r \ell(\tilde{r}_{1,1}, \tilde{u}_{1,1}, y_1, c_1, \epsilon_1) s_1 \right] + O\left( \frac{\text{polylog}(n)}{\sqrt{n}} \right)$$

Note that $\partial_r \ell \in \mathcal{F}$ from Assumption A.3. Thus, one can apply Lemma A.26, from which it follows that

$$\left| \mathbb{E}\left[ \partial_r \ell(\tilde{r}_{1,1}, \tilde{u}_{1,1}, y_1, c_1, \epsilon_1) s_1 \right] - \mathbb{E}\left[ \partial_r \ell(r, u, g_3, c, \epsilon) g_3 \right] \right| \xrightarrow{n\to\infty} 0.$$

This proves the Lemma. $\qquad\square$

A similar characterization for $m_c$ follows from identical steps.

**Lemma A.28** (Characterization of $m_c$)**.** *For any $c \in \mathcal{V}$, the summary statistic $m_c = \mathbb{E}[\mathfrak{m}_c]$ is asymptotically characterized by the self-consistent equation*

$$\left| m_c + \frac{1}{\lambda}\mathbb{E}\left[ \partial_r \ell(r, u, g_3, c, \epsilon) g_c \right] \right| = O\left( \frac{\text{polylog}(n)}{n^{\frac{1}{4}}} \right),$$

*where $g_c$ is the entry of $g$ associated to $c$.*

*Proof.* The proof proceeds in identical fashion to that of Lemma A.27. Taking an inner product on the optimality condition,

$$\langle \hat{w}, \mu_c \rangle = -\frac{1}{\lambda}\frac{1}{n}\sum_{i\in[n]} \partial_r \ell(r_i, u_i, y_i, c_i, \epsilon_i) \langle x_i, \mu_c \rangle.$$

Once more, we approximate the full residuals $r_i, u_i$ by the surrogates $\tilde{r}_{i,i}, \tilde{u}_{i,i}$. The associated correction can be controlled as

$$\left| \frac{1}{n}\sum_{i\in[n]} \left( \partial_r \ell(r_i, u_i, y_i, c_i, \epsilon_i) - \partial_r \ell(\tilde{r}_{i,i}, \tilde{u}_{i,i}, y_i, c_i, \epsilon_i) \right) \langle x_i, \mu_c \rangle \right|$$

$$\leq \left( \left\| \partial_r^2 \ell \right\|_\infty \vee \left\| \partial_r \partial_u \ell \right\|_\infty \right) \left( \sup_{i\in[n]} |r_i - \tilde{r}_{i,i}| + \sup_{i\in[n]} |u_i - \tilde{u}_{i,i}| \right) \sup_{i\in[n]} |\langle x_i, \mu_c \rangle| = O_{L_k}\left( \frac{\text{polylog}(n)}{\sqrt{n}} \right).$$

We used Proposition A.5, Proposition A.14 and Lemma A.42 on the normalized vector $\mu_c/\|\mu\|_c$. The remainder of the proof is identical to that of Lemma A.27. $\qquad\square$

**Lemma A.29** (Characterization of $t$). *The summary statistic $t = \mathbb{E}[\mathfrak{t}]$ is asymptotically characterized by the self-consistent equation*

$$\left| t + \frac{1}{\lambda} \mathbb{E}[\partial_r \ell(r, u, g_3, c, \epsilon) u] \right| = O\left( \frac{\text{polylog}(n)}{n^{\frac{1}{4}}} \right).$$

*Proof.* We again follow the lines of Lemma A.27. Taking an inner product on the optimality condition,

$$\langle \hat{w}, \hat{w}_0 \rangle = -\frac{1}{\lambda} \frac{1}{n} \sum_{i \in [n]} \partial_r \ell(r_i, u_i, y_i, c_i, \epsilon_i) u_i.$$

Once more, we approximate the full residuals $r_i, u_i$ by the surrogates $\tilde{r}_{i,i}, \tilde{u}_{i,i}$. The associated correction can be controlled as

$$\left| \frac{1}{n} \sum_{i \in [n]} \left( \partial_r \ell(r_i, u_i, y_i, c_i, \epsilon_i) - \partial_r \ell(\tilde{r}_{i,i}, \tilde{u}_{i,i}, y_i, c_i, \epsilon_i) \right) u_i + \partial_r \ell(\tilde{r}_{i,i}, \tilde{u}_{i,i}, y_i, c_i, \epsilon_i)(u_i - \tilde{u}_{i,i}) \right|$$

$$\leq \left( \|\partial_r^2 \ell\|_\infty \vee \|\partial_r \partial_u \ell\|_\infty \right) \left( \sup_{i \in [n]} |r_i - \tilde{r}_{i,i}| + \sup_{i \in [n]} |u_i - \tilde{u}_{i,i}| \right) \sup_{i \in [n]} |u_i| + \|\partial_r \ell\|_\infty \sup_{i \in [n]} |u_i - \tilde{u}_{i,i}|.$$

Controlling $\sup_{i \in [n]} |u_i|$ requires slightly more work. One has

$$\sup_{i \in [n]} |u_i| \leq \sup_{i \in [n]} |u_i - u_{i,\setminus i}| + \sup_{i \in [n]} |u_{i,\setminus i}| \leq O_{L_k}\left( \frac{\text{polylog}(n)}{\sqrt{n}} \right) + O_{L_k}(\text{polylog}(n)).$$

We used Proposition A.5 with Lemma A.10 with a Cauchy-Schwartz bound for the first term, and Lemma A.42 in conjunction with Lemma A.37 for the second. Thus,

$$\left| \frac{1}{n} \sum_{i \in [n]} \left( \partial_r \ell(r_i, u_i, y_i, c_i, \epsilon_i) - \partial_r \ell(\tilde{r}_{i,i}, \tilde{u}_{i,i}, y_i, c_i, \epsilon_i) \right) u_i \right| = O_{L_k}\left( \frac{\text{polylog}(n)}{\sqrt{n}} \right).$$

Thus, taking expectations and by symmetry,

$$t = -\frac{1}{\lambda} \mathbb{E}[\partial_r \ell(\tilde{r}_{1,1}, \tilde{u}_{1,1}, y_1, c_1, \epsilon_1) \tilde{u}_{1,1}] + O\left( \frac{\text{polylog}(n)}{\sqrt{n}} \right).$$

It remains to check that the function $\varphi : r, u, s, c, \epsilon \to \partial_r \ell(r, u, \phi(s, c, \epsilon), c, \epsilon) u$ belongs to the class $\mathcal{F}$. For any pairs $(r_1, u_1), (r_2, u_2)$, the mean value theorem guarantees the existence of $(\check{r}, \check{u})$ on the line connecting the points such that

$$|\varphi(r_1, u_1, s, c, \epsilon) - \varphi(r_2, u_2, s, c, \epsilon)| \leq \|\nabla_{r,u} \varphi(\check{r}, \check{u}, s, c, \epsilon)\| \sqrt{(r_1 - r_2)^2 + (u_1 - u_2)^2}$$

where $\|\nabla_{r,u} \varphi(\check{r}, \check{u}, s, c, \epsilon)\|$ denotes the norm of the gradient of $\varphi$ with respect to its first two arguments, evaluated at $(\check{r}, \check{u})$. This norm can in turn be bounded as

$$\|\nabla_{r,u} \varphi(\check{r}, \check{u}, s, c, \epsilon)\| = \sqrt{(\partial_r^2 \ell(\check{r}, \check{u}, \phi(s, c, \epsilon), c, \epsilon) \check{u})^2 + (\partial_r \partial_u \ell(\check{r}, \check{u}, \phi(s, c, \epsilon), c, \epsilon) \check{u} + \partial_r \ell(\check{r}, \check{u}, \phi(s, c, \epsilon), c, \epsilon))^2}$$

$$\leq \sqrt{\|\partial_r^2 \ell\|_\infty^2 \check{u}^2 + 2\|\partial_r \partial_u \ell\|_\infty^2 \check{u}^2 + 2\|\partial_r \ell\|_\infty^2}$$

$$\leq \sqrt{\|\partial_r^2 \ell\|_\infty^2 + 2\|\partial_r \partial_u \ell\|_\infty^2} (|u_1| + |u_2|) + \sqrt{2\|\partial_r \ell\|_\infty^2}.$$

Thus $\varphi$ satisfies the local Lipschitz property (18). Furthermore, $\varphi$ vanishes when evaluated at $0$, implying trivially the existence of a polynomial majorant. Thus, $\varphi \in \mathcal{F}$, and from Lemma A.26,

$$|\mathbb{E}[\partial_r \ell(\tilde{r}_{1,1}, \tilde{u}_{1,1}, y_1, c_1, \epsilon_1) \tilde{u}_{1,1}] - \mathbb{E}[\partial_r \ell(r, u, g_3, c, \epsilon) u]| \xrightarrow{n \to \infty} 0.$$

This proves the Lemma. $\qquad\square$

**Lemma A.30** (Characterization of $q$)**.** *The summary statistic $q = \mathbb{E}[\mathfrak{q}]$ is asymptotically characterized by the self-consistent equation*

$$\left| q + \frac{1}{\lambda} \mathbb{E}[\partial_r \ell(r, u, g_3, c, \epsilon) r] \right| = O\left( \frac{\mathrm{polylog}(n)}{n^{\frac{1}{4}}} \right).$$

*Proof.* We again follow the lines of Lemma A.27. Taking an inner product on the optimality condition, and from steps identical to those of the previous Lemmas,

$$\begin{aligned}
\|\hat{w}\|^2 &= -\frac{1}{\lambda} \frac{1}{n} \sum_{i \in [n]} \partial_r \ell(r_i, u_i, y_i, c_i, \epsilon_i) r_i \\
&= -\frac{1}{\lambda} \frac{1}{n} \sum_{i \in [n]} \partial_r \ell(\tilde{r}_{i,i}, \tilde{u}_{i,i}, y_i, c_i, \epsilon_i) \tilde{r}_{i,i} + O_{L_k}(\mathrm{polylog}(n)).
\end{aligned}$$

Thus, taking expectations and by symmetry,

$$q = -\frac{1}{\lambda} \mathbb{E}[\partial_r \ell(\tilde{r}_{1,1}, \tilde{u}_{1,1}, y_1, c_1, \epsilon_1) \tilde{r}_{1,1}] + O\left( \frac{\mathrm{polylog}(n)}{\sqrt{n}} \right)$$

Again, before applying Lemma A.26, one needs to check that the function $\varphi : r, u, s, c, \epsilon \to \partial_r \ell(r, u, \phi(s, c, \epsilon), c, \epsilon) r$ belongs to the class $\mathcal{F}$. It suffices similarly to the proof of Lemma A.29 to bound the norm of the gradient $\|\nabla_{r,u} \varphi(\check{r}, \check{u}, s, c, \epsilon)\|$ evaluated at $(\check{r}, \check{u})$ on the line connecting $(r_1, u_1), (r_2, u_2)$. This norm can in turn be bounded as

$$\begin{aligned}
\|\nabla_{r,u} \varphi(\check{r}, \check{u}, s, c, \epsilon)\| &= \sqrt{(\partial_r \partial_u \ell(\check{r}, \check{u}, \phi(s, c, \epsilon), c, \epsilon) \check{r})^2 + (\partial_r^2 \ell(\check{r}, \check{u}, \phi(s, c, \epsilon), c, \epsilon) \check{r} + \partial_r \ell(\check{r}, \check{r}, \phi(s, c, \epsilon), c, \epsilon))^2} \\
&\leq \sqrt{\|\partial_r \partial_u \ell\|_\infty^2 + 2\|\partial_\ell\|_\infty^2}(|r_1| + |r_2|) + \sqrt{2\|\partial_r \ell\|_\infty^2}.
\end{aligned}$$

Thus, $\varphi \in \mathcal{F}$, and from Lemma A.26,

$$|\mathbb{E}[\partial_r \ell(\tilde{r}_{1,1}, \tilde{u}_{1,1}, y_1, c_1, \epsilon_1) \tilde{r}_{1,1}] - \mathbb{E}[\partial_r \ell(r, u, g_3, c, \epsilon) r]| \xrightarrow{n \to \infty} 0.$$

This proves the Lemma. $\qquad\square$

**Lemma A.31** (Characterization of $V$)**.** *The summary statistic $V = \mathbb{E}[\mathfrak{V}]$ is asymptotically characterized by the self-consistent equation*

$$\left| -\frac{1}{n/dV} + \mathbb{E}\left[ \frac{\partial_r^2 \ell(r, u, g_3, c, \epsilon)}{1 + V \partial_r^2 \ell(r, u, g_3, c, \epsilon)} \right] + \lambda \right| = O\left( \frac{\mathrm{polylog}(n)}{n^{\frac{1}{4}}} \right).$$

*Proof.* The proof strategy follows closely that of Lemma 15 in (Barnfield et al., 2025). By definition of $H$, the identity

$$\frac{1}{n^2} \sum_{i \in [n]} \partial_r^2 \ell_i(r_i, u_i) x_i^\top H^{-1} x_i + \lambda \mathfrak{V} = \frac{d}{n}$$

follows. Applying the matrix inversion lemma yields the more compact expression

$$\frac{1}{n} \sum_{i \in [n]} \frac{\partial_r^2 \ell_i(r_i, u_i) \hat{\gamma}_i}{1 + \partial_r^2 \ell_i(r_i, u_i) \hat{\gamma}_i} + \lambda \mathfrak{V} = \frac{d}{n}, \tag{22}$$

where we defined $\hat{\gamma}_i = {}^1\!/n x_i^\top \hat{H}_i^{-1} x_i$ and

$$\hat{H}_i = \frac{1}{n} \sum_{j \neq i} \partial_r^2 \ell_j(r_j, u_j) x_j x_j^\top + \lambda I_d = H - \frac{1}{n} \partial_r^2 \ell_i(r_i, u_i) x_i x_i^\top.$$

To use Lemma A.21, we first establish that $\hat{\gamma}_i \approx \gamma_i$. To see this, we start from a coarse Cauchy-Schwartz bound

$$
\sup_{i \in [n]} |\hat{\gamma}_i - \gamma_i| \leq \sup_{i \in [n]} \frac{\|x_i\|^2}{n} \frac{1}{\lambda^2} \sup_{i \in [n]} \left\| \hat{H}_i - H_{\setminus i} \right\|
$$

$$
= \sup_{i \in [n]} \frac{\|x_i\|^2}{n} \frac{1}{\lambda^2} \sup_{i \in [n]} \frac{1}{n} \left\| X_{\setminus i}^\top \Lambda X_{\setminus i} \right\|
$$

$$
\leq \sup_{i \in [n]} \frac{\|x_i\|^2}{n} \frac{1}{\lambda^2} \sup_{i \in [n]} \frac{\|X_{\setminus i}\|^2}{n} \sup_{i \in [n]} \|\Lambda_i\|,
$$

introducing the diagonal matrix $\Lambda_i \in \mathbb{R}^{(n-1) \times (n-1)}$ with entries $\Lambda_{i,jj} = \partial_r^2 \ell_j(r_j, u_j) - \partial_r^2 \ell_j(r_{j,\setminus i}, u_{j,\setminus i})$. Proposition A.5 and Proposition A.14 allow to bound the norm of $\Lambda$ as

$$
\sup_{i \in [n]} \|\Lambda_i\| \leq \left( \|\partial_r^3 \ell\|_\infty \vee \|\partial_r^2 \partial_u \ell\|_\infty \right) \left( \sup_{i \in [n]} \sup_{j \neq i} |r_j - r_{j,\setminus i}| + \sup_{i \in [n]} \sup_{j \neq i} |u_j - u_{j,\setminus i}| \right) = O_{L_k} \left( \frac{\text{polylog}(n)}{\sqrt{n}} \right).
$$

Thus

$$
\sup_{i \in [n]} |\hat{\gamma}_i - \gamma_i| = O_{L_k} \left( \frac{\text{polylog}(n)}{\sqrt{n}} \right).
$$

As a consequence,

$$
\left| \frac{1}{n} \sum_{i \in [n]} \frac{\partial_r^2 \ell_i(r_i, u_i) \hat{\gamma}_i}{1 + \partial_r^2 \ell_i(r_i, u_i) \hat{\gamma}_i} - \frac{1}{n} \sum_{i \in [n]} \frac{\partial_r^2 \ell_i(r_i, u_i) V}{1 + \partial_r^2 \ell_i(r_i, u_i) V} \right| \leq \|\partial_r^2 \ell\|_\infty \left( \sup_{i \in [n]} |\hat{\gamma}_i - \gamma_i| + \sup_{i \in [n]} |\mathfrak{V} - \gamma_i| + |\mathfrak{V} - V| \right)
$$

$$
= O_{L_2} \left( \frac{\text{polylog}(n)}{\sqrt{n}} \right).
$$

using the concentration results of Lemma A.21. It now remains to approximate the full residuals $r_i, u_i$ by the surrogates $\tilde{r}_{i,i}, \tilde{u}_{i,i}$. The difference can be bounded as

$$
\sup_{i \in [n]} \left| \partial_r^2 \ell_i(r_i, u_i) - \partial_r^2 \ell_i(\tilde{r}_{i,i}, \tilde{u}_{i,i}) \right| \leq \left( \|\partial_r^3 \ell\|_\infty \vee \|\partial_r^2 \partial_u \ell\|_\infty \right) \left( \sup_{i \in [n]} |r_i - \tilde{r}_{i,i}| + \sup_{i \in [n]} |u_i - \tilde{u}_{i,i}| \right)
$$

$$
= O_{L_k} \left( \frac{\text{polylog}(n)}{\sqrt{n}} \right),
$$

using Proposition A.14. Thus,

$$
\left| \frac{1}{n} \sum_{i \in [n]} \frac{\partial_r^2 \ell_i(r_i, u_i) V}{1 + \partial_r^2 \ell_i(r_i, u_i) V} - \frac{1}{n} \sum_{i \in [n]} \frac{\partial_r^2 \ell_i(\tilde{r}_{i,i}, \tilde{u}_{i,i}) V}{1 + \partial_r^2 \ell_i(\tilde{r}_{i,i}, \tilde{u}_{i,i}) V} \right| = O_{L_k} \left( \frac{\text{polylog}(n)}{\sqrt{n}} \right).
$$

Returning to (22),

$$
\frac{1}{n} \sum_{i \in [n]} \frac{\partial_r^2 \ell_i(\tilde{r}_{i,i}, \tilde{u}_{i,i}) V}{1 + \partial_r^2 \ell_i(\tilde{r}_{i,i}, \tilde{u}_{i,i}) V} + \lambda V + O_{L_2} \left( \frac{\text{polylog}(n)}{\sqrt{n}} \right) = \frac{d}{n}.
$$

Taking expectations, and again appealing to the permutation symmetry of the samples,

$$
\mathbb{E} \left[ \frac{\partial_r^2 \ell_i(\tilde{r}_{1,1}, \tilde{u}_{1,1}, y_1, c_1, \epsilon_1) V}{1 + \partial_r^2 \ell(\tilde{r}_{1,1}, \tilde{u}_{1,1}, y_1, c_1, \epsilon_1) V} \right] + \lambda V + O_{L_2} \left( \frac{\text{polylog}(n)}{\sqrt{n}} \right) = \frac{d}{n}.
$$

Since $\varphi : r, u, s, c, \epsilon \to {}^1/_{1 + \partial_r^2 \ell(r, u, \phi(s, c, \epsilon), c, \epsilon) V}$ is Lipschitz and bounded, it belongs to $\mathcal{F}$. Thus, applying Lemma A.26,

$$
\left| \mathbb{E} \left[ \frac{\partial_r^2 \ell_i(\tilde{r}_{1,1}, \tilde{u}_{1,1}, y_1, c_1, \epsilon_1)}{1 + \partial_r^2 \ell(\tilde{r}_{1,1}, \tilde{u}_{1,1}, y_1, c_1, \epsilon_1) V} \right] - \mathbb{E} \left[ \frac{\partial_r^2 \ell(r, u, g_3, c, \epsilon)}{1 + V \partial_r^2 \ell(r, u, g_3, c, \epsilon)} \right] \right| = O \left( \frac{\text{polylog}(n)}{n^{\frac{1}{4}}} \right).
$$

This completes the proof. $\qquad \square$

Before moving on to characterizing $\chi$, we first need to establish, as an intermediary step, a similar result for $\Omega = \mathbb{E}[\mathfrak{W}]$.

**Lemma A.32** (Characterization of $\Omega$). *The summary statistic $\Omega = \mathbb{E}[\mathfrak{W}]$ is asymptotically characterized by the equation*

$$\left| \Omega - \frac{d}{n} \frac{\mathbb{E}\left[ \frac{\partial_r \partial_u \ell(r, u, g_3, c, \epsilon)}{1 + V_0 \partial_u^2 \ell_0(u, g_3, c, \epsilon)} \right]}{\lambda_0 + \mathbb{E}\left[ \frac{\partial_u^2 \ell_0(u, g_3, c, \epsilon)}{1 + V_0 \partial_u^2 \ell_0(u, g_3, c, \epsilon)} \right]} \right| = O\left( \frac{\mathrm{polylog}(n)}{n^{\frac{1}{4}}} \right).$$

*Proof.* We start from the identity

$$G = \frac{1}{n} \sum_{i \in [n]} \partial_r \partial_u \ell_i(r_i, u_i) x_i x_i^\top H_0^{-1} G + \lambda_0 H_0^{-1} G.$$

Taking the trace and denoting $\mathfrak{G} = 1/n \, \mathrm{tr}[G]$,

$$\mathfrak{G} = \frac{1}{n} \sum_{i \in [n]} \partial_u^2 \ell_{0,i}(u_i) x_i^\top \left( \hat{H}_{0,i} + \partial_u^2 \ell_{0,i}(u_i) \frac{x_i x_i^\top}{n} \right)^{-1} \left( \hat{G}_i + \partial_r \partial_u \ell_i(r_i, u_i) \frac{x_i x_i^\top}{n} \right) x_i + \lambda_0 \mathfrak{W},$$

where we defined

$$\hat{H}_0 = \frac{1}{n} \sum_{j \neq i} \partial_u^2 \ell_{0,j}(u_j) x_j x_j^\top + \lambda_0 I_d,$$

$$G = \frac{1}{n} \sum_{j \neq i} \partial_r \partial_u \ell_j(r_j, u_j) x_j x_j^\top.$$

Denoting $\hat{\gamma}_{0,i} = 1/n x_i^\top \hat{H}_{0,i}^{-1} x_i, \hat{\omega}_i = 1/n x_i^\top \hat{H}_{0,i}^{-1} \hat{G}_i x_i$, and using the matrix inversion lemma, one has

$$\mathfrak{G} = \frac{1}{n} \sum_{i \in [n]} \partial_u^2 \ell_{0,i}(u_i) \left[ \hat{\omega}_i - \frac{\partial_u^2 \ell_{0,i}(u_i)}{1 + \partial_u^2 \ell_{0,i}(u_i) \hat{\gamma}_{0,i}} \hat{\gamma}_{0,i} \hat{\omega}_i + \frac{d}{n} \partial_r \partial_u \ell_i(r_i, u_i) \hat{\gamma}_{0,i} - \frac{d}{n} \hat{\gamma}_{0,i}^2 \frac{\partial_u^2 \ell_{0,i}(u_i) \partial_r \partial_u \ell_i(r_i, u_i)}{1 + \partial_u^2 \ell_{0,i}(u_i) \hat{\gamma}_{0,i}} \right]$$
$$+ \lambda_0 \mathfrak{G}.$$

Using the same line of reasoning as the proof of Lemma A.31, one reaches the approximations

$$\sup_{i \in [n]} |\hat{\gamma}_{0,i} - \omega_i| = O_{L_k}\left( \frac{\mathrm{polylog}(n)}{\sqrt{n}} \right),$$

$$\sup_{i \in [n]} |\hat{\omega}_i - \omega_i| = O_{L_k}\left( \frac{\mathrm{polylog}(n)}{\sqrt{n}} \right).$$

Thus,

$$\left| \frac{1}{n} \sum_{i \in [n]} \partial_u^2 \ell_{0,i}(u_i) \hat{\omega}_i - \frac{1}{n} \sum_{i \in [n]} \partial_u^2 \ell_{0,i}(u_i) \Omega \right| \leq \|\partial_r \partial_u \ell\|_\infty \left( \sup_{i \in [n]} |\hat{\omega}_i - \omega_i| + \sup_{i \in [n]} |\mathfrak{W} - \omega_i| + |\mathfrak{W} - \Omega| \right)$$
$$= O_{L_2}\left( \frac{\mathrm{polylog}(n)}{\sqrt{n}} \right).$$

By the same token

$$\left| \frac{1}{n} \sum_{i \in [n]} \frac{\partial_u^2 \ell_{0,i}(u_i)^2}{1 + \partial_u^2 \ell_{0,i}(u_i) \hat{\gamma}_{0,i}} \hat{\gamma}_{0,i} \hat{\omega}_i - \frac{1}{n} \sum_{i \in [n]} \frac{\partial_u^2 \ell_{0,i}(u_i)^2}{1 + \partial_u^2 \ell_{0,i}(u_i) V_0} V_0 \Omega \right| = O_{L_2}\left( \frac{\mathrm{polylog}(n)}{\sqrt{n}} \right)$$

$$\left| \frac{1}{n} \sum_{i \in [n]} \partial_u^2 \ell_{0,i}(u_i) \partial_r \partial_u \ell_i(r_i, u_i) \hat{\gamma}_{0,i} - \frac{1}{n} \sum_{i \in [n]} \partial_u^2 \ell_{0,i}(u_i) \partial_r \partial_u \ell_i(r_i, u_i) V_0 \right| = O_{L_2}\left( \frac{\mathrm{polylog}(n)}{\sqrt{n}} \right)$$

$$\left| \frac{1}{n} \sum_{i \in [n]} \hat{\gamma}_{0,i}^2 \frac{\partial_u^2 \ell_{0,i}(u_i)^2 \partial_r \partial_u \ell_i(r_i, u_i)}{1 + \partial_u^2 \ell_{0,i}(u_i) \hat{\gamma}_{0,i}} - \frac{1}{n} \sum_{i \in [n]} V_0^2 \frac{\partial_u^2 \ell_{0,i}(u_i)^2 \partial_r \partial_u \ell_i(r_i, u_i)}{1 + \partial_u^2 \ell_{0,i}(u_i) V_0} \right| = O_{L_2}\left( \frac{\mathrm{polylog}(n)}{\sqrt{n}} \right).$$

Thus,

$$\mathfrak{G} = \frac{1}{n}\sum_{i\in[n]}\partial_u^2\ell_{0,i}(u_i)\left[\Omega - \frac{\partial_u^2\ell_{0,i}(u_i)}{1+\partial_u^2\ell_{0,i}(u_i)V_0}V_0\Omega + \frac{d}{n}\partial_r\partial_u\ell_i(r_i,u_i)V_0 - \frac{d}{n}V_0^2\frac{\partial_u^2\ell_{0,i}(u_i)\partial_r\partial_u\ell_i(r_i,u_i)}{1+\partial_u^2\ell_{0,i}(u_i)V_0}\right]$$
$$+ \lambda_0\mathfrak{W} + O_{L_2}\left(\frac{\mathrm{polylog}(n)}{\sqrt{n}}\right).$$

Now using the approximations

$$\sup_{i\in[n]}|\partial_r\partial_u\ell_i(r_i,u_i) - \partial_r\partial_u\ell_i(\tilde{r}_{i,i},\tilde{u}_{i,i})| = O_{L_k}\left(\frac{\mathrm{polylog}(n)}{\sqrt{n}}\right),$$
$$\sup_{i\in[n]}\left|\partial_u^2\ell_{0,i}(u_i) - \partial_u^2\ell_{0,i}(\tilde{u}_{i,i})\right| = O_{L_k}\left(\frac{\mathrm{polylog}(n)}{\sqrt{n}}\right),$$

that follow from Proposition A.5 and A.14, one has (using that $\Omega = O(\mathrm{polylog}(n))$):

$$\mathfrak{G} = \frac{1}{n}\sum_{i\in[n]}\partial_u^2\ell_{0,i}(\tilde{u}_{i,i})\left[\frac{\Omega}{1+\partial_u^2\ell_{0,i}(\tilde{u}_{i,i})V_0} + \frac{d}{n}\frac{\partial_r\partial_u\ell_i(\tilde{r}_{i,i},\tilde{u}_{i,i})V_0}{1+\partial_u^2\ell_{0,i}(\tilde{u}_{i,i})V_0}\right] + \lambda_0\mathfrak{W} + O_{L_2}\left(\frac{\mathrm{polylog}(n)}{\sqrt{n}}\right).$$

Taking the expectation, and using again the argument of symmetry between sample indices, yields

$$\Gamma = \mathbb{E}\left[\partial_u^2\ell_{0,i}(\tilde{u}_{1,1})\left[\frac{\Omega}{1+\partial_u^2\ell_{0,i}(\tilde{u}_{1,1})V_0} + \frac{d}{n}\frac{\partial_r\partial_u\ell_i(\tilde{r}_{1,1},\tilde{u}_{1,1})V_0}{1+\partial_u^2\ell_{0,i}(\tilde{u}_{1,1})V_0}\right]\right] - \lambda_0\Omega + O\left(\frac{\mathrm{polylog}(n)}{\sqrt{n}}\right),$$

denoting $\Gamma = \mathbb{E}[\mathfrak{G}]$. One can readily verify that all the functions in the expectation are Lipschitz with respect to $\tilde{r}_{1,1}, \tilde{u}_{1,1}$. Applying Lemma A.26,

$$\Gamma - \mathbb{E}\left[\partial_u^2\ell_0(u,g_3,c,\epsilon)\left[\frac{\Omega}{1+\partial_u^2\ell_0(u,g_3,c,\epsilon)V_0} + \frac{d}{n}\frac{\partial_r\partial_u\ell(r,u,g_3,c,\epsilon)V_0}{1+\partial_u^2\ell_0(u,g_3,c,\epsilon)V_0}\right]\right] + \lambda_0\Omega = O\left(\frac{\mathrm{polylog}(n)}{n^{\frac{1}{4}}}\right).$$

It remains to compute $\Gamma$. By definition

$$\mathfrak{G} = \frac{1}{n}\sum_{j\neq i}\partial_r\partial_u\ell_j(r_j,u_j)\frac{\|x_i\|^2}{n} = \frac{d}{n^2}\sum_{j\neq i}\partial_r\partial_u\ell_j(r_j,u_j) + O_{L_k}\left(\frac{\mathrm{polylog}(n)}{\sqrt{n}}\right).$$

Since using Lemma A.41, $\sup_{i\in[n]}|\|x_i\|^2/d - 1| = O_{L_k}(\mathrm{polylog}(n)/\sqrt{n})$. Taking expectation, and following the same steps as above yields

$$\Gamma = \frac{d}{n}\mathbb{E}[\partial_r\partial_u\ell(r,u,g_3,c,\epsilon)].$$

Thus, finally

$$\Omega\left(\lambda_0 + \mathbb{E}\left[\frac{\partial_u^2\ell_0(u,g_3,c,\epsilon)}{1+\partial_u^2\ell_0(u,g_3,c,\epsilon)V_0}\right]\right) - \mathbb{E}\left[\frac{\partial_r\partial_u\ell(u,g_3,c,\epsilon)}{1+\partial_u^2\ell_0(u,g_3,c,\epsilon)V_0}\right] = O\left(\frac{\mathrm{polylog}(n)}{n^{\frac{1}{4}}}\right),$$

proving the Lemma. $\square$

**Lemma A.33** (Characterization of $\chi$). *The summary statistic $\chi = \mathbb{E}[\mathfrak{X}]$ is asymptotically characterized by the equation*

$$\left|\chi - \frac{\frac{d}{n}\frac{\mathbb{E}\left[\frac{\partial_r\partial_u\ell(r,u,g_3,c,\epsilon)}{1+V_0\partial_u^2\ell_0(u,g_3,c,\epsilon)}\right]}{\lambda_0+\mathbb{E}\left[\frac{\partial_u^2\ell_0(u,g_3,c,\epsilon)}{1+V_0\partial_u^2\ell_0(u,g_3,c,\epsilon)}\right]} - \mathbb{E}\left[\frac{\partial_r^2\ell(r,u,g_3,c,\epsilon)\partial_r\partial_u\ell(r,u,g_3,c,\epsilon)V_0V}{(1+V\partial_r^2\ell(r,u,g_3,c,\epsilon))(1+V_0\partial_u^2\ell_0(u,g_3,c,\epsilon))}\right]}{\lambda+\mathbb{E}\left[\frac{\partial_r^2\ell(r,u,g_3,c,\epsilon)}{(1+V\partial_r^2\ell(r,u,g_3,c,\epsilon))(1+V_0\partial_u^2\ell_0(u,g_3,c,\epsilon))}\right]}\right| = O\left(\frac{\mathrm{polylog}(n)}{n^{\frac{1}{4}}}\right).$$

*Proof.* We again adopt the same proof strategy, taking as a starting point the identity

$$GH_0^{-1} = \frac{1}{n} \sum_{i \in [n]} \partial_r^2 \ell_i(r_i, u_i) x_i x_i^\top H^{-1} G H_0^{-1} + \lambda H^{-1} G H_0^{-1},$$

namely, by taking a normalized trace

$$\mathfrak{W} = \frac{1}{n} \sum_{i \in [n]} \partial_r^2 \ell_i(r_i, u_i) \frac{1}{n} x_i^\top H^{-1} G H_0^{-1} x_i + \lambda \mathfrak{X}.$$

Using the matrix inversion lemma,

$$\mathfrak{W} = \frac{1}{n} \sum_{i \in [n]} \partial_r^2 \ell_i(r_i, u_i) \left[ \hat{v}_i - \frac{\partial_r^2 \ell_i(r_i, u_i)}{1 + \hat{\gamma}_i \partial_r^2 \ell_i(r_i, u_i)} \hat{v}_i \hat{\gamma}_i - \frac{\partial_u^2 \ell_{0,i}(u_i)}{1 + \hat{\gamma}_{0,i} \partial_u^2 \ell_{0,i}(u_i)} \hat{\gamma}_{0,i} \hat{v}_i + \partial_r \partial_u \ell_i(r_i, u_i) \hat{\gamma}_{0,i} \hat{\gamma}_i \right.$$

$$- \frac{\partial_r \partial_u \ell_i(r_i, u_i) \partial_r^2 \ell_i(r_i, u_i)}{1 + \hat{\gamma}_i \partial_r^2 \ell_i(r_i, u_i)} \hat{\gamma}_{0,i} \hat{\gamma}_i^2 - \frac{\partial_r \partial_u \ell_i(r_i, u_i) \partial_u^2 \ell_{0,i}(u_i)}{1 + \hat{\gamma}_{0,i} \partial_u^2 \ell_{0,i}(u_i)} \hat{\gamma}_{0,i}^2 \hat{\gamma}_i$$

$$+ \frac{\partial_u^2 \ell_{0,i}(u_i)}{1 + \hat{\gamma}_{0,i} \partial_u^2 \ell_{0,i}(u_i)} \frac{\partial_r^2 \ell_i(r_i, u_i)}{1 + \hat{\gamma}_i \partial_r^2 \ell_i(r_i, u_i)} \hat{\gamma}_{0,i} \hat{v}_i \hat{\gamma}_i$$

$$\left. + \partial_r \partial_u \ell_i(r_i, u_i) \frac{\partial_u^2 \ell_{0,i}(u_i)}{1 + \hat{\gamma}_{0,i} \partial_u^2 \ell_{0,i}(u_i)} \frac{\partial_r^2 \ell_i(r_i, u_i)}{1 + \hat{\gamma}_i \partial_r^2 \ell_i(r_i, u_i)} \hat{\gamma}_{0,i}^2 \hat{\gamma}_i^2 \right] + \lambda \mathfrak{X}$$

defining $\hat{v}_i = \frac{1}{n} \operatorname{tr}\left[ \hat{H}_i^{-1} \hat{G}_i \hat{H}_{0,i}^{-1} \right]$. Using the concentration Lemmas A.21 and A.22,

$$\mathfrak{W} = \frac{1}{n} \sum_{i \in [n]} \partial_r^2 \ell_i(r_i, u_i) \left[ \chi - \frac{\partial_r^2 \ell_i(r_i, u_i)}{1 + V \partial_r^2 \ell_i(r_i, u_i)} \chi V - \frac{\partial_u^2 \ell_{0,i}(u_i)}{1 + V_0 \partial_u^2 \ell_{0,i}(u_i)} V_0 \chi + \partial_r \partial_u \ell_i(r_i, u_i) V_0 V \right.$$

$$- \frac{\partial_r \partial_u \ell_i(r_i, u_i) \partial_r^2 \ell_i(r_i, u_i)}{1 + V \partial_r^2 \ell_i(r_i, u_i)} V_0 V^2 - \frac{\partial_r \partial_u \ell_i(r_i, u_i) \partial_u^2 \ell_{0,i}(u_i)}{1 + V_0 \partial_u^2 \ell_{0,i}(u_i)} V_0^2 V + \frac{\partial_u^2 \ell_{0,i}(u_i)}{1 + V_0 \partial_u^2 \ell_{0,i}(u_i)} \frac{\partial_r^2 \ell_i(r_i, u_i)}{1 + V \partial_r^2 \ell_i(r_i, u_i)} V_0 \chi V$$

$$\left. + \partial_r \partial_u \ell_i(r_i, u_i) \frac{\partial_u^2 \ell_{0,i}(u_i)}{1 + V_0 \partial_u^2 \ell_{0,i}(u_i)} \frac{\partial_r^2 \ell_i(r_i, u_i)}{1 + V \partial_r^2 \ell_i(r_i, u_i)} V_0^2 V^2 \right] + \lambda \mathfrak{X} + O_{L_2} \left( \frac{\operatorname{polylog}(n)}{\sqrt{n}} \right)$$

$$= \chi \frac{1}{n} \sum_{i \in [n]} \partial_r^2 \ell_i(r_i, u_i) \frac{1}{1 + V \partial_r^2 \ell_i(r_i, u_i)} \frac{1}{1 + V_0 \partial_u^2 \ell_{0,i}(u_i)}$$

$$+ V V_0 \frac{1}{n} \sum_{i \in [n]} \frac{\partial_r^2 \ell_i(r_i, u_i) \partial_r \partial_u \ell_i(r_i, u_i)}{1 + V \partial_r^2 \ell_i(r_i, u_i)} \frac{1}{1 + V_0 \partial_u^2 \ell_{0,i}(u_i)} + \lambda \mathfrak{X} + O_{L_2} \left( \frac{\operatorname{polylog}(n)}{\sqrt{n}} \right)$$

Using Proposition A.5 and Proposition A.14, one can replace the full residuals by the surrogates $\tilde{r}_{i,i}, \tilde{u}_{i,i}$ at the price of a $O_{L_k}(\operatorname{polylog}(n)/\sqrt{n})$ correction:

$$\mathfrak{W} = \chi \frac{1}{n} \sum_{i \in [n]} \partial_r^2 \ell_i(\tilde{r}_{i,i}, \tilde{u}_{i,i}) \frac{1}{1 + V \partial_r^2 \ell_i(\tilde{r}_{i,i}, \tilde{u}_{i,i})} \frac{1}{1 + V_0 \partial_u^2 \ell_{0,i}(\tilde{u}_{i,i})}$$

$$+ V V_0 \frac{1}{n} \sum_{i \in [n]} \frac{\partial_r^2 \ell_i(\tilde{r}_{i,i}, \tilde{u}_{i,i}) \partial_r \partial_u \ell_i(\tilde{r}_{i,i}, \tilde{u}_{i,i})}{1 + V \partial_r^2 \ell_i(\tilde{r}_{i,i}, \tilde{u}_{i,i})} \frac{1}{1 + V_0 \partial_u^2 \ell_{0,i}(\tilde{u}_{i,i})} + \lambda \mathfrak{X} + O_{L_2} \left( \frac{\operatorname{polylog}(n)}{\sqrt{n}} \right)$$

Taking an expectation,

$$\Omega = \chi \mathbb{E} \left[ \partial_r^2 \ell_i(\tilde{r}_{1,1}, \tilde{u}_{1,1}) \frac{1}{1 + V \partial_r^2 \ell_i(\tilde{r}_{1,1}, \tilde{u}_{1,1})} \frac{1}{1 + V_0 \partial_u^2 \ell_{0,i}(\tilde{u}_{1,1})} \right]$$

$$+ V V_0 \mathbb{E} \left[ \frac{\partial_r^2 \ell_i(\tilde{r}_{1,1}, \tilde{u}_{1,1}) \partial_r \partial_u \ell_i(\tilde{r}_{1,1}, \tilde{u}_{1,1})}{1 + V \partial_r^2 \ell_i(\tilde{r}_{1,1}, \tilde{u}_{1,1})} \frac{1}{1 + V_0 \partial_u^2 \ell_{0,i}(\tilde{u}_{1,1})} \right] + \lambda \chi + O \left( \frac{\operatorname{polylog}(n)}{\sqrt{n}} \right).$$

One can verify that all functions appearing on the right-hand side are in the class $\mathcal{F}$. Thus, using Lemma A.26,

$$\Omega - \chi \left( \lambda + \mathbb{E} \left[ \frac{\partial_r^2 \ell(r, u, g_3, c, \epsilon)}{(1 + V\partial_r^2 \ell(r, u, g_3, c, \epsilon))(1 + V_0\partial_u^2 \ell_0(u, g_3, c, \epsilon))} \right] \right)$$
$$- VV_0\mathbb{E} \left[ \frac{\partial_r^2 \ell(r, u, g_3, c, \epsilon)\partial_r\partial_u\ell(r, u, g_3, c, \epsilon)}{(1 + V\partial_r^2 \ell(r, u, g_3, c, \epsilon))(1 + V_0\partial_u^2 \ell_0(u, g_3, c, \epsilon))} \right] = O\left( \frac{\text{polylog}(n)}{n^{\frac{1}{4}}} \right).$$

This completes the proof. $\qquad\qquad\square$

Having established a characterization of all summary statistics, we now detail how they can be used to access a characterization of test metrics, such as the generalization error.

**Lemma A.34.** *Consider any test metric*

$$\mathcal{E}_{\text{gen}} = \mathbb{E}_{x,y,c,\epsilon} \left[ L \left( \langle \hat{w}, x \rangle, s, c, \epsilon \right) \right], \tag{23}$$

*where $L$ is $O(\text{polylog}(n))-$Lipschitz in its first variable, and $s, \epsilon \to L(0, s, c, \epsilon)$ admits a polynomial majorant with $O(1)$ degree and $O(\text{polylog}(n))$ coefficients. In (23), $(x, y, c, \epsilon)$ is a test sample from the same distribution as the training samples, and thus satisfies in particular Assumption A.4. Then $\mathcal{E}_{\text{gen}}$ converges in $L_1$ to*

$$\mathcal{E}_{\text{gen}} = \mathbb{E} \left[ L \left( g_1, g_3, c, \epsilon \right) \right] + O_{L_1} \left( \frac{\text{polylog}(n)}{n^{\frac{1}{4}}} \right).$$

*Proof.* Let us introduce $\tilde{\mathfrak{g}} \sim \mathcal{N}(0_2, \tilde{\mathfrak{S}}), \tilde{g} \sim \mathcal{N}(0_2, \tilde{\Phi})$, where

$$\tilde{\mathfrak{S}} = \begin{bmatrix} \mathfrak{q} & \mathfrak{h} \\ \mathfrak{h} & \varrho \end{bmatrix}, \qquad\qquad\qquad \tilde{\Phi} = \begin{bmatrix} q & \theta \\ \theta & \varrho \end{bmatrix}.$$

We will also need, for $c \in \mathcal{V}$,

$$\tilde{\mathfrak{M}}_c = \begin{bmatrix} \mathfrak{m}_c \\ \nu_c \end{bmatrix}, \qquad\qquad\qquad \tilde{\Psi}_c = \begin{bmatrix} m_c \\ \nu_c \end{bmatrix}.$$

To ease notations, we will also denote $\varphi : \mathbb{R}^2 \times \mathcal{V} \times \mathcal{E} \to \mathbb{R}$, with

$$\varphi(R, c, \epsilon) = L(R_1, R_2, c, \epsilon).$$

Note in particular $\mathbb{E} \left[ \varphi(\tilde{\Psi}_c + \tilde{g}, c, \epsilon) \right] = \mathbb{E} \left[ L \left( g_1, g_3, c, \epsilon \right) \right]$. We are now in a position to start the proof. We have

$$\left| \mathcal{E}_{\text{gen}} - \mathbb{E} \left[ L \left( g_1, g_3, c, \epsilon \right) \right] \right| = \left| \mathbb{E} \left[ \varphi(\tilde{\mathfrak{M}}_c + \tilde{\mathfrak{g}}, c, \epsilon) \right] - \mathbb{E} \left[ \varphi(\tilde{\Psi}_c + \tilde{g}, c, \epsilon) \right] \right|.$$

First notice that $\tilde{g}_2 \overset{d}{=} \tilde{\mathfrak{g}}_2 = g_\varrho$. Conditioning on this component, and using the Lipschitzness of $\varphi$ in its first variable, yields

$$\left| \mathbb{E} \left[ \varphi(\tilde{\mathfrak{M}}_c + \tilde{\mathfrak{g}}, c, \epsilon) \right] - \mathbb{E} \left[ \varphi(\tilde{\Psi}_c + \tilde{g}, c, \epsilon) \right] \right|$$
$$\leq \|L\|_{\text{Lip}} \mathbb{E}_{g_\varrho, \hat{g}} \left[ \left| (\mathfrak{h} - \theta/\varrho)g_\varrho + \left( \sqrt{\mathfrak{q} - \mathfrak{h}^2/\varrho} - \sqrt{q - \theta^2/\varrho} \right) \hat{g} \right| + \sup_c |\mathfrak{m}_c - m_c| \right]$$
$$\leq \|L\|_{\text{Lip}} \mathbb{E}_{g_\varrho, \hat{g}} \left[ \left| (\mathfrak{h} - \theta/\varrho)g_\varrho + \left( \sqrt{\mathfrak{q} - \mathfrak{h}^2/\varrho} - \sqrt{q - \theta^2/\varrho} \right) \hat{g} \right|^2 \right]^{\frac{1}{2}} + \|L\|_{\text{Lip}} \sup_c |\mathfrak{m}_c - m_c|$$
$$\leq \|L\|_{\text{Lip}} \left( \frac{(\mathfrak{h} - \theta)^2}{\varrho} + \left( \sqrt{\mathfrak{q} - \mathfrak{h}^2/\varrho} - \sqrt{q - \theta^2/\varrho} \right)^2 \right)^{\frac{1}{2}} + \|L\|_{\text{Lip}} \sup_c |\mathfrak{m}_c - m_c|$$
$$\leq \|L\|_{\text{Lip}} \left( \frac{(\mathfrak{h} - \theta)^2}{\varrho} + \left| \mathfrak{q} - q - \frac{(\mathfrak{h} - \theta)(\mathfrak{h} + \theta)}{\varrho} \right| \right)^{\frac{1}{2}} + \|L\|_{\text{Lip}} \sup_c |\mathfrak{m}_c - m_c|$$

where $\hat{g} \sim \mathcal{N}(0,1)$. From Lemma A.20 and Lemma A.18,

$$\frac{(\mathfrak{h} - \theta)^2}{\varrho} + |\mathfrak{q} - q| = O_{L_1}\left(\frac{\operatorname{polylog}(n)}{\sqrt{n}}\right).$$

On the other hand, since $\mathfrak{h}, \theta \leq \varrho\sqrt{\|\tilde{e}\|_\infty / \lambda}$ from Lemma A.38, Lemma A.18 guarantees the control

$$|\mathfrak{h} - \theta|(\mathfrak{h} + \theta) = O_{L_2}\left(\frac{\operatorname{polylog}(n)}{\sqrt{n}}\right).$$

Finally, from the concentration result of Lemma A.17,

$$\sup_c |\mathfrak{m}_c - m_c| = O_{L_2}\left(\frac{\operatorname{polylog}(n)}{\sqrt{n}}\right)$$

Therefore,

$$\left|\mathbb{E}\left[\varphi(\tilde{\mathfrak{M}} + \tilde{\mathfrak{g}}, c, \epsilon)\right] - \mathbb{E}\left[\varphi(\tilde{\Psi} + \tilde{g}, c, \epsilon)\right]\right| = O_{L_1}\left(\frac{\operatorname{polylog}(n)}{n^{\frac{1}{4}}}\right)$$

This proves the Lemma. $\qquad\square$

We are now in a position to state the main technical theorem.

**Theorem A.35.** *Consider any test metric*

$$\mathcal{E}_{\text{gen}} = \mathbb{E}_{x,y,c,\epsilon}\left[L\left(\langle\hat{w}, x\rangle, s, c, \epsilon\right)\right],$$

*where $L$ satisfies the conditions of Lemma A.34. Then $\mathcal{E}_{\text{gen}}$ converges in $L_1$ to*

$$\mathcal{E}_{\text{gen}}^* = \mathbb{E}\left[L\left(g_1, g_3, c, \epsilon\right)\right] + O_{L_1}\left(\frac{\operatorname{polylog}(n)}{n^{\frac{1}{4}}}\right),$$

*where $g_1, g_3$ are jointly Gaussian with law $g \sim \mathcal{N}(\Psi_c, \Phi)$ denotes a Gaussian variable with mean and covariance*

$$\Psi_c = \left[\begin{array}{c} m_c \\ m_{0,c} \\ \nu_c \\ \hline \rho_c \end{array}\right], \qquad\qquad \Phi = \left[\begin{array}{ccc|c} q & t & \theta & m \\ t & q_0 & \theta_0 & m_0 \\ \theta & \theta_0 & \varrho & \nu \\ \hline m & m_0 & \nu & \rho \end{array}\right],$$

*conditionally on c. The summary statistics $q, t, \theta, m, m_0, q_0, \theta_0$ verify the self-consistent equations*

$$\theta = -\frac{1}{\lambda}\mathbb{E}\left[\partial_r\ell(r, u, g_3, c, \epsilon)g_3\right] + O\left(\frac{\operatorname{polylog}(n)}{n^{\frac{1}{4}}}\right)$$

$$q = -\frac{1}{\lambda}\mathbb{E}\left[\partial_r\ell(r, u, g_3, c, \epsilon)r\right] + O\left(\frac{\operatorname{polylog}(n)}{n^{\frac{1}{4}}}\right).$$

$$t = -\frac{1}{\lambda}\mathbb{E}\left[\partial_r\ell(r, u, g_3, c, \epsilon)u\right] + O\left(\frac{\operatorname{polylog}(n)}{n^{\frac{1}{4}}}\right)$$

$$m_c = -\frac{1}{\lambda}\mathbb{E}\left[\partial_r\ell(r, u, g_3, c, \epsilon)g_c\right] + O\left(\frac{\operatorname{polylog}(n)}{n^{\frac{1}{4}}}\right)$$

*and*

$$\theta_0 = -\frac{1}{\lambda_0}\mathbb{E}\left[\partial_u\ell_0(u, g_3, c, \epsilon)g_3\right] + O\left(\frac{\operatorname{polylog}(n)}{n^{\frac{1}{4}}}\right)$$

$$q_0 = -\frac{1}{\lambda_0}\mathbb{E}\left[\partial_u\ell_0(u, g_3, c, \epsilon)u\right] + O\left(\frac{\operatorname{polylog}(n)}{n^{\frac{1}{4}}}\right).$$

$$m_{0,c} = -\frac{1}{\lambda_0}\mathbb{E}\left[\partial_u\ell_0(u, g_3, c, \epsilon)g_c\right] + O\left(\frac{\operatorname{polylog}(n)}{n^{\frac{1}{4}}}\right).$$

*We defined the random variables*

$$r = \text{prox}_{V\ell(\cdot,u,g_3,c,\epsilon)}\left(g_1 + \chi\partial_u\ell_0(u, g_3, c, \epsilon), g_3, c, \epsilon\right),$$

$$u = \text{prox}_{V_0\ell_0(\cdot,g_3,c,\epsilon)}(g_2),$$

*and introduced the auxiliary statistics $V, V_0, \chi$ that satisfy*

$$\frac{1}{\alpha V} = \mathbb{E}\left[\frac{\partial_r^2\ell(r, u, g_3, c, \epsilon)}{1 + V\partial_r^2\ell(r, u, g_3, c, \epsilon)}\right] + \lambda + O\left(\frac{\text{polylog}(n)}{n^{\frac{1}{4}}}\right),$$

$$\frac{1}{\alpha V_0} = \mathbb{E}\left[\frac{\partial_u^2\ell_0(u, g_3, c, \epsilon)}{1 + V_0\partial_u^2\ell_0(u, g_3, c, \epsilon)}\right] + \lambda_0 + O\left(\frac{\text{polylog}(n)}{n^{\frac{1}{4}}}\right),$$

$$\chi = \frac{\frac{1}{\alpha}\frac{\mathbb{E}\left[\frac{\partial_r\partial_u\ell(r,u,g_3,c,\epsilon)}{1+V_0\partial_u^2\ell_0(u,g_3,c,\epsilon)}\right]}{\lambda_0 + \mathbb{E}\left[\frac{\partial_u^2\ell_0(u,g_3,c,\epsilon)}{1+V_0\partial_u^2\ell_0(u,g_3,c,\epsilon)}\right]} - \mathbb{E}\left[\frac{\partial_r^2\ell(r,u,g_3,c,\epsilon)\partial_r\partial_u\ell(r,u,g_3,c,\epsilon)V_0V}{(1+V\partial_r^2\ell(r,u,g_3,c,\epsilon))(1+V_0\partial_u^2\ell_0(u,g_3,c,\epsilon))}\right]}{\lambda + \mathbb{E}\left[\frac{\partial_r^2\ell(r,u,g_3,c,\epsilon)}{(1+V\partial_r^2\ell(r,u,g_3,c,\epsilon))(1+V_0\partial_u^2\ell_0(u,g_3,c,\epsilon))}\right]} + O\left(\frac{\text{polylog}(n)}{n^{\frac{1}{4}}}\right).$$

## A.6. Auxiliary lemmas

This subsection regroups for ease of reading miscellaneous auxiliary lemmas that do not directly belong to the main body of the proof of Theorem 1.1.

The first result formalizes the intuition discussed in Section 2 in the main text that the number of acquired samples $\mathfrak{n} = 1/n\sum_{j\in[n]}\pi_j(u_j)$ selected by a policy $\pi$ concentrates in $L_2$, under some smoothness conditions on $\pi$.

**Lemma A.36.** *For any function $\pi : \mathbb{R} \to \mathbb{R}_+$ such that $\pi$ is everywhere twice differentiable in its first argument, and with $\pi, \partial_u\pi, \partial_u^2\pi$ are bounded as $O(\text{polylog}(n))$, the empirical average*

$$\mathfrak{n} = \frac{1}{n}\sum_{j\in[n]}\pi_j(u_j)$$

*concentrates, namely*

$$\text{Var}\left[\mathfrak{n}\right] = O\left(\frac{\text{polylog}(n)}{n}\right).$$

*Proof.* From the Efron-Stein lemma,

$$\text{Var}\left[\mathfrak{n}\right] \leq \sum_{i\in[n]}\mathbb{E}\left[\left(\frac{1}{n}\sum_{j\in[n]}\pi_j(u_j) - \frac{1}{n}\sum_{j\neq i}\pi_j(u_{j,\setminus i})\right)^2\right]$$

$$\leq \frac{3}{n^2}\sum_{i\in[n]}\left[\mathbb{E}\left[\pi_i(u_i)^2\right] + \mathbb{E}\left[\left(\sum_{j\neq i}(\pi_j(\tilde{u}_{j,i}) - \pi_j(u_j))\right)^2\right] + \mathbb{E}\left[\left(\sum_{j\neq i}(\pi_j(u_{j,\setminus i}) - \pi_j(\tilde{u}_{j,i}))\right)^2\right]\right]$$

By assumption, the first term is bounded as

$$3\sum_{i\in[n]}\frac{1}{n^2}\mathbb{E}\left[\pi_i(u_i)^2\right] \leq \frac{3\|\pi\|_\infty}{n}.$$

We now turn to the second term.

$$\left|\sum_{j\neq i}(\pi_j(\tilde{u}_{j,i}) - \pi_j(u_j))\right| \leq \|\partial_u\pi\|_\infty\|\tilde{u}_{,i} - u\|_1 \leq \|\partial_u\pi\|_\infty\sqrt{n}\|\tilde{u}_{,i} - u\|_2 = O_{L_k}\left(\frac{\|\partial_u\pi\|_\infty\sqrt{n}\text{polylog}(n)}{\sqrt{n}}\right).$$

In the last line, we viewed $\tilde{u}_{,i} = (\tilde{u}_{j,i})_{j\neq i}$, $u = (u_j)_{j\neq i}$ as vectors in $\mathbb{R}^{n-1}$ and used Lemma A.6. Hence,

$$\frac{3}{n^2} \sum_{i\in[n]} \mathbb{E}\left[\left(\sum_{j\neq i}(\pi_j(\tilde{u}_{j,i}) - \pi_j(u_j))\right)^2\right] = O\left(\frac{\mathrm{polylog}(n)}{n}\right).$$

We finally turn to the third term. From a Taylor expansion,

$$\left(\sum_{j\neq i}(\pi_j(u_{j,\backslash i}) - \pi_j(\tilde{u}_{j,i}))\right)^2 \leq 2\left(\sum_{j\neq i}\partial_u\pi_j(u_{j\backslash i})(u_{j,\backslash i} - \tilde{u}_{j,i})\right)^2 + \frac{1}{2}\left(\sum_{j\neq i}\partial_u^2\pi_j(\check{u}_{j,i})(u_{j,\backslash i} - \tilde{u}_{j,i})^2\right)^2$$

for some $\check{u}_{j,i} \in (u_{j,\backslash i}, \tilde{u}_{j,i})$. The last term is bounded as

$$\frac{1}{2}\left(\sum_{j\neq i}\partial_u^2\pi_j(\check{u}_{j,i})(u_{j,\backslash i} - \tilde{u}_{j,i})^2\right)^2 \leq \frac{\left\|\partial_u^2\pi\right\|_\infty^2\|\partial_u\ell_0\|_\infty^2}{2}\left(n\sup_{j\neq i}\left|\frac{x_j^\top H_{0,\backslash i}x_i}{n}\right|^2\right)^2.$$

Note that from Lemma C.4 in (El Karoui, 2018),

$$\sup_{j\neq i}\left|\frac{x_j^\top H_{0,\backslash i}x_i}{n}\right| = O_{L_k}\left(\frac{\mathrm{polylog}(n)}{\sqrt{n}}\right).$$

Thus,

$$\frac{1}{2}\left(\sum_{j\neq i}\partial_u^2\pi_j(\check{u}_{j,i})(u_{j,\backslash i} - \tilde{u}_{j,i})^2\right)^2 = O_{L_k}(\mathrm{polylog}(n)).$$

Finally,

$$\sum_{j\neq i}\partial_u\pi_j(u_{j\backslash i})(u_{j,\backslash i} - \tilde{u}_{j,i}) = \left\langle\frac{1}{n}\sum_{j\neq i}\partial_u\pi_j(u_{j\backslash i})\partial_u\ell_{0,i}(\tilde{u}_{i,i})H_{0,\backslash i}^{-1}x_j, x_i\right\rangle$$

Thus

$$\mathbb{E}\left[\left(\sum_{j\neq i}\partial_u\pi_j(u_{j\backslash i})(u_{j,\backslash i} - \tilde{u}_{j,i})\right)^2\right] = \mathbb{E}_{\{x_j\}_{j\neq i}}\left[\left\|\frac{1}{n}\sum_{j\neq i}\partial_u\pi_j(u_{j\backslash i})\partial_u\ell_{0,i}(\tilde{u}_{i,i})H_{0,\backslash i}^{-1}x_j\right\|^2\right]$$

Introducing $h \in \mathbb{R}^{n-1}$ with components $h_j = \partial_u\pi_j(u_{j\backslash i})\partial_u\ell_{0,i}(\tilde{u}_{i,i})$, and $X_{\backslash i} \in \mathbb{R}^{(n-1)\times d}$ the matrix with rows $\{x_j\}_{j\neq i}$, one can compactly rewrite the norm as

$$\left\|\frac{1}{n}\sum_{j\neq i}\partial_u\pi_j(u_{j\backslash i})\partial_u\ell_{0,i}(\tilde{u}_{i,i})H_{0,\backslash i}^{-1}x_j\right\| = \frac{1}{n}\left\|h^\top X_{\backslash i}H_{0,\backslash i}^{-1}\right\|$$

$$\leq \|\partial_u\pi\|_\infty\|\partial_u\ell_0\|_\infty\frac{1}{\lambda_0}\frac{\left\|X_{\backslash i}\right\|}{\sqrt{n}}.$$

Using Lemma A.39 and putting all these results together yields

$$\mathbb{E}\left[\left(\sum_{j\neq i}\partial_u\pi_j(u_{j\backslash i})(u_{j,\backslash i} - \tilde{u}_{j,i})\right)^2\right] = O\left(\mathrm{polylog}(n)\right)$$

and finally that

$$\frac{3}{n^2} \sum_{i \in [n]} \mathbb{E}\left[\left(\sum_{j \neq i} (\pi_j(u_{j,\setminus i}) - \pi_j(\tilde{u}_{j,i}))\right)^2\right] = O\left(\frac{\mathrm{polylog}(n)}{n}\right),$$

implying

$$\mathrm{Var}\,[\mathfrak{n}] = O\left(\frac{\mathrm{polylog}(n)}{n}\right),$$

concluding the proof. □

**Lemma A.37** (Norm of the base estimators). *For any $k \in \mathbb{N}$, the norm of the estimators $\hat{w}_0, \hat{w}_{0,\setminus i}, \tilde{w}_{0,i}$ are bounded as*

$$\|\hat{w}_0\| \leq \sqrt{\frac{1}{\lambda_0}\left\|\tilde{\ell}_0\right\|_\infty},$$

$$\sup_{i \in [n]} \|\hat{w}_{0,\setminus i}\| \leq \sqrt{\frac{1}{\lambda_0}\left\|\tilde{\ell}_0\right\|_\infty},$$

$$\sup_{i \in [n]} \|\tilde{w}_{0,i}\| = O_{L_k}(\mathrm{polylog}(n)).$$

*Proof.* From the definition of $\hat{w}_0$,

$$\frac{\lambda_0}{2}\|\hat{w}_0\|^2 + \frac{1}{n}\sum_{j \in [n]} \ell_{0,j}(u_j) \leq \hat{\mathcal{R}}_0(0_d) = \frac{1}{n}\sum_{j \in [n]} \ell_{0,j}(0).$$

Thus

$$\|\hat{w}\|^2 \leq \frac{1}{\lambda_0}\left\|\tilde{\ell}_0\right\|_\infty.$$

The same bound holds for $\sup_{i \in [n]}\left\|\hat{w}_{0,\setminus i}\right\|$, following the same reasoning. Finally

$$\sup_{i \in [n]} \|\tilde{w}_{0,i}\| \leq \sup_{i \in [n]}\left\|\hat{w}_{0,\setminus i}\right\| + \sup_{i \in [n]}\left\|\frac{1}{n}\partial_u\ell_0(\tilde{u}_{i,i})H_{0,\setminus i}^{-1}x_i\right\|$$

$$\leq \sqrt{\frac{1}{\lambda}\left\|\tilde{\ell}_0\right\|_\infty} + \|\partial_u\ell_0\|_\infty \frac{1}{\lambda_0\sqrt{n}}\sup_{i \in [n]}\frac{\|x_i\|}{\sqrt{n}} = O_{L_k}(\mathrm{polylog}(n)),$$

using Lemma A.40. □

Establishing the equivalent Lemma for the reweighted estimators warrants a finer analysis.

**Lemma A.38** (Norm of the reweighted estimators). *The norm of the estimators $\hat{w}, \hat{w}_{\setminus i}, \tilde{w}_i$ are bounded as*

$$\|\hat{w}\| \leq \sqrt{\frac{1}{\lambda}\left\|\tilde{\ell}\right\|_\infty},$$

$$\sup_{i \in [n]} \|\hat{w}_{\setminus i}\| \leq \sqrt{\frac{1}{\lambda}\left\|\tilde{\ell}\right\|_\infty},$$

$$\sup_{i \in [n]} \|\tilde{w}_i\| = O_{L_k}(\mathrm{polylog}(n)).$$

*Proof.* From the definition of $\hat{w}$,

$$\frac{\lambda}{2}\|\hat{w}\|^2 + \frac{1}{n}\sum_{j \in [n]} \ell_j(r_j, u_j) \leq \hat{\mathcal{R}}(0_d) = \frac{1}{n}\sum_{j \in [n]} \ell_j(0, u_j).$$

Thus

$$\|\hat{w}\|^2 \leq \frac{1}{\lambda}\frac{1}{n}\sum_{j\in[n]}|\ell_j(0,u_j)| \leq \frac{1}{\lambda}\left\|\tilde{\ell}\right\|_\infty.$$

The bound on $\sup_{i\in[n]}\left\|\hat{w}_{\backslash i}\right\|$ follows identically. Finally

$$\sup_{i\in[n]}\|\tilde{w}_i\| \leq \sup_{i\in[n]}\left\|\hat{w}_{\backslash i}\right\| + \sup_{i\in[n]}\|\zeta_i\| + \sup_{i\in[n]}\|\eta_i\| = O_{L_k}(\text{polylog}(n))$$

using Lemma A.10 to bound the second and third term. □

The following Lemmas describe concentration properties of the data. While most of these results are very close to classical result, we state them here for the present setting for self-containedness. For simplicity, we state them for any random variables $z = x - \mu_c$ (see Assumption A.4) satisfying the tail bound

$$\mathbb{P}\left(|\varphi(z) - \mathbb{E}\left[\varphi(z)\right]| > t\right) \leq C_n e^{-c_n t^2}$$

for any convex $1-$Lipschitz function $\varphi$ and any $t > 0$, with $C_n < C, 1/c_n = O(\text{polylog}(n))$, where $C$ is an absolute constant. In the main proof, we apply those results in the Gaussian case of $z \sim \mathcal{N}(0, I_d)$, with $C_n = 2, c_n = 1/2$.

**Lemma A.39** (Operator norm of the empirical covariance). *Let $\hat{\Sigma} = \frac{X^\top X}{n}$ be the empirical covariance matrix. Then*

$$\left\|\hat{\Sigma}\right\| \leq O_{L_k}\left(\frac{\text{polylog}(n)}{c_n}\right).$$

*and*

$$\frac{\|X\|}{\sqrt{n}} \leq O_{L_k}\left(\frac{\text{polylog}(n)}{c_n}\right).$$

*The same bounds hold for the leave-one-row-out matrix $X_{\backslash i}$.*

*Proof.* Let $P \in \mathbb{R}^{n\times d}$ be the matrix with rows $\{\mu_{c_i}\}_{i\in[n]}$. Then $\|X\| = \|P + Z\|$, decomposing the samples into their mean plus random part. Then

$$\|X\| \leq \|P\|_F + \sqrt{\|Z^\top Z\|} \leq \sqrt{n}M + \sqrt{O_{L_k}\left(\frac{n\text{polylog}(n)}{c_n}\right)} = \sqrt{n}O_{L_k}\left(\frac{\text{polylog}(n)}{c_n}\right).$$

We used Lemma G.1 of (El Karoui, 2018) to bound the operator norm of the empirical covariance. □

**Lemma A.40** (Norm of the samples). *One can control the sample norms as*

$$\sup_{i\in[n]}\frac{\|x_i\|}{\sqrt{n}} = O_{L_k}(\text{polylog}(n)).$$

*Furthermore,*

$$\sup_{i\in[n]}\frac{\|X_{\backslash i}\|}{\sqrt{n}} = O_{L_k}(\text{polylog}(n))$$

*Proof.* The lemma follows from a very minor adaptation of (El Karoui, 2018) for data with non-zero mean.

$$\sup_{i\in[n]}\frac{\|x_i\|}{\sqrt{n}} \leq \frac{M}{\sqrt{n}} + \sup_{i\in[n]}\frac{\|z_i\|}{\sqrt{n}} = \frac{M}{\sqrt{n}} + O_{L_k}(\text{polylog}(n)).$$

The last steps follows from (El Karoui, 2018). It remains to control the norm $\sup_{i\in[n]}\|X_{\backslash i}\|/\sqrt{n}$. Note that

$$\sup_{i\in[n]}\frac{\|X_{\backslash i}\|}{\sqrt{n}} = \sqrt{\sup_{i\in[n]}\left\|\hat{\Sigma}_{\backslash i}\right\|} = \sqrt{\sup_{i\in[n]}\left\|\hat{\Sigma} - \frac{x_i x_i^\top}{n}\right\|} \leq \sqrt{\left\|\hat{\Sigma}\right\| + \left(\sup_{i\in[n]}\frac{\|x_i\|}{\sqrt{n}}\right)^2}.$$

The control of the two terms follow from Lemma A.39 and the first point. □

**Lemma A.41.** *(Quadratic forms) Let $M_{\backslash i} \in \mathbb{R}^{d \times d}$ be a matrix which does not depend on $x_i$, but generically depends on all other samples. Then for any $k \in [\lceil 4\log(n) \rceil]$,*

$$\sup_{i \in [n]} \sup_{j \neq i} \left| \frac{1}{n} \langle x_j, M_{\backslash i} x_i \rangle \right| = O_{L_k} \left( \frac{\sup_{i \in [n]} \|M_{\backslash i}\| \mathrm{polylog}(n)}{\sqrt{n}} \right).$$

*Proof.* We first decompose

$$\left| \frac{1}{n} \langle x_j, M_{\backslash i} x_i \rangle \right| \leq \left| \frac{1}{n} \langle \mu_{c_j}, M_{\backslash i} \mu_{c_i} \rangle \right| + \left| \frac{1}{n} \langle \mu_{c_j}, M_{\backslash i} z_i \rangle \right| + \left| \frac{1}{n} \langle z_j, M_{\backslash i} \mu_{c_i} \rangle \right| + \left| \frac{1}{n} \langle z_j, M_{\backslash i} z_i \rangle \right|$$

$$\leq \frac{1}{n} M^2 \|M_{\backslash i}\| + 2M \|M_{\backslash i}\| O_{L_k} \left( \frac{\mathrm{polylog}(n)}{\sqrt{n}} \right) + \left| \frac{1}{n} \langle z_j, M_{\backslash i} z_i \rangle \right|,$$

using Lemma A.40. It thus suffices to control the last term. Introduce

$$\mathsf{L} = \sup_{j \in [n]} \frac{\sup_{i \in [n]} \|M_{\backslash i}\| \|z_j\|}{\sqrt{n}}.$$

Observe that from a union bound

$$\mathbb{P} \left[ \frac{1}{\mathsf{L}} \sup_{i \in [n]} \sup_{j \neq i} \left| \frac{1}{\sqrt{n}} \langle z_j, M_{\backslash i} z_i \rangle \right| > t \right] \leq 1 \wedge \sum_{i \in [n]} \sum_{j \neq i} \mathbb{P} \left[ \frac{1}{\mathsf{L}} \left| \frac{1}{n} \langle z_j, M_{\backslash i} z_i \rangle \right| > t \right]$$

Thus for any $u > 0, k \in \mathbb{N}$ satisfying $k \leq 4\log(n) + 1$, it follows that

$$\mathbb{E} \left[ \left( \frac{1}{\mathsf{L}} \sup_{i \in [n]} \sup_{j \neq i} \left| \frac{1}{\sqrt{n}} \langle z_j, M_{\backslash i} z_i \rangle \right| \right)^k \right] \leq u^k + \sum_{i \in [n]} \sum_{j \neq i} \int_u^\infty k t^{k-1} \mathbb{P} \left[ \frac{1}{\mathsf{L}} \left| \frac{1}{\sqrt{n}} \langle z_j, M_{\backslash i} z_i \rangle \right| > t \right]$$

$$\leq u^k + n(n-1) C_n \int_u^\infty k t^{k-1} e^{-c_n t^2},$$

using the fact that $\varphi : z_i \to {}^1/{\mathsf{L}\sqrt{n}} \langle z_j, M_{\backslash i} z_i \rangle$ is $1-$Lipschitz with $\mathbb{E}[\varphi(x_i)] = 0$, and using Assumption A.4. Note the identity

$$\int_u^\infty k t^{k-1} e^{-c_n t^2} \leq \frac{1}{2c_n} u^{k-2} e^{-c_n u^2} + \frac{k-1}{2 c_n u^2} \int_u^\infty k t^{k-1} e^{-c_n t^2},$$

which follows from an integration by parts. Thus for any $u > \sqrt{k - {}^1/{2c_n}}$,

$$\mathbb{E} \left[ \left( \frac{1}{\mathsf{L}} \sup_{i \in [n]} \sup_{j \neq i} \left| \frac{1}{\sqrt{n}} \langle z_j, M_{\backslash i} z_i \rangle \right| \right)^k \right] \leq u^k + \frac{k n(n-1) C_n}{1 - \frac{k-1}{2 c_n u^2}} \frac{1}{2 c_n} u^{k-2} e^{-c_n u^2}.$$

Choosing $u = \sqrt{\log(n^2)/c_n}$, which satisfies the condition $u > \sqrt{k - {}^1/{2c_n}}$ by assumption on $k$,

$$\mathbb{E} \left[ \left( \frac{1}{\mathsf{L}} \sup_{i \in [n]} \sup_{j \neq i} \left| \frac{1}{\sqrt{n}} \langle z_j, M_{\backslash i} z_i \rangle \right| \right)^k \right] \leq \left( \frac{\log(n^2)}{c_n} \right)^{\frac{k}{2}} \left[ 1 + \frac{k C_n}{2 \log(n^2) - k + 1} \right]$$

$$\leq \left( \frac{\log(n^2)}{c_n} \right)^{\frac{k}{2}} [1 + 4C \log(n)]$$

$$\leq \left( \frac{\log(n^2)(1 + 4C \log(n))^2}{c_n} \right)^{\frac{k}{2}}$$

Thus for any $k \in [[\lceil 4 \log(n) \rceil]]$

$$\sup_{i \in [n]} \sup_{j \neq i} \left| \frac{1}{n} \langle z_j, M_{\backslash i} z_i \rangle \right| = \mathsf{LO}_{L_k}(\mathrm{polylog}(n)) = \sup_{i \in [n]} \|M_{\backslash i}\| O_{L_k} \left( \frac{\mathrm{polylog}(n)}{\sqrt{n}} \right).$$

$\square$

**Lemma A.42** (Sup of one-dimensional projections). *Le $v_{\backslash i}$ be vectors that do not depend on $x_i$, but can generically depend on all other samples. Then for any $k \in [[\lceil 2 \log(n) \rceil]]$,*

$$\sup_{i \in [n]} |\langle v_{\backslash i}, x_i \rangle| = O_{L_k} \left( \sup_{i \in [n]} \|v_{\backslash i}\| \sqrt{\frac{\mathrm{polylog}(n)}{c_n}} \right).$$

*Proof.* Since

$$\sup_{i \in [n]} |\langle v_{\backslash i}, x_i \rangle| \leq M \sup_{i \in [n]} \|v_{\backslash i}\| + \sup_{i \in [n]} |\langle v_{\backslash i}, z_i \rangle|,$$

it is sufficient to bound $\sup_{i \in [n]} |\langle v_{\backslash i}, z_i \rangle|$. We start from the union bound, for any $u > \sqrt{k - 1/2c_n}$,

$$\mathbb{E}\left[ \left( \frac{\sup_{i \in [n]} |\langle v, z_i \rangle|}{\sup_{i \in [n]} \|v_{\backslash i}\|} \right)^k \right] \leq u^k + n \int_u^\infty k t^{k-1} C_n e^{-c_n t^2}$$

$$\leq u^k + \frac{k n C_n}{1 - \frac{k-1}{2 c_n u^2}} \frac{1}{2 c_n} u^{k-2} e^{-c_n u^2}.$$

We choose $u = \sqrt{\log(n)/c_n}$, which is satisfies the condition $u > \sqrt{k - 1/2c_n}$ by assumption on $k$. Then,

$$\mathbb{E}\left[ \left( \frac{\sup_{i \in [n]} |\langle v, z_i \rangle|}{\sup_{i \in [n]} \|v_{\backslash i}\|} \right)^k \right] \leq \left( \frac{\log(n)}{c_n} \right)^{\frac{k}{2}} \left[ 1 + \frac{k C_n}{2 \log(n) - k + 1} \right]$$

$$\leq \left( \frac{\log(n)}{c_n} \right)^{\frac{k}{2}} [1 + 2C \log(n)] \leq \left( \frac{\log(n)(1 + 2C \log(n))^2}{c_n} \right)^{\frac{k}{2}},$$

which completes the proof. $\square$

## A.7. Comments on the multi-stage case

The present works analyzes the problem of two successive ERMs, leveraging the leave-one-out approach. A natural extension would be to consider as in e.g. (Celentano et al., 2025) the multi-stage case

$$\hat{w}_{k+1} = \mathrm{argmin}_w \frac{1}{n} \sum_{i \in [n]} \ell(\langle w, x_i \rangle, \langle \hat{w}_k, x_i \rangle) + \frac{\lambda}{2} \|w\|.$$

We anticipate that this multi-stage scenario can also be studied using similar techniques, and sketch below the expected main steps of the characterization. A full treatment is left to future work.

In the multi-stage case, the statistical correlations with a given sample $x_i$ are compounded over the successive iterations. These correlations introduce additional corrections in the leave-one-out surrogate stage $k + 1$ estimator $\tilde{w}_{k+1,i}$, which we anticipate needs to take the form

$$\tilde{w}_{k+1,i} = \hat{w}_{k+1,\backslash i} - \frac{1}{n} \partial_1 \ell(\tilde{r}_{k+1,i,i}, \tilde{r}_{k,i,i}) H_{k+1,\backslash i}^{-1} x_i$$

$$+ \sum_{m=0}^k (-1)^{k-m} \partial_1 \ell(\tilde{r}_{m,i,i}, \tilde{r}_{m-1,i,i}) \left( H_{k+1,\backslash i}^{-1} G_{k+1,\backslash i} H_{k,\backslash i}^{-1} G_{k,\backslash i} \ldots H_{m+1,\backslash i}^{-1} G_{m+1,\backslash i} \right) H_{m,\backslash i}^{-1} x_i,$$

where

$$H_{m,\backslash i} = \frac{1}{n} \sum_{j \neq i} \partial_1^2 \ell(r_{m,j,\backslash i}, r_{m-1,j,\backslash i}) x_j x_j^\top + \lambda I_d$$

$$G_{m,\backslash i} = \frac{1}{n} \sum_{j \neq i} \partial_1 \partial_2 \ell(r_{m,j,\backslash i}, r_{m-1,j,\backslash i}) x_j x_j^\top$$

$\partial_1 \ell(\tilde{r}_{m,i}, \tilde{r}_{m-1,i})$ is understood as $\partial \ell_0(\tilde{r}_{m,i})$ for $m = 0$. Leveraging this decomposition, the leave-one-out proof is anticipated to proceed along similar lines as the ones expounded in the above proof. In particular, the final expressions should involve effective across-time responses of the form $\chi_{m,k} = \frac{1}{n} \operatorname{tr}\left[H_k^{-1} G_k H_k^{-1} G_k \dots H_{m+1}^{-1} G_{m+1} H_m^{-1}\right]$, that capture how correlations at the level of the $m-$th stage estimator influence those at the level of the $k-$th stage estimator. These quantities generalize the role played by the statistic $\chi$ in the two-stage case of Theorem 1.1.

# B. Special cases

## B.1. Quadratic loss

Theorem 1.1 provides a sharp asymptotic characterization for the test error of the second-stage ERM estimator. Generically, the low-dimensional system of equations needs to be solved numerically, by repeated iteration until convergence, in order to determine the summary statistics involved in the expression of the test error. In some simple settings however, this system simplifies, and a direct closed-form expression for the test error is accessible. In this appendix, we expound such a setting, in the context of the margin-based active binary classification task of Section 2.3 of the main text, with quadratic loss $\ell_0, \ell$.

More precisely, consider the simple case of quadratic loss $\ell_0 = \ell = \ell_{\text{quad.}}$ with $\ell_{\text{quad.}}(y, z) = 1/2(y - z)^2$. Placing ourselves in the case of binary classification, we assume Gaussian covariates $x_i \sim \mathcal{N}(0, I_d)$, and labels $y_i = \text{sign}(\langle \beta, x_i \rangle)$ for some fixed $\beta$. As in Section 2.3, we adopt the selection policy $\pi(u) = H(\kappa - |u|)$, where $\kappa = \kappa(\psi, \gamma)$ is determined by requiring the budget to be saturated in expectation. As mentioned in the main text, the square loss does not satisfy the technical assumptions of Theorem 1.1. However, the analytical expressions are still found to closely match numerical experiments, see for instance Fig. 1.

In this simple case, the equations of Theorem 1.1 for the summary statistics of the base model all admit closed form expressions:

$$\theta_0 = \frac{\psi\sqrt{2/\pi}}{\psi\lambda_0(1 + V_0) + \psi}$$

$$q_0 = \frac{\psi(V_0 - (V_0 - 1)\theta_0\sqrt{2/\pi})}{\psi\lambda_0(1 + V_0)^2 + \psi}$$

$$V_0 = \frac{-(\psi + \psi\lambda_0)\alpha + 1 + \sqrt{((\psi + \psi\lambda_0)\alpha - 1)^2 + 4\psi\lambda_0\alpha}}{2\psi\lambda_0\alpha}$$

Similarly, the summary statistics describing the second-stage estimator admit closed-form expressions:

$$\theta = \frac{-(1 - \psi)tA + (1 - \psi)t\theta_0/q_0 B + (1 - \psi)C + \frac{\psi}{1+V}\left(-\frac{\chi}{1+V_0}(\theta_0 - \sqrt{2/\pi}) + \sqrt{2/\pi}\right)}{\lambda\gamma + \frac{\psi}{1+V} + (1 - \psi)B}$$

$$q = \frac{-\frac{\psi}{(1+V)^2}\left[\frac{2\chi}{1+V_0}t - \frac{2\chi}{1+V_0}\sqrt{\frac{2}{\pi}}\theta + \sqrt{\frac{2}{\pi}}\theta(V - 1) + \frac{\chi^2}{(1+V_0)^2}\left(q_0 + 1 - 2\theta_0\sqrt{\frac{2}{\pi}}\right) + \frac{\chi(V-1)}{1+V_0}\left(\theta_0\sqrt{\frac{2}{\pi}} - 1\right) - V\right]}{\lambda\gamma + \frac{\psi}{(1+V)^2} + (1 - \psi)N}$$

$$- \frac{(1-\psi)\left[\frac{t^2}{q_0^2}M - \frac{t^2}{q_0}N + \left(\frac{t}{q_0} - \frac{\theta_0}{q_0}\frac{\theta - \frac{\theta_0 t}{q_0}}{1 - \frac{\theta_0^2}{q_0}}\right)O + \frac{\theta - \frac{\theta_0 t}{q_0}}{1 - \frac{\theta_0^2}{q_0}}P - VQ\right]}{\lambda\gamma + \frac{\psi}{(1+V)^2} + (1 - \psi)N}$$

$$t = \frac{(1-\psi)F - \frac{\psi}{(1+V)(1+V_0)}\left[V_0\frac{(1-\psi)C + \frac{\psi}{1+V}\left(-\frac{L}{1+V_0}(\theta_0 - \sqrt{2/\pi}) + \sqrt{2/\pi}\right)}{\lambda\gamma + \frac{\psi}{1+V} + (1-\psi)B}\sqrt{2/\pi} + \frac{Lq_0}{1+V_0} + \left(\frac{L(V_0-1)}{1+V_0} - 1\right)\theta_0\sqrt{2/\pi} - \left(\frac{L}{1+V_0} + 1\right)V_0\right]}{\lambda\gamma + \frac{\psi}{(1+V)(1+V_0)} + (1-\psi)\frac{q_0}{\theta_0}A + \frac{\psi}{(1+V)(1+V_0)}\left[K\left(\frac{q_0}{1+V_0} + \frac{V_0-1}{1+V_0}\theta_0\sqrt{\frac{2}{\pi}} - \frac{V_0}{1+V_0}\right) + \frac{-(1-\psi)A + (1-\psi)\theta_0/q_0 B + \frac{\psi}{1+V}\left(-\frac{K}{1+V_0}(\theta_0 - \sqrt{2/\pi})\right)}{\lambda\gamma + \frac{\psi}{1+V} + (1-\psi)B}\right]}$$

$$V = \frac{-\gamma(1 + \lambda)\alpha + 1 + \sqrt{(\gamma(1 + \lambda)\alpha - 1)^2 + 4\gamma\lambda\alpha}}{2\gamma\lambda\alpha}$$

$$\chi = \frac{\frac{1-\psi}{\alpha}\frac{\frac{t}{q_0}G - H}{\psi\lambda_0 + \frac{\psi}{1+V_0}} - (1 - \psi)\left(\frac{t}{q_0}I - J\right)}{\lambda\gamma + \frac{\psi}{(1+V)(1+V_0)} + (1 - \psi)B} = Kt + L$$

We have introduced the shorthands

$$A = \frac{\theta_0}{q_0^2} \mathbb{E}\left[\frac{\pi(g)}{1 + V\pi(g)}g^2\right]$$

$$B = \mathbb{E}\left[\frac{\pi(g)}{1 + V\pi(g)}\right] = \frac{\gamma - \psi}{1 - \psi}\frac{1}{1 + V}$$

$$P = \mathbb{E}\left[\frac{\pi(g)}{1 + V\pi(g)}(V\pi(g) - 1)\left(\frac{\theta_0}{q_0}\mathrm{gerf}\left(\frac{\theta_0/q_0 g}{\sqrt{2(1 - \theta_0^2/q_0)}}\right) + \sqrt{\frac{2}{\pi}}\sqrt{1 - \theta_0^2/q_0}e^{-\frac{1}{2}\frac{\theta_0^2/q_0^2 g^2}{1 - \theta_0^2/q_0}}\right)\right]$$

$$C = \mathbb{E}\left[\frac{\pi(g)^2}{1 + V\pi(g)}\left(\frac{\theta_0}{q_0}\mathrm{gerf}\left(\frac{\theta_0/q_0 g}{\sqrt{2(1 - \theta_0^2/q_0)}}\right) + \sqrt{\frac{2}{\pi}}\sqrt{1 - \theta_0^2/q_0}e^{-\frac{1}{2}\frac{\theta_0^2/q_0^2 g^2}{1 - \theta_0^2/q_0}}\right)\right]$$

$$O = \mathbb{E}\left[\frac{\pi(g)^2 g}{1 + V\pi(g)}(V\pi(g) - 1)\mathrm{erf}\left(\frac{\theta_0/q_0 g}{\sqrt{2(1 - \theta_0^2/q_0)}}\right)\right]$$

$$F = \mathbb{E}\left[\frac{\pi(g)g}{1 + V\pi(g)}\mathrm{erf}\left(\frac{\theta_0/q_0 g}{\sqrt{2(1 - \theta_0^2/q_0)}}\right)\right]$$

$$G = \mathbb{E}\left[\frac{\pi'(g)g}{1 + V\pi(g)}\right]$$

$$H = \mathbb{E}\left[\frac{\pi'(g)}{1 + V\pi(g)}\mathrm{erf}\left(\frac{\theta_0/q_0 g}{\sqrt{2(1 - \theta_0^2/q_0)}}\right)\right]$$

$$I = \mathbb{E}\left[\frac{\pi(g)\pi'(g)g}{(1 + V\pi(g))^2}\right]$$

$$J = \mathbb{E}\left[\frac{\pi(g)\pi'(g)}{(1 + V\pi(g))^2}\mathrm{erf}\left(\frac{\theta_0/q_0 g}{\sqrt{2(1 - \theta_0^2/q_0)}}\right)\right]$$

$$K = \frac{\frac{1 - \psi}{\alpha}\frac{\frac{1}{q_0}G}{\psi\lambda_0 + \frac{\psi}{1 + V_0}} - (1 - \psi)\frac{VV_0}{q_0}I}{\lambda\gamma + \frac{\psi}{(1 + V)(1 + V_0)} + (1 - \psi)B}$$

$$L = \frac{\frac{1 - \psi}{\alpha}\frac{-H}{\psi\lambda_0 + \frac{\psi}{1 + V_0}} + (1 - \psi)VV_0 J}{\lambda\gamma + \frac{\psi}{(1 + V)(1 + V_0)} + (1 - \psi)B}$$

$$M = \mathbb{E}\left[\frac{\pi(g)}{(1 + V\pi(g))^2}g^2\right]$$

$$N = \mathbb{E}\left[\frac{\pi(g)}{(1 + V\pi(g))^2}\right]$$

$$Q = \mathbb{E}\left[\frac{\pi(g)^2}{(1 + V\pi(g))^2}\right].$$

In the above display, expectations bear over the scalar Gaussian variable $g \sim \mathcal{N}(0, q_0)$. Taken together, these equations provide a closed-form expression for the test error.

## B.2. Sample-splitting

The expressions of Theorem 1.1 also simplify if one assumes the first and second ERM are performed on *disjoint* datasets, an assumption made in e.g. (Kolossov et al., 2023; Sorscher et al., 2022; Askari-Hemmat et al., 2025; Dohmatob et al., 2025). This sample-splitting scenario is a special case of the problem considered in the present manuscript, and corresponds to introducing Bernoulli class labels $c_i \sim \mathcal{B}(p)$ for every sample, and using sample $x_i$ in the first (resp. second) ERM when $c_i = 0$ (resp. $c_i = 1$). For the purpose of clarity, let us assume in this subsection there exists no other class structure (e.g. in clusters) that superimposes onto this partition. This split of the training set can be effectively enforced by choosing losses of

the form

$$\ell_0(u, s, c, \epsilon) = (1 - c)\tilde{\ell}_0(u, s, \epsilon), \qquad\qquad \ell(r, u, s, c, \epsilon) = c\tilde{\ell}(r, u, s, \epsilon),$$

so that the $i-$th individual loss term is active only on the relevant subset. The equations for the summary statistics $q, t, \theta, m, m_0, q_0, \theta_0$ then simplify as

$$\theta = -1/\lambda(1 - p)\mathbb{E}\left[\partial_r\tilde{\ell}(r, g_2, g_3, \epsilon)g_3\right] + \tilde{O}(n^{-1/4}),$$

$$q = -1/\lambda(1 - p)\mathbb{E}\left[\partial_r\tilde{\ell}(r, g_2, g_3, \epsilon)r\right] + \tilde{O}(n^{-1/4}),$$

$$t = -1/\lambda(1 - p)\mathbb{E}\left[\partial_r\tilde{\ell}(r, g_2, g_3, \epsilon)g_2\right] + \tilde{O}(n^{-1/4}),$$

$$m_c = -1/\lambda(1 - p)\mathbb{E}\left[\partial_r\tilde{\ell}(r, g_2, g_3, \epsilon)g_c\right] + \tilde{O}(n^{-1/4}),$$

and

$$\theta_0 = -1/\lambda_0 p\mathbb{E}\left[\partial_u\tilde{\ell}_0(u, g_3, c, \epsilon)g_3\right] + \tilde{O}(n^{-1/4}),$$

$$q_0 = -1/\lambda_0 p\mathbb{E}\left[\partial_u\tilde{\ell}_0(u, g_3, c, \epsilon)u\right] + \tilde{O}(n^{-1/4}),$$

$$m_{0,c} = -1/\lambda_0 p\mathbb{E}\left[\partial_u\tilde{\ell}_0(u, g_3, c, \epsilon)g_c\right] + \tilde{O}(n^{-1/4}).$$

We defined the random variables

$$r = \text{prox}_{V\tilde{\ell}(\cdot, g_2, g_3, c, \epsilon)}(g_1),$$

$$u = \text{prox}_{V_0\tilde{\ell}_0(\cdot, g_3, c, \epsilon)}(g_2).$$

This recovers equations reached under prior works under the sample-splitting assumption. Remarkably, the equations of the summary statistics $q, m, t, \theta$ associated to the second-stage estimator no longer involve the residual $u$ of the first ERM, but the simpler Gaussian variable $g_2$, which now plays a comparable role to the label pre-activation $g_3$. This simplification directly arises from the decoupling of the correlations between the first estimator and the samples contained in the second training set.

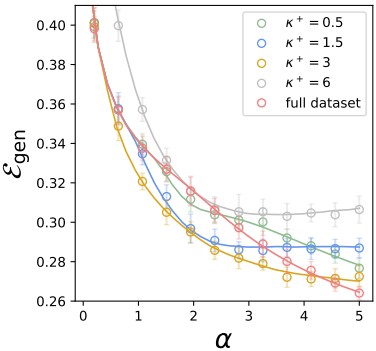

*Figure 5.* Test error , as a function of the normalized number of samples $\alpha = n/d$, of a logistic classifier ($\tilde{\ell}(y,z) = \log(1 + e^{-yz})$, $\lambda = 0.01$) trained on a subset $\mathcal{S} \subset [n]$ pruned using a large-margin selection policy $\pi(u) = \mathbb{1}_{|u| > \kappa^+}$. The base estimator was trained on the whole dataset ($\mathcal{S}_0 = [n]$), also with $\tilde{\ell}(y,z) = \log(1 + e^{-yz})$, $\lambda = 0.01$. The data is a binary isotropic Gaussian mixture with centroid norm $\|\mu\| = 0.8$. Different curves represent different thresholds $\kappa^+$, with higher values implying smaller training subsets $|\mathcal{S}|$. Solid lines: theoretical predictions of Theorem 1.1. Dots: numerical experiments in dimension $d = 2000$. Error bars represent one standard deviation over 30 trials.

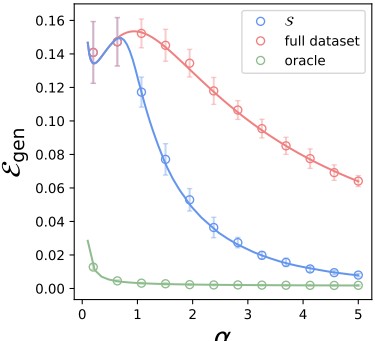

*Figure 6.* Test error of a logistic classifier ($\tilde{\ell}(y,z) = \log(1 + e^{-yz})$, $\lambda = 0.01$) trained on a subset $\mathcal{S}$ constructed with a base estimator with the same loss and regularization, trained on the whole dataset ($\mathcal{S}_0 = [n]$), as a function of the normalized number of samples $\alpha = n/d$. The data is a binary isotropic Gaussian mixture centered at $\pm\mu$, with centroid norm $\|\mu\| = 0.8$, and class labels $c_i \in -1, +1$. The labels are corrupted by a random flipping noise $\epsilon \in \{-1, +1\}$, which with $\mathbb{P}(\epsilon = -1) = 0.2$, as $y_i = \epsilon_i c_i$. The selection policy $\pi(u, c, \epsilon) = \mathbb{1}_{\text{sign}(u)=c\epsilon}$ only retains samples correctly classified by the base estimator. Solid lines: theoretical predictions of Theorem 1.1. The green line represents the test error of a oracle classifier trained only on samples whose labels were not flipped. Dots: numerical experiments in dimension $d = 2000$. Error bars represent one standard deviation over 30 trials.

## C. Comments on data pruning

In this Appendix, we depart from the active learning framework of Section 2, in order to connect with and revisit some observations of related prior works (Sorscher et al., 2022; Kolossov et al., 2023; Askari-Hemmat et al., 2025) in the setting where the base training set $\mathcal{S}_0$ is re-used for training the final classifier, instead of being held-out. In this section only, we assume all labels are available, and the base classifier is trained on the whole dataset, namely $\mathcal{S}_0 = [n]$, then used to construct a final and *smaller* training set $\mathcal{S} \subset [n]$ through a policy $\pi$. Several of the above works report settings in which training a model on a smaller subset $\mathcal{S} \subset [n]$ can yield *better* accuracy than training on the full dataset. These conclusions hold in some cases even accounting for the data used in the held-out dataset. We show through two examples that these observations can also be made in the more realistic setting where the base classifier is trained on the whole dataset, then used to prune it, namely without assuming an oracle classifier or a held-out dataset. Once more, these analyses are enabled by Theorem 1.1, which characterizes iterated ERMs with data re-use.

**Example 1 —** Fig. 5 illustrates the test error achieved by logistic regression trained on a binary Gaussian mixture classification task with only a subset of the full training set, keeping only larger margin samples using a policy $\pi(u) = \mathbb{1}_{|u| > \kappa^+}$. Interestingly, strategies with $\kappa^+ > 0$ can outperform the model trained on the full dataset (red), despite using less

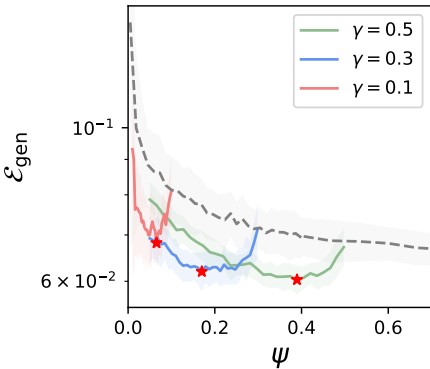

*Figure 7.* Test error as a function of the fraction $\psi \in (0, \gamma)$ allocated to the first stage of ERM, for the pneumonia diagnosis on the chest X-ray dataset (Kermany et al., 2018), pre-processed through a 3-layer neural network feature map pre-trained on 1000 held-out samples. The logistic loss $\ell(z, y) = \ell_0(z, y) = \log(1 + \exp(-yz))$ was employed, with $\alpha = 1.5, \lambda = 0.01$. The samples closest to the decision boundary of the first classifier were retained, until the budget was saturated. Different curves represent different total budget $\gamma$. Shades represent one standard deviation over 30 trials.

data. Intuitively, large margin samples contain clear information on the centroid of the mixture, affording the model a easy grasp on the data structure. This observation echoes closely related findings in (Sorscher et al., 2022; Kolossov et al., 2023; Hacohen et al., 2022) for a different task.

**Example 2 —** Fig. 6 also illustrate a binary Gaussian mixture classification task with logistic regression. The labels $c_i = \pm 1$, indicating the cluster membership, were corrupted by a flipping noise $\epsilon_i = \pm 1$ with probability $\mathbb{P}(\epsilon = -1) = 0.2$, complicating the task. Directly fitting a classifier on the whole dataset yields mediocre accuracy (red). In order to mitigate the effect of the noise, it is possible to leverage a data pruning strategy as a mean of regularization: a base estimator is trained on the whole dataset, and the samples it incorrectly classifies are discarded. This selection process is motivated by the intuition that a large fraction of misclassified samples are, in fact, flipped. The final classifier is trained on the resulting pruned subset $\mathcal{S}$ (blue), and displays improved accuracy. This observation was also made by (Feng et al., 2024) in a simpler setting.

It is seminal to nuance that all these observations, and those reported in prior works (Sorscher et al., 2022; Kolossov et al., 2023; Feng et al., 2024; Dohmatob et al., 2025), have been made for a fixed model choice $\tilde{\ell}, \lambda$. A precise study of whether and when these phenomena still hold true when for instance the regularization $\lambda$ is optimized over is still very largely missing, and an important direction for future investigations.

## D. Additional experiments

Fig. 2 in the main text illustrated how the budget allocation tradeoff could also be observed for real datasets, such as the pneumonia diagnosis task on chest X-ray images (Kermany et al., 2018). In Fig. 2, images were processed through a scattering transform. In this Appendix, we complement this illustration by reproducing the experiment in the case where the images are rather processed by a neural network feature map. In the experiment, a 3-layer fully-connected network with hidden widths 2000 and ReLU activation was trained for 50 epochs with full-bacth Adam with default `Pytorch` (Paszke et al., 2019) parameters , on 1000 held-out samples. A new training set was then constituted out by selecting randomly $n = 3000$ points from the remaining samples, and pre-processed through the learned neural feature map. The same experiment as in Fig. **??** was then performed. Fig. 7 represents the test error for different initial budget allocations $\psi$. Learning curves also exhibit a marked $U-$shape, a manifestation of the budget allocation tradeoff observed in Fig. 2 for a different preprocessing.

