# OpenReview forum: "Asymptotic Theory of Iterated Empirical Risk Minimization, with Applications to Active Learning"
_ICML.cc/2026/Conference — ICML 2026 regular_

### Official Review · Reviewer_aDpw · 2026-02-28

**Soundness:** 3
**Presentation:** 2
**Significance:** 3
**Originality:** 3
**Overall Recommendation:** 4
**Confidence:** 4

**Summary:**

This paper consider the problem of two successives ERM procedures where the second depends on the first model's output, creating dependencies due to data reuse. The authors provided a high-dimensional asymptotic prediction of the second-stage test error for linear models with convex losses under guaussian mixtures. The results are further  applied to active learning, revealing a tradeoff in label allocation, and uncovering a double-descent effect driven by data selection.

**Compliance With Llm Reviewing Policy:**

Affirmed.

**Final Justification:**

The work is technically sound and original especially in its analysis of two-stage ERM under data reuse and its implications for active learning. My concerns  were fully addressed during the rebuttal, which reinforced my original positive review, and i encourage the author to incorporate these clarifications into the revised version.

**Key Questions For Authors:**

(1) The study considers a specific data distribution.  Can the analysis be easily extended to more general data distributions?


(2) The Theorem states the test metric converges (in L1) towards a quantity. Is this quantity the  optimal test metric? is there any relationship between the loss L  and $\ell$  (or $\ell_0$ ) ?

(3) Can the theoretical analysis be easily extended to SVMs (hinge loss)?

**Limitations:**

Yes

**Strengths And Weaknesses:**

Soundness: The paper provides a theoretical analysis with comparaisons to prior works and applies the result to the active learning setting. The analysis are supported by experiments, which illustrate phenomena such as double descent.

Presentation: the paper is generally well-written. However, Section 1 is particularly dense and difficult to follow. The notation is occasionally inconsistent (e.g the test metric is denoted in the same way as the quantity it is supposed to converge to,  \tilde{O}_{L_1}, prox are not explained), some assumption are not discussed. This Section would benefit from further clarification and polishing.

Significance: the work addresses a relevant problem with important implications for active learning, particularly by enabling analysis without relying on restrictive assumptions used in prior work.

Originality: The work provides new theoretical insights into two-stage ERM with data reuse. It also reveals novel phenomena in active learning, includidng a label allocation tradeoff and a data-driven double-descent effect.

---

> ### Author Rebuttal · Authors · 2026-03-31
>
> We thank the reviewer for their appreciation of our manuscript, and insightful suggestions. We answer their questions below.
>
> > However, Section 1 is particularly dense and difficult to follow. The notation is occasionally inconsistent (e.g the test metric is denoted in the same way as the quantity it is supposed to converge to, $\tilde{O}_{L_1}$, prox are not explained), some assumption are not discussed. This Section would benefit from further clarification and polishing.
>
> We thank the reviewer for pointing out these confusing items, which we will fix in the revised version. For two random variables $x,y$, the notation $x=O_{L_1}(y)$ signifies $\mathbb{E}(|x|)\le \mathbb{E}(|y|)$. The proximal $\mathrm{prox}_\ell(x)$ operator of a convex function $\ell$ evaluated at $x$ is the argmin over $z$ of $(z-x)^2+\ell(z)$. We will make use of the extra space allowed for the final version to include further clarifications of such technical points in Section 1.
>
>
> >(1) The study considers a specific data distribution. Can the analysis be easily extended to more general data distributions?
>
> We thank the reviewer for the very relevant question. Indeed, the leave-one-out proof technique we employ is not as sensitive to the data distribution as the Gaussian min-max approach, for instance. We thus expect that the results extend to a larger class of distribution, satisfying a subgaussian concentration property for Lipschitz convex functions of the data, see for instance El Karoui, 2018, assumption O.4. We besides anticipate the possibility to accommodate elliptical data of the form $\sigma z$, where $z$ is vector satisfying the above properties, and $\sigma$ is a scalar random variable with finite moments up to the fourth (see e.g. El Karoui, 2018, or Barnfield et al, 2025). We will highlight these possible generalizations in the revised version.
>
>
> > (2) The Theorem states the test metric converges (in L1) towards a quantity. Is this quantity the optimal test metric? is there any relationship between the loss L and $\ell_0$  (or  $\ell$) ?
>
> To be as general as possible, we allowed $L$ to be any function that satisfies the assumptions of Lemma A.34, namely $L$ is $O(polylog(n))-$Lipschitz in its first variable, and $s,\epsilon\to L(0,s,c,\epsilon)$ admits a polynomial majorant with $O(1)$ degree and $O(polylog(n))$ coefficients. We do not require any relationship between the test metric $L$ and the losses $\ell, \ell_0$.
>
>
> > (3) Can the theoretical analysis be easily extended to SVMs (hinge loss)?
>
> We believe the results of Theorem 1.1 also describe SVMs, although a rigorous proof of this fact presents some technical subtleties.
> Because the hinge loss is not differentiable, it does not directly satisfy the assumptions of Theorem 1.1. Note on the other hand that the logistic loss does, and also yields the max-margin solution in the limit of vanishing regularization. One could also consider approximations of the hinge loss obtained by smoothening a $O(1/\mathrm{polylog}(n))$ neighborhood of $0$, following standard mollification arguments, that fall in the range of applicability of the Theorem.

---

> > ### Author Rebuttal · Reviewer_aDpw · 2026-04-02
> >
> > Thank you for the clarifications, all my concerns have been addressed. I encourage the authors to incorporate these clarifications into the revised version and i will maintain my current score.

---

### Official Review · Reviewer_Wi8h · 2026-03-05

**Soundness:** 3
**Presentation:** 3
**Significance:** 3
**Originality:** 3
**Overall Recommendation:** 4
**Confidence:** 3

**Summary:**

The paper mainly studied the theory related to iterated empirical risk minimization. In particular, the authors established the asymptotic bound for some test metric associate with the second stage estimator under some conditions on the metric. Then the authors apply this general theory to margin-based binary classification.

**Compliance With Llm Reviewing Policy:**

Affirmed.

**Key Questions For Authors:**

1) Since you mentioned that the theories are built upon scenarios like mixture of Gaussian distributions, I am wondering why you cannot develop non-asymptotic bound on the test metric. I think a nonasymptotic bound may be more desirable than this asymptotic bound. If this is not feasible, please clarify the technical difficulties.

2) The high dimensional regime considered here is $n \asymp d$. Can you develop similar theories in more challenging regimes, say $d \asymp \exp{n^{\alpha}$ for some positive $\alpha$? Such settings are pretty common in high dimensional linear regression with some proper regularization such as LASSO.

**Limitations:**

1) In your Section 1, you mentioned that $c_i$ is a latent variable. Thus I thought that these $c_i$'s are not observable. But it seems that you use $c_i$ to estimate $w_0$ and $w_1$ in equations (3) and (4), respectively. Please clarify this.

2) Given a real dataset and an active learning problem with a certain budget constrain, could you provide a guideline in terms of choosing policy $\pi$? Finding an optimal policy might be challenging. But a "good" guideline should be helpful. Otherwise, we may question what is the benefit we can gain from taking this theory of iterated empirical risk minimization into consideration.

**Strengths And Weaknesses:**

Overall speaking, the paper provides a clear description of the asymptotic theory of the iterative empirical risk minimization under general high dimensional settings. The theories developed in the work are solid and the presentation is clear. The theory may find its application in active learning, though I personally do not think it is so important in general machine learning settings.

---

> ### Author Rebuttal · Authors · 2026-03-31
>
> We thank the reviewer for their reading of our manuscript and constructive comments. We address their questions below.
>
> >Since you mentioned that the theories are built upon scenarios like mixture of Gaussian distributions, I am wondering why you cannot develop non-asymptotic bound on the test metric. I think a nonasymptotic bound may be more desirable than this asymptotic bound. If this is not feasible, please clarify the technical difficulties.
>
> We indeed chose to set our analysis in the high-dimensional regime $n/d=O(1)$ as $n,d\to\infty$. This regime is classical in the high-dimensional statistics literature, and constitutes the setting of many prior analyses of this problem, including e.g. (Kolossov et al., 2023; Sorscher et al., 2022; Askari Hemmat et al., 2025; Feng et al., 2024; Dohmatob et al.,
> 2025), under stronger oracle assumptions. We note that Theorem 1.1 does give non-asymptotic $O(polylog(n) n^{-1/4})$ bounds on the error terms, although this rate is not necessarily tight.
>
> >The high dimensional regime considered here is $n\asymp d$. Can you develop similar theories in more challenging regimes, say $d \asymp n^\alpha$ for some alpha $\alpha>0$? Such settings are pretty common in high dimensional linear regression with some proper regularization such as LASSO.
>
> The study of polynomial scaling regimes $n\asymp d^\alpha$ (beyond the linear regime we considered) would indeed constitute an extremely interesting research direction. As evidenced by several recent works, such regimes display highly non-trivial behaviors [a]. Their exploration however pose several sizable technical challenges, that we leave for future works.
>
> [a] Wen et al, When does Gaussian equivalence fail and how to fix it: Non-universal behavior of random features with quadratic scaling, 2025
>
> >In your Section 1, you mentioned that $c$ is a latent variable. Thus I thought that these c's are not observable. But it seems that you use  to estimate $w_0, w_1$  in equations (3) and (4), respectively. Please clarify this.
>
>  We apologize for the confusion, and will remove the reference to $c$ as a "latent" variable. In fact, in the case of Gaussian mixture classification, $c$ would correspond to the cluster label, and will thus play the role of the label in the learning problem. It is therefore indeed a variable accessible to the statistician.
>
> > Given a real dataset and an active learning problem with a certain budget constrain, could you provide a guideline in terms of choosing policy ? Finding an optimal policy might be challenging. But a "good" guideline should be helpful. Otherwise, we may question what is the benefit we can gain from taking this theory of iterated empirical risk minimization into consideration.
>
> Fig. 4 displays the theoretical predictions for a family of policies, and evidences that the best policy does not necessarily consist in retaining samples closest to the decision boundary, nuancing a widespread intuition. As we discuss in subsection 2.4, the theoretical expressions would in principle allow to optimize over the policy $\pi$ in order to ascertain the optimal scheme. While this is generically hard for the selection problem, where $\pi$ is binary, we believe it is tractable in the related problem of subsampling, where $\pi$ is valued in $(0,1)$ and can be easily optimized over. This constitutes an interesting direction for future work, which we will discuss in the revised version. Note that even if it does not provide any ready-for-use practical guideline, our theoretical work clarifies the role of intricate statistical effects (such as data re-use) in data selection, and contributes to the theoretical comprehension of this class of problems, which we believe is just as crucial.

---

> > ### Author Rebuttal · Reviewer_Wi8h · 2026-04-03
> >
> > The authors have partially addressed my concerns. 1) In theory it is important, but probably challenging to establish theories under high dimensional scenarios like $d \asymp n^{\alpha}$ or ultra high-dimensional regimes.
> > 2) I do not think using mixture models for clustering analysis, you can observe cluster label. This can be estimated but not directly observable.

---

> > > ### Author Response · Authors · 2026-04-08
> > >
> > > > 1) In theory it is important, but probably challenging to establish theories under high dimensional scenarios like n\sim d^\alpha or ultra high-dimensional regimes.
> > >
> > > We thank the reviewer for this important question. For the model and task considered in the present work, $n\sim d^c$ regimes are captured by taking the limit $\alpha\to0$ (resp. $\alpha\to\infty$) when $c\in(0,1)$ (resp $c>1$) in Theorem 1.1. On the other hand, we fully agree that such regimes would be non-trivial for instance for kernel regression or classification, and constitute important directions for future research.
> > >
> > > >2) I do not think using mixture models for clustering analysis, you can observe cluster label. This can be estimated but not directly observable.
> > >
> > > We thank the reviewer for highlighting this important distinction, which can be confusing: we did not mean unsupervised Gaussian Mixture Model (GMM) _clustering_. We instead refered to GMM _classification_, which is an extensively studied theoretical class of synthetic data used to model supervised classification tasks (see e.g. Loureiro et al, Learning Gaussian Mixtures with Generalised Linear Models: Precise Asymptotics in High-dimensions, NeurIPS 2021). In this model of data, covariates are drawn from a GMM. The supervised classification task then consists in learning the cluster assignments, which for this task constitute the labels, and are observable.

---

### Official Review · Reviewer_Xo3d · 2026-03-13

**Soundness:** 3
**Presentation:** 3
**Significance:** 3
**Originality:** 3
**Overall Recommendation:** 5
**Confidence:** 3

**Summary:**

This paper evaluates two-stage minimization of an empirical risk, performing the two steps
$$
\\hat{w}_0 = \\arg \\min \\frac{1}{n} \\sum_i^n \\ell_0(\\langle w, x_i \\rangle, y_i) + \\frac{\\lambda_0}{2} ||w||^2
$$
and then
$$
\\hat{w} = \\arg \\min \\frac{1}{n} \\sum_i \\ell(\\langle w, x_i \\rangle, \\langle \\hat{w}_0, x_i, \\rangle, y_i) + \\frac{\\lambda}{2} ||w||^2
$$
They give an asymptotic analysis in the proportional scaling limit that $d/n \\to c$ for a constant $c$.

The authors show an asymptotic message-passing type generalization guarantee (in Theorem 1.1), and then apply the results they have derived to certain active learning and semi-supervised scenarios.

**Compliance With Llm Reviewing Policy:**

Affirmed.

**Final Justification:**

The rebuttal reinforced my prior assessment and I believe the paper merits acceptance.

While we have no enforcement mechanism, I hope the authors take my commentary about presentation, framing, and clarity seriously in revising the paper.

**Key Questions For Authors:**

See my review above.

**Limitations:**

It's a theory paper. It has the usual limitations that it's not always clear how this actually makes predictions in real life or practice. That's fine--theory papers should be theory papers.

**Strengths And Weaknesses:**

Presentation:
The presentation seemed reasonable to me, though there were a few niggles that I had, which also relate to soundness:
1. Why do the limiting terms in Theorem 1.1 involve anything depending on $n$? Given that we take $n \\to \\infty$, should these not be all 0?
2. Relatedly, in Theorem 1.1, what does the dependence on the class index $c$ in the definition of $\\Psi_c$ and the gaussian $g \\sim \\mathcal{N}(\\Psi_c, \\Phi)$ mean? Because it seems that $c$ was a random class label (to be used to indicate whether it is a labeled data point or not, i.e., included in the supervised data), so is this sampling conditional on $c$? Or does this mean that it depends on the distribution of $c$? Or am I missing something? (To be clear: the appendix was some 50 pages, and I did not read it. While I am, in broad strokes, familiar with AMP and high-dimensional scaling limits and their techniques, it seems unreasonable to expect a reviewer to handle 50 pages of appendices.)
3. It would be nice if the authors could give a simplified view of their results in *any* setting at all. because as written, they are difficult to parse. I recognize that AMP and friends often yield such implicit limits as is the nature of the analyses. Yet some type of indication of the types of predictions this makes could be useful, even in very simple settings.
4. The authors claim their results make predictions, but it is unclear how to actually evaluate the predicted behavior. If they could elaborate on that it would be nice (related ot point 3).

Soundness:
See above.

Originality:
1. Broadly, fine. It is a natural idea to perform a large-sample analysis of a data selection scheme, but (as far as I know) no one had done it.
2. I have some issues with the framing (this could plausibly be a presentation feedback). The authors claim that "classical optimization algorithms" are a sequential ERM. This is true, but at the end of the day, we know that what we obtain at the end of the procedure is not a two-stage solution but instead a (regularized) empirical risk minimizer: these procedures are intermediaries that we typically would not analyze. Moreover, other papers, e.g., the original AMP papers and the cited paper of Celentano, do a much better job tracking sequential procedures and randomness than the current paper. So I think this framing is misleading. Similarly, I think including AdaBoost is mis-leading. I would remove this to more honestly reflect the main contribution of the paper: item 3 in the introduction on uncertainty-based active learning is the appropriate (and frankly, good enough) motivation.

---

> ### Author Rebuttal · Authors · 2026-03-31
>
> We thank the reviewer for their appreciation of our work and many insightful questions, which we address below.
>
> > Why do the limiting terms in Theorem 1.1 involve anything depending on $n$? Given that we take $n\to\infty$, should these not be all 0?
>
> We thank the reviewer for pointing out this confusing expression. In the considered limit, $n/d$ is assumed to converge to a strictly positive limit $\alpha$ of order one. This ratio can thus be replaced by $\alpha$ in all equations of Theorem 1.1, and none of the terms in fact vanishes.
>
> >Relatedly, in Theorem 1.1, what does the dependence on the class index  in the definition of $\Psi_c$ and the gaussian  mean? Because it seems that  was a random class label (to be used to indicate whether it is a labeled data point or not, i.e., included in the supervised data), so is this sampling conditional on $c$?
>
> The reviewer is correct that this subscript indicates that the distribution of $g\sim\mathcal{N}(\Psi_c,\Phi)$ is stated conditionally on $c$, as we mention below the expressions of $\Psi_c,\Phi$ in Theorem 1.1. This means that expectations can be carried out by taking an outer average over the finite number of values that can be taken by the discrete random variable $c$, with the inner expectation over $g$ involving the conditional mean $\Psi_c$. We remind that the $c$ is valued in a set with finite cardinal as the high-dimensional limit is taken: therefore, if there are $|\mathcal{V}|=3$ classes for instance, there are $3$ mean vectors $\Psi_c$ involved in the final expressions.
>
> >It would be nice if the authors could give a simplified view of their results in any setting at all. because as written, they are difficult to parse. I recognize that AMP and friends often yield such implicit limits as is the nature of the analyses. Yet some type of indication of the types of predictions this makes could be useful, even in very simple settings.
>
> In the case of the square loss function, the expressions considerably simplify, and a closed-form expression of the test error can be reached. We will include this special case in the revised manuscript. Furthermore, we will provide more discussion of the special case for which the training sets of the first and second stage are disjoint, and recover the expressions derived in prior works under sample-splitting assumptions.
>
> Finally, we will also provide an intuitive description of the technical results. In the high-dimensional limit, the $n$ components of the response vector $X\hat{w}$ are asymptotically independent, making it possible to describe the statistics of a single, scalar component. The scalar responses $u,r$ of a given component are given by eq. (6), and are driven by an underlying low-dimensional Gaussian process $g$, whose covariance structure is given by the parameters $q_0, q, \theta_0, \theta...$.
>
> >The authors claim their results make predictions, but it is unclear how to actually evaluate the predicted behavior.
>
> For the square loss, as we mention in the previous answer, the equations simplify, and yield a closed-form expression for the test error. For generic convex loss, the system of equations in Theorem 1.1 would generically need to be solved numerically, typically by repeatedly iterating the low-dimensional deterministic equations of Theorem 1.1 until convergence. We will add a comment on this point in the revised manuscript.
>
> > I have some issues with the framing (this could plausibly be a presentation feedback). [...] I think this framing is misleading. Similarly, I think including AdaBoost is mis-leading. I would remove this to more honestly reflect the main contribution of the paper: item 3 in the introduction on uncertainty-based active learning is the appropriate (and frankly, good enough) motivation.
>
> We will rephrase the introduction to more accurately reflect or main motivation, namely the study of data selection. We will mention briefly that we anticipate that the analysis can be extended to multi-stage scenarii, which will open the door to the study of optimization or iterative sample reweighting procedures.

---

### Official Review · Reviewer_Zbv5 · 2026-03-15

**Soundness:** 3
**Presentation:** 3
**Significance:** 2
**Originality:** 3
**Overall Recommendation:** 5
**Confidence:** 3

**Summary:**

This paper studies the statistical behavior of two-stage iterative ERM where a model is first trained using standard ERM and its predictions are then incorporated into the loss of the second ERM trained on the same data. Although such setup shows up commonly in machine learning applications e.g. active learning and optimization, it has been challenging to theoretically analyze due to the statistical dependency between the second stage loss and predictions from the first stage.

The authors provide the generalization performance of the two-stage iterative ERM for linear models with convex loss under Gaussian mixture covariates in Section 1. The main theorem shows that the test error converges to a deterministic limit.

This framework is applied to the pool-based active learning with a fixed labeling budget in Section 2. The theory provides several important insights: a tradeoff in how to allocate the labeling budget between stages and a selection-driven double-descent behavior. Experiments on synthetic data and a chest X-ray image classification task support the theoretical derivation.

**Compliance With Llm Reviewing Policy:**

Affirmed.

**Final Justification:**

Although there are some strong assumptions for its theoretical derivation, articulating iterated ERM in the perspective of active learning is a great contribution to the active learning community, and as the authors stated having such theoretical understanding is a necessary condition for principled development of active learning algorithms.

**Key Questions For Authors:**

1. Do the authors expect the theoretical framework to extend to multi-stage active learning framework? What are the main technical challenges to do so?

2. Have the authors considered to apply the theoretical result to more realistic setting like large-scale image classification using pre-trained deep feature extractors? What are the assumptions that are not applicable in this case? I expect all three assumptions can be met to the reasonable extent with a sophisticated design for the experiment setting

**Limitations:**

yes

**Strengths And Weaknesses:**

Strengths
1. The paper studies iterated ERM with important components considered: prediction dependent losses and data reuse. Despite common use of this framework, there have been limited attention in theoretical analysis
2. Unlike some previous theoretical works, the theoretical analysis provided in this paper does not rely on unrealistic assumptions e.g. oracle base classifiers or sample-splitting
3. The paper provides experiments on both synthetic and real X-ray image data qualitatively supporting the theoretical result

Weaknesses
1. Strong assumptions: the analysis replies on linear models, convex losses and Gaussian mixture covariates, which limit the applicability of the theoretical results to real-world deep learning applications
2. Restriction to two-stage: many real-world applications rely on multi-stage iterative training, but the theoretical result in this paper only considers two-stage ERM training
3. Practical take-away: due to the strong assumptions stated in the first weakness, it is hard to provide a practical guidance for real-world applications. For instance, although the theoretical result provides interesting insights for two-stage active learning for the binary classification task, it is unclear how to apply or extend the result to more general active learning tasks, or implement it as a concrete algorithm

---

> ### Author Rebuttal · Authors · 2026-03-31
>
> We thank the reviewer for their reading of our work and constructive comments. We address their questions below.
>
> >Strong assumptions: [...] which limit the applicability of the theoretical results to real-world deep learning applications
>
> Our work indeed relies on these assumptions to allow for a tractable analysis, and precisely clarify  the effects of statistical correlations caused by data re-use and selection in a widely studied problem in theoretical machine learning. To conduct such a fine-grained analysis, we restrict ourselves to the simple setting of linear models with convex loss. Already in this clean setting, the effects we evidence (budget tradeoff, double descent, and non-optimality of small-margin selection) are intricate, and contribute non-trivially to the theoretical understanding of data selection. A detailed theoretical analysis of these subtle effects in more complex neural networks indeed represents an ambitious next step that would warrant sizable additional technical work, that we leave to future investigations.
> Regarding the data assumptions, the tradeoff phenomena we uncover extend well beyond the Gaussian data assumption to real-world data, as we evidence in Fig. 2 for a X-ray scan dataset. Furthermore, several works, e.g.[a,b], have evidenced in the past how the learning curves derived from Gaussian mixture data assumptions could quantitatively capture simulations on real image data sets such as MNIST or CIFAR, or such data pre-processed by a neural network feature extractor, to echo the reviewer's last question, a quantitative match often refered to under the umbrella of "Gaussian universality" in the literature.
>
>
> >Do the authors expect the theoretical framework to extend to multi-stage active learning framework?
>
> We thank the reviewer for this very important question. We indeed think that an extension to the multi-stage case is possible, and are currently working on this. To provide more intuition on the way this extension can be carried out, note that the expression for the estimator after two stages (l.726) includes a correction term related to the statistical correlations created by the dependence on first stage predictions. This correction term takes the form of a product of (inverses of) two Hessian-type random matrices, whose trace $\chi$ intervenes in the final expression, see eq. 6. Similarly, in the multi-stage case, the $k-$ stage estimator involves products of up to $k$ such random matrices. To adapt the proof, one would thus need to similarly establish concentration of the corresponding normalized traces (in the likeness of Lemma 22), and derive the deterministic equivalent (Lemma 33). While this agenda is well-defined, it warrants non-trivial additional technical work, which falls out of the scope of the present manuscript. We will add a sketch of this generalization in the revised manuscript. A detailed study of the multi-stage case on the other hand requires significant technical work and new proof ideas, that warrant a separate subsequent work.
>
> > it is unclear how to apply or extend the result to more general active learning tasks, or implement it as a concrete algorithm
>
> We share the reviewer's hope that our results will eventually pave the way to new applications, although the aim of the present manuscript is primarily theoretical.
> Namely, our goal was to provide a refined analysis of a well-studied problem in the theoretical study of active learning, which was the object of several past works under much stronger and restrictive oracle assumptions. Already in this simple setting, the statistical effects of  data re-use and selection in the final performance are highly subtle. Our work affords a fine-grained analysis of these effects, and reveals several non-trivial phenomena (budget tradeoff, double-descent in composition, non-optimality of small-margin selection), thereby contributing to the theoretical comprehension of such algorithms. Arguably, such a theoretical understanding is a necessary condition to ensure a principled development of novel applications.
>
>
> >Have the authors considered to apply the theoretical result to more realistic setting like large-scale image classification using pre-trained deep feature extractors?
>
> We thank the reviewer for their suggestion of this additional experiment. Much like the experiment reported in Fig. 2 (for X-ray scans pre-processed by scattering transforms, which are feature extractors well-suited to image data), we expect that experiments with neural network feature extractors will similarly feature the U-shape characterizing the budget-allocation tradeoff discussed in subsection 2.3. We will include these additional experiments in the revised manuscript.
>
>
> [a] Loureiro et al, Learning Gaussian Mixtures with Generalised Linear Models: Precise Asymptotics in High-dimensions, 2021
>
> [b] Loureiro et al, Learning curves of generic features maps for realistic datasets with a teacher-student model, 2021

---

> > ### Author Rebuttal · Reviewer_Zbv5 · 2026-04-02
> >
> > I appreciate the authors for providing detailed answers to my questions. Although there are some strong assumptions for its theoretical derivation, articulating iterated ERM in the perspective of active learning is a great contribution to the active learning community, and as the authors stated having such theoretical understanding is a necessary condition for principled development of active learning algorithms. Therefore, I raised my score to 5.

---

### Decision · Program_Chairs · 2026-04-30

**Decision:**

Accept (regular)

**Comment:**

The paper studies two successive empirical risk minization problems performed on the same dataset, where the first estimator plays a role in the loss of the second estimator. An asymptotic characterization of the test error is provided. An application to the pool-based active learning problem is also given. Experimental evidence is provided.

The reviewers unanimously agree to accept this paper. For a camera-ready version, please take into the account the comments from all reviewers regarding for instance improvements on the presentation.